# Contextual AI models for single-cell protein biology

Michelle M. Li[1], Yepeng Huang[1], Marissa Sumathipala[1], Man Qing Liang[1], Alberto Valdeolivas ⓘ [2], Ashwin N. Ananthakrishnan[1,3], Katherine Liao[1,4], Daniel Marbach ⓘ [2] & Marinka Zitnik ⓘ [1,5,6,7] ✉

Understanding protein function and developing molecular therapies require deciphering the cell types in which proteins act as well as the interactions between proteins. However, modeling protein interactions across biological contexts remains challenging for existing algorithms. Here we introduce PINNACLE, a geometric deep learning approach that generates context-aware protein representations. Leveraging a multiorgan single-cell atlas, PINNACLE learns on contextualized protein interaction networks to produce 394,760 protein representations from 156 cell type contexts across 24 tissues. PINNACLE's embedding space reflects cellular and tissue organization, enabling zero-shot retrieval of the tissue hierarchy. Pretrained protein representations can be adapted for downstream tasks: enhancing 3D structure-based representations for resolving immuno-oncological protein interactions, and investigating drugs' effects across cell types. PINNACLE outperforms state-of-the-art models in nominating therapeutic targets for rheumatoid arthritis and inflammatory bowel diseases and pinpoints cell type contexts with higher predictive capability than context-free models. PINNACLE's ability to adjust its outputs on the basis of the context in which it operates paves the way for large-scale context-specific predictions in biology.

Proteins are the functional units of cells, and their interactions enable different biological functions. The development of high-throughput methods has facilitated the characterization of large maps of protein interactions. Leveraging these protein interaction networks, computational methods[1,2] have been developed to improve the understanding of protein structure[3], accurately predict functional annotations[4,5] and inform the design of therapeutic targets[6,7]. Among them, representation learning methods have emerged as a leading strategy to model proteins[8–10]. These approaches can resolve protein interaction networks across tissues[11–13] and cell types by integrating molecular cell atlases[14] and extending our understanding of the relationship between protein and function[15]. Protein representation learning methods can predict multicellular functions across human tissues[12], design target-binding proteins[16] and novel protein interactions[17], and predict interactions between transcription factors and genes[15].

Proteins can have distinct roles in different biological contexts[18,19]. While nearly every cell contains the same genome, the expression of genes and the function of proteins encoded by these genes depend on cellular and tissue contexts[11,20,21]. Gene expression and the function of proteins can also differ significantly between healthy and disease states[21,22]. Methods incorporating biological contexts can improve the characterization of proteins and provide precise, context-specific

[1]Department of Biomedical Informatics, Harvard Medical School, Boston, MA, USA. [2]Roche Pharma Research and Early Development, Pharmaceutical Sciences, Roche Innovation Center Basel, F. Hoffmann-La Roche Ltd, Basel, Switzerland. [3]Division of Gastroenterology, Massachusetts General Hospital, Boston, MA, USA. [4]Division of Rheumatology, Inflammation, and Immunity, Brigham and Women's Hospital, Boston, MA, USA. [5]Kempner Institute for the Study of Natural and Artificial Intelligence, Harvard University, Allston, MA, USA. [6]Broad Institute of MIT and Harvard, Cambridge, MA, USA. [7]Harvard Data Science Initiative, Cambridge, MA, USA. ✉e-mail: marinka@hms.harvard.edu

insights. However, deep learning methods produce protein representations (or embeddings) that are context-free: each protein has only one representation learned from either a single context or an integrated view across many contexts[15,23]. These methods generate one representation for each protein, providing an integrated summary. Context-free protein representations are not tailored to specific biological contexts, such as cell types and disease states. These representations cannot identify protein functions that vary across different cell types, which in turn hamper the prediction of pleiotropy and protein roles in a cell type-specific manner.

Sequencing technologies that measure gene expression with single-cell resolution pave the way toward addressing this challenge. Single-cell transcriptomic atlases[20,24–27] measure activated genes across many cellular contexts. Through attention-based deep learning[28,29], which specifies models that can pay attention to large inputs and learn the most important elements to focus on in each context, single-cell atlases can be leveraged to boost the mapping of gene regulatory networks that drive disease progression and reveal treatment targets[30]. However, incorporating the expression of protein-coding genes into protein interaction networks remains a challenge. Existing algorithms, including protein representation learning, cannot contextualize protein representations.

We introduce PINNACLE (Protein Network-based Algorithm for Contextual Learning), a context-specific model for comprehensive protein understanding. PINNACLE is a geometric deep learning model adept at generating protein representations through the analysis of protein interactions within various cellular contexts. Leveraging single-cell transcriptomics combined with networks of protein–protein interactions (PPIs), cell type-to-cell type interactions and a tissue hierarchy, PINNACLE generates high-resolution protein representations tailored to each cell type. In contrast to existing methods that provide a single representation for each protein, PINNACLE generates a distinct representation for each cell type in which a protein-coding gene is activated. With 394,760 contextualized protein representations produced by PINNACLE, where each protein representation is imbued with cell type specificity, we demonstrate PINNACLE's capability to integrate protein interactions with the underlying protein-coding gene transcriptomes of 156 cell type contexts. PINNACLE models support a broad array of tasks; they can enhance three-dimensional (3D) structural protein representations, analyze the effects of drugs across cell type contexts, nominate therapeutic targets in a cell type-specific manner, retrieve tissue hierarchy in a zero-shot manner and perform context-specific transfer learning. PINNACLE models dynamically adjust their outputs on the basis of the context in which they operate and can pave the way for the broad use of foundation models tailored to diverse biological contexts.

## Results

### Constructing context-specific networks

Generating protein representations embedded with cell type context calls for protein interaction networks that consider the same context. We assembled a dataset of context-sensitive protein interactomes, beginning with a multiorgan single-cell transcriptomic atlas[20] that encompasses 24 tissue and organ samples sourced from 15 human donors (Fig. 1a). We compile activated genes for every expert-annotated cell type in this dataset by evaluating the average gene expression in cells from that cell type relative to a designated reference set of cells (Fig. 1a and 'Construction of multiscale networks' section in Methods). Here, 'activated genes' are defined as those demonstrating a higher average expression in cells annotated as a particular type than the remaining cells documented in the dataset. Based on these activated gene lists, we extracted the corresponding proteins from the comprehensive reference protein interaction network and retained the largest connected component (Fig. 1a). As a result, we have 156 context-aware protein interaction networks, each with 2,530 ± 677 proteins, that are

maximally similar to the global reference protein interaction network and still highly cell type specific (Extended Data Figs. 1 and 2). Our context-aware protein interaction networks from 156 cell type contexts span 62 tissues of varying biological scales.

Further, we constructed a network of cell types and tissues (metagraph) to model cellular interactions and the tissue hierarchy ('Construction of multiscale networks' section in Methods). Given the cell type annotations designated by the multiorgan transcriptomic atlas[20], the network consists of 156 cell type nodes. We incorporated edges between pairs of cell types based on the existence of significant ligand–receptor (LR) interactions and validated that the proteins correlating to these interactions are enriched in the context-aware protein interaction networks in comparison to a null distribution ('Construction of multiscale networks' section in Methods and Extended Data Fig. 1c,d). Leveraging information on tissues in which the cell types were measured, we began with 24 tissue nodes and established edges between cell type nodes and tissue nodes if the cell type was derived from the corresponding tissue. We then identified all ancestor nodes, including the root, of the 24 tissue nodes within the tissue hierarchy ('Construction of multiscale networks' section in Methods) to feature 62 tissue nodes interconnected by parent–child relationships. Our dataset thus comprises 156 context-aware protein interaction networks and a metagraph reflecting cell type and tissue organization.

### Overview of PINNACLE model

PINNACLE is a geometric deep learning model capable of generating protein representations predicated on protein interactions within a spectrum of cell type contexts. Trained on an integrated set of context-aware protein interaction networks, complemented by a network capturing cellular interactions and tissue hierarchy (Fig. 1b,c), PINNACLE generates contextualized protein representations that are tailored to cell types in which protein-coding genes are activated (Fig. 1d). Unlike context-free models, PINNACLE produces multiple representations for every protein, each contingent on its specific cell type context. Additionally, PINNACLE produces representations of the cell type contexts and representations of the tissue hierarchy (Fig. 1d,e). This approach ensures a multifaceted understanding of protein interaction networks, taking into account the myriad of contexts in which proteins act.

Given multiscale model inputs, PINNACLE learns the topology of proteins, cell types and tissues by optimizing a unified latent representation space. PINNACLE integrates different context-specific data into one context-aware model (Fig. 1f) and transfers knowledge between protein-, cell type- and tissue-level data to contextualize representations (Fig. 1g). To infuse cellular and tissue organization into this embedding space, PINNACLE employs protein-, cell type- and tissue-level attention along with respective objective functions (Fig. 1b,c and 'Multiscale graph neural network' section in Methods). Conceptually, pairs of proteins that physically interact (that is, are connected by edges in input networks) are closely embedded. Similarly, proteins are embedded near their respective cell type contexts while maintaining a substantial distance from unrelated ones. This ensures that interacting proteins within the same cell type context are situated proximally within the embedding space yet are separated from proteins from other cell type contexts. This approach yields an embedding space that accurately represents the intricacies of relationships between proteins, cell types and tissues.

PINNACLE disseminates graph neural network messages between proteins, cell types and tissues using a series of attention mechanisms tailored to each specific node and edge type ('Multiscale graph neural network' section in Methods). The protein-level pretraining tasks consider self-supervised link prediction on protein interactions and cell type classification on protein nodes. These tasks enable PINNACLE to sculpt an embedding space that encapsulates the topology of the context-aware protein interaction networks and the cell type identity

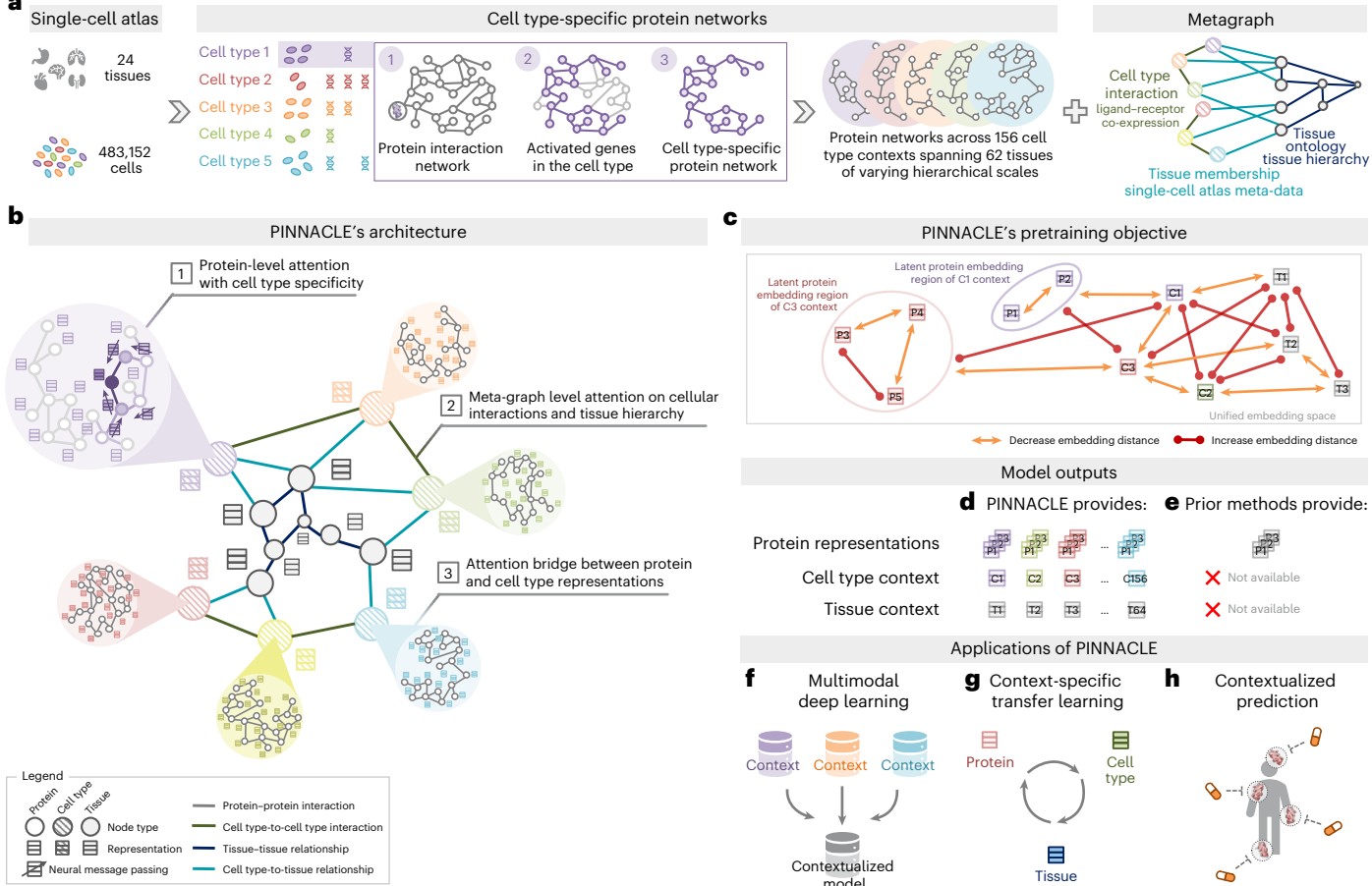

**Fig. 1 | Overview of PINNACLE. a**, Cell type-specific protein interaction networks and metagraph of cell type and tissue organization are constructed from a multiorgan single-cell transcriptomic atlas of humans, a human reference protein interaction network and a tissue ontology. **b**, PINNACLE has protein-, cell type- and tissue-level attention mechanisms that enable the algorithm to generate contextualized representations of proteins, cell types and tissues in a single unified embedding space. **c**, PINNACLE is designed such that the nodes (that is, proteins, cell types and tissues) that share an edge are embedded closer (decreased embedding distance) to each other than nodes that do not share an edge (increased embedding distance); proteins activated in the same cell type are embedded more closely (decreased embedding distance) than proteins activated in different cell types (increased embedding distance), and cell types are embedded closer to their activated proteins (decreased embedding distance) than other proteins (increased embedding distance). **d**, As a result, PINNACLE generates protein representations injected with cell type and tissue context; a unique representation is produced for each protein activated in each cell type. PINNACLE simultaneously generates representations for cell types and tissues. **e**, Existing methods, however, are context-free. They generate a single embedding per protein, representing only one condition or context for each protein, without any notion of cell type or tissue context. **f**–**h**, The PINNACLE algorithm and its outputs enable multimodal deep learning (for example, single-cell transcriptomic data with interactomes) (**f**), context-specific transfer learning (for example, between proteins, cell types and tissues) (**g**) and contextualized predictions (for example, efficacy and safety of therapeutics) (**h**).

of the proteins. PINNACLE's cell type- and tissue-specific pretraining tasks rely exclusively on self-supervised link prediction, facilitating the learning of cellular and tissue organization. The topology of cell types and tissues is imparted to the protein representations through an attention bridge mechanism, effectively enforcing tissue and cellular organization onto the protein representations. PINNACLE's contextualized protein representations capture the structure of context-aware protein interaction networks. The regional arrangement of these contextualized protein representations in the latent space reflects the cellular and tissue organization represented by the metagraph. This leads to a comprehensive and context-specific representation of proteins within a unified cell type- and tissue-specific framework.

## PINNACLE captures cellular and tissue organization

PINNACLE generates protein representations for each of the 156 cell type contexts spanning 62 tissues of varying hierarchical scales. In total, PINNACLE's unified multiscale embedding space comprises 394,760 protein representations, 156 cell type representations and 62 tissue representations (Fig. 1a). We show that PINNACLE learns an embedding space where proteins are positioned based on cell type context. We first quantify the spatial enrichment of PINNACLE's protein embedding regions using a systematic method, SAFE[31] ('Spatial enrichment analysis of PINNACLE's protein embeddings' section in Methods). PINNACLE's contextualized protein representations self-organize in PINNACLE's embedding space as evidenced by the enrichment of spatial embedding regions for protein representations that originate from the same cell type context (significance cutoff $\alpha = 0.05$; Fig. 2 and Extended Data Figs. 3 and 4).

Next, we evaluate embedding regions to confirm that they are separated by cell type and tissue identity by calculating the similarities between protein representations across cell type contexts. Protein representations from the same cell type are more similar than those from different cell types (Fig. 3a). In contrast, a model without cellular or tissue context fails to capture any differences between protein representations across cell type contexts (Fig. 3b). Further, we expect the representations of proteins that act on multiple cell types to be highly

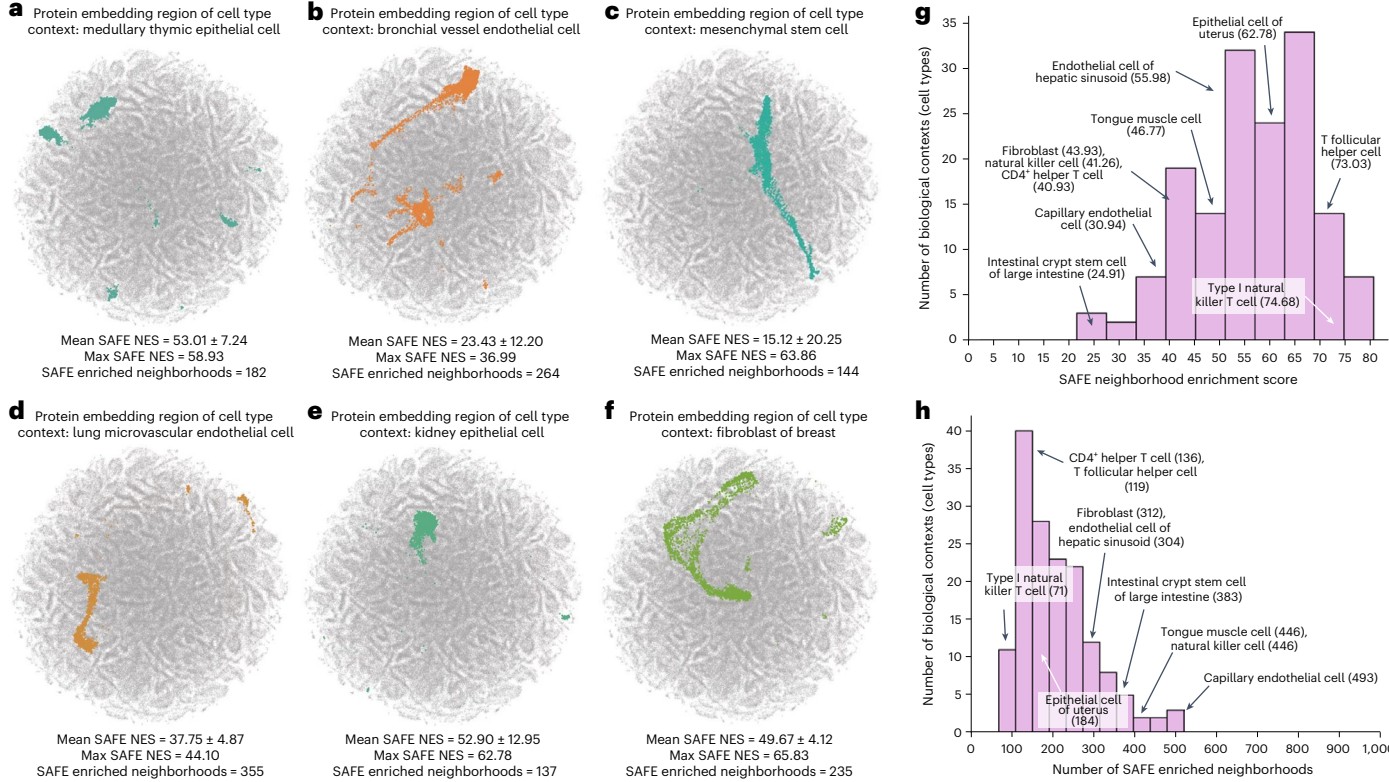

**Fig. 2 | Enrichment of PINNACLE's protein embedding regions. a–f,** Two-dimensional UMAP plots of contextualized protein representations generated by PINNACLE from six different cell type contexts: medullary thymic epithelial cell (**a**), bronchial vessel endothelial cell (**b**), mesenchymal stem cell (**c**), lung microvascular endothelial cell (**d**), kidney epithelial cell (**e**) and fibroblast of breast (**f**). Each dot is a protein representation. Colored dots indicate cell type context regions, and gray dots represent proteins from other cell types. Each protein embedding region is expected to be enriched neighborhoods that are spatially localized according to cell type context. To quantify this, we compute spatial enrichment of each protein embedding region using SAFE[31] and provide the mean and max neighborhood enrichment scores (NES) and the number of enriched neighborhoods output by the tool ('Metrics and statistical analyses' section in Methods and Extended Data Figs. 3 and 4). **g,h,** Distribution of the maximum SAFE NES (**g**) and the number of enriched neighborhoods (**h**) for 156 cell type contexts (each context has a $P$ value <0.05; hypergeometric test, adjusted using the Benjamini–Hochberg false discovery rate correction with significance cutoff $\alpha = 0.05$). Ten randomly sampled cell type contexts are annotated, with their maximum SAFE NES or number of enriched neighborhoods in parentheses.

dissimilar, reflecting specialized cell type-specific protein functions (Supplementary Note 1). We calculate the similarities of protein representations (that is, cosine similarities of a protein's representations across cell type contexts) based on the number of cell types in which the protein is active (Extended Data Fig. 5a,b). Representational similarities of proteins negatively correlate with the number of cell types in which they act (Spearman's $\rho = -0.9798$; $P < 0.001$), and the correlation is weaker in the ablated model with cellular and tissue metagraph turned off (Spearman's $\rho = -0.6334$; $P < 0.001$).

We additionally examine whether protein embedding regions are organized by the tissue hierarchy. We leverage PINNACLE's tissue representations to perform zero-shot retrieval of the tissue hierarchy and then compare tissue ontology distance to tissue embedding distance. Tissue ontology distance is defined as the sum of the shortest path lengths from two tissue nodes to the lowest common ancestor node in the tissue hierarchy, and tissue embedding distance is the cosine distance between the corresponding tissue representations. We expect a positive correlation: the farther apart the nodes are according to the tissue hierarchy, the more dissimilar the tissue representations are. As hypothesized, embedding distances in the latent space and the corresponding distances in the tissue ontology of the same tissues are positively correlated (Spearman's $\rho = 0.36$; $P = 1.85 \times 10^{-119}$; Fig. 3c), and the distribution of tissue embedding distances cannot be attributed to random effects (Kolmogorov–Smirnov two-sided test 0.50; $P < 0.001$). When the tissue ontology is randomly shuffled, the correlation with distances in the embedding space diminishes significantly (Spearman's $\rho = 0.005$; $P = 0.349$; Fig. 3c). Since PINNACLE uses the metagraph to systematically integrate tissue organization into both cell type and protein representations, it follows that all of PINNACLE's representations inherently reflect this tissue organization ('Multiscale graph neural network' section in Methods and Extended Data Fig. 6).

## PINNACLE enhances 3D structural representations of PPIs

Protein–protein interactions (PPIs) depend on both 3D structure conformations of the proteins[32,33] and cell type contexts within which the proteins act[34]. However, protein representations produced by existing artificial intelligence (AI) models based on 3D molecular structures lack cell type context information. We hypothesize that incorporating cellular context information can better differentiate binding from nonbinding proteins (Fig. 3d). Because 3D structures of molecules (containing precise atom or residue level contact information) provide complementary knowledge to PPI networks (summarizing binary interactions between proteins), we expect that context-aware protein interaction networks can improve the ability to differentiate between binding and nonbinding proteins across different cell types[35]. As no large-scale dataset with matched structural biology and genomic readouts currently exists to perform systematic analyses, we focus on PD-1/PD-L1 and B7-1/CTLA-4 interacting proteins, important immune checkpoint protein interactors involved in cancer immunotherapies[36].

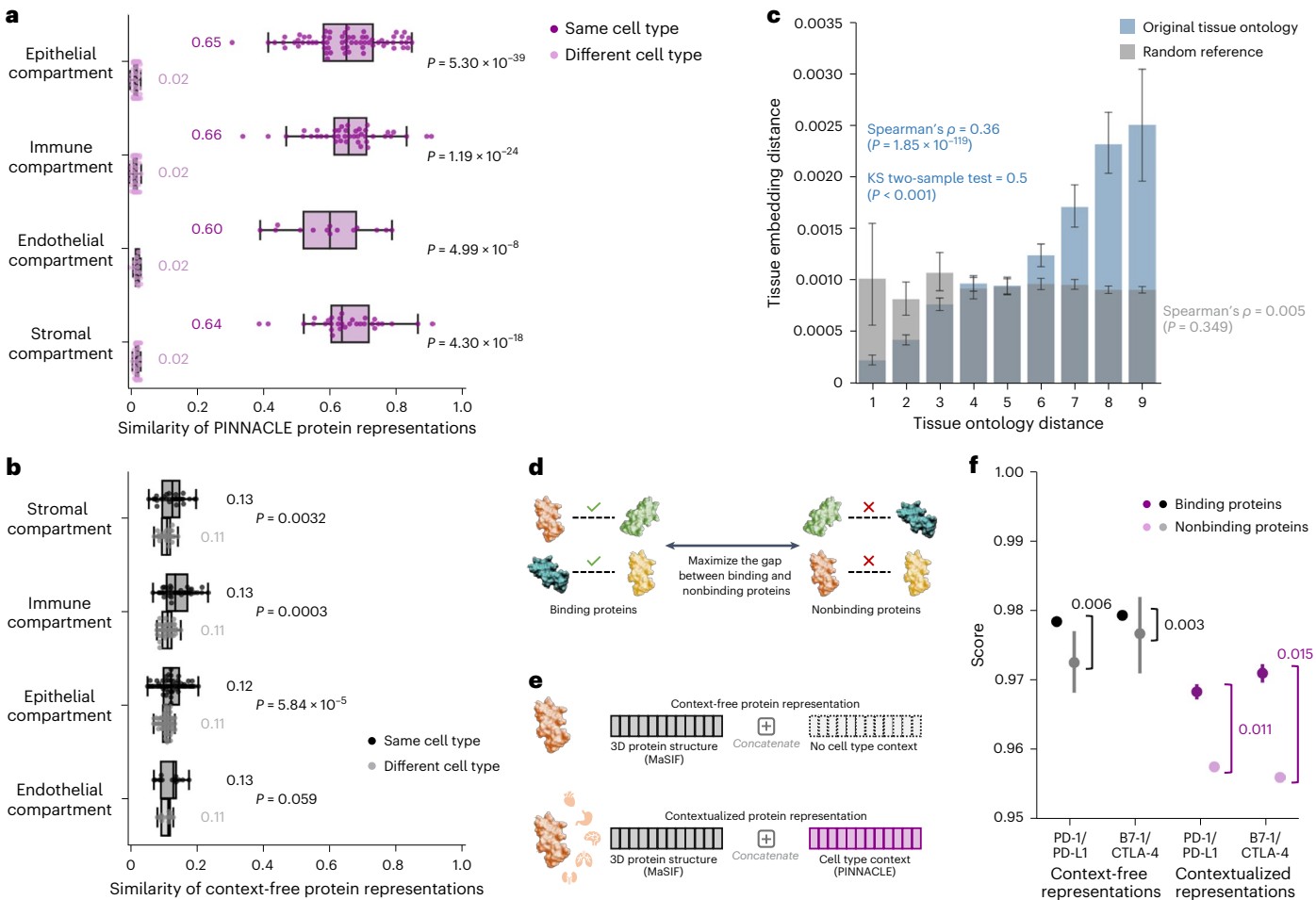

**Fig. 3 | Evaluation of PINNACLE's contextual representations. a,b**, Gap between embedding similarities using PINNACLE's protein representations (**a**) and a noncontextualized model's protein representations (**b**) on $n = 394{,}760$ samples (that is, cell type-specific protein representations). Similarities are calculated between pairs of proteins in the same cell type (dark shade of color) or different cell types (light shade of color), and stratified by the compartment from which the cell types are derived. We use the two-sided two-sample Kolmogorov–Smirnov test for goodness of fit. Annotations indicate median values. The noncontextualized model is an ablated version of PINNACLE without any notion of tissue or cell type organization (that is, remove cell type and tissue network and all cell type- and tissue-related components of PINNACLE's architecture and objective function). The bounds of the box show the quartiles of the data, the center indicates the median value of the data and the whiskers represent the farthest data point within $1.5 \times$ interquartile range. **c**, Embedding distance of PINNACLE's 62 tissue representations as a function of tissue ontology distance. The gray bars indicate a null distribution (refer to 'Metrics and statistical analyses' section in Methods for more details). Both the Spearman correlation ($P = 1.85 \times 10^{-119}$) and Kolmogorov–Smirnov ($P < 0.001$) statistical tests are two-sided. The data are represented as mean values with error bars indicating a 95% confidence interval. **d**, Prediction task in which protein representations

are optimized to maximize the gap between binding and nonbinding proteins. **e**, Cell type context (provided by PINNACLE) is injected into context-free structure-based protein representations (provided by MaSIF[3], which learns a protein representation from the protein's 3D structure) via concatenation to generate contextualized protein representations. Lack of cell type context is defined by an average of PINNACLE's protein representations. **f**, Comparison of context-free and contextualized representations in differentiating between binding and nonbinding proteins. The scores are computed using cosine similarity on $n = 22$ unique protein pairs (2 binding and 20 nonbinding); since PINNACLE generates multiple representations per protein based on context, there are $n = 7{,}956$ pairwise computations (180 binding and 7,776 nonbinding) for the contextualized representations. The binding proteins evaluated are PD-1/PD-L1 and B7-1/CTLA-4. Pairwise scores also are calculated for each of these four proteins and proteins that they do not bind with (that is, RalB, RalBP1, EPO, EPOR, C3 and CFH). The gap between the average scores of binding and nonbinding proteins is annotated for context-free and contextualized representations. The significance of the score gaps between binding and nonbinding proteins is measured using a one-sided nonparametric permutation test. The data are represented as mean values with error bars indicating a 95% confidence interval.

We compare contextualized and context-free protein representations for binding proteins (that is, PD-1/PD-L1 and B7-1/CTLA-4) and nonbinding proteins (that is, one of the four binding proteins paired with RalB, RalBP1, EPO, EPOR, C3 or CFH). Cell type context is incorporated into 3D structure-based protein representations[3,17] by concatenating them with PINNACLE's protein representation (Fig. 3e and 'Generating contextualized 3D protein representations' section in Methods). Context-free protein representations are generated by concatenating 3D structure-based representations[3,17] with an average of PINNACLE's protein representations across all cell type

contexts ('Generating contextualized 3D protein representations' section in Methods). Contextualized representations, resulting from a combination of protein representations based on 3D structure and context-aware PPI networks, give scores (via cosine similarity) for binding and nonbinding proteins of $0.9690 \pm 0.0049$ and $0.9571 \pm 0.0127$, respectively. Using PINNACLE's context-specific protein representations, which have no 3D structure information, binding and nonbinding proteins are scored $0.0385 \pm 0.1531$ and $0.0218 \pm 0.1081$, respectively. In contrast, using context-free representations, binding and nonbinding proteins are scored at $0.9789 \pm 0.0004$ and

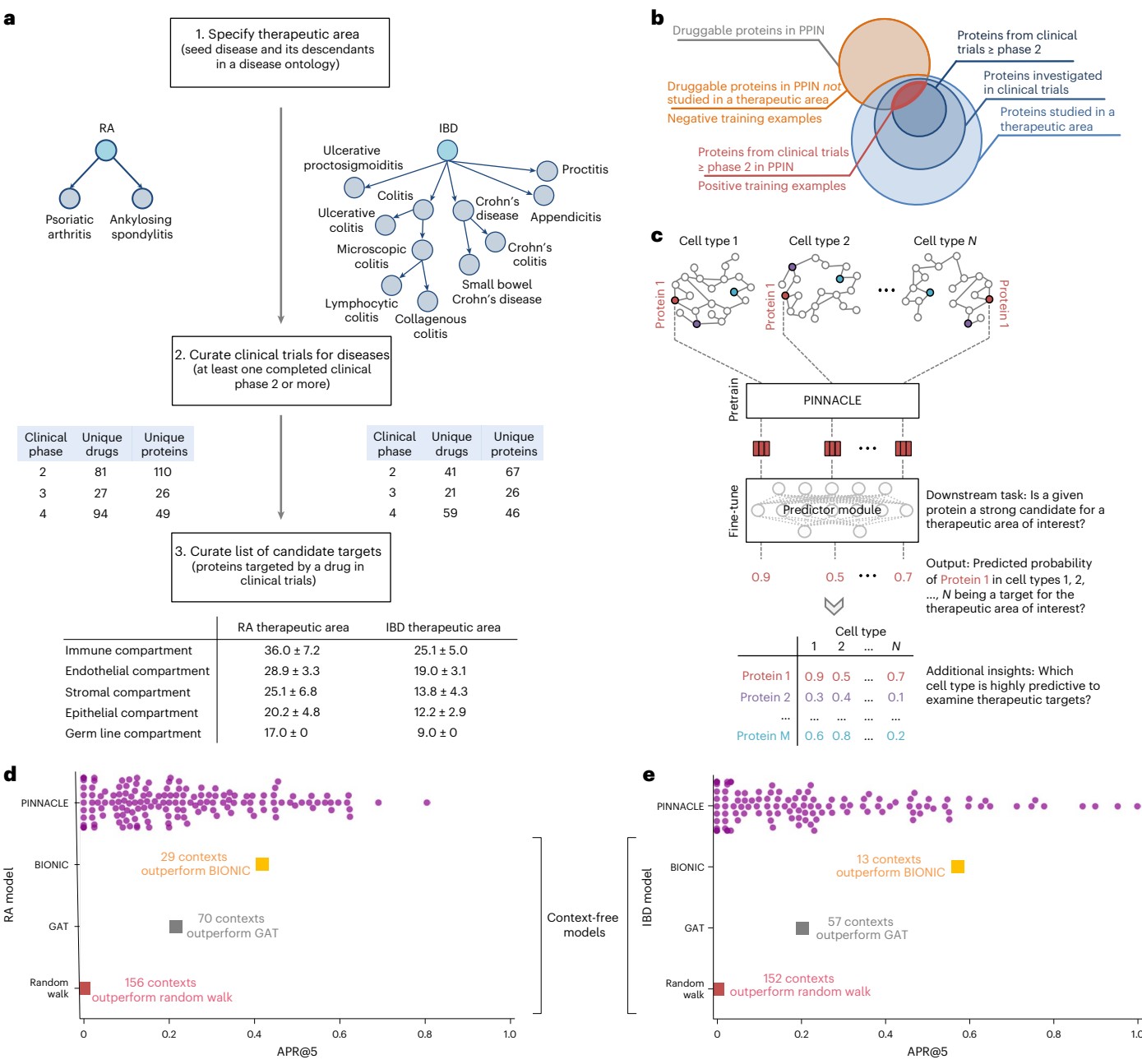

**Fig. 4 | Fine-tuning contextualized protein representations for therapeutic target prioritization. a**, Workflow to curate positive training examples for RA (left) and IBD (right) therapeutic areas. **b**, We construct positive examples by selecting proteins from our protein–protein interaction network (PPIN) that are targeted by compounds that have at least completed phase 2 for treating the therapeutic area of interest. These proteins are deemed safe and potentially efficacious for humans with the disease. We construct negative examples by selecting proteins from our PPIN that do not have associations with the therapeutic area yet have been targeted by at least one existing drug/compound. **c**, Cell type-specific protein interaction networks are embedded by PINNACLE,

and fine-tuned for a downstream task. Here, the predictor module (that is, MLP) fine-tunes the (pretrained) contextualized protein representations for predicting whether a given protein is a strong candidate for the therapeutic area of interest. Additional insights of our setup include hypothesizing highly predictive cell types for examining candidate therapeutic targets. **d,e**, Benchmarking of context-aware and context-free approaches for RA (**d**) and IBD (**e**) therapeutic areas. Each dot is the performance (averaged across ten random seeds) of protein representations from a given context (that is, cell type context for PINNACLE, context-free global reference protein interaction network for GAT and random walk, and context-free multimodal protein interaction network for BIONIC).

0.9742 ± 0.0078, respectively. Further, comparative analysis of the gap in scores between interacting versus noninteracting proteins yields gaps of 0.011 (PD-1/PD-L1) and 0.015 (B7-1/CTLA-4) for PINNACLE's contextualized representations (*P* = 0.0299; Extended Data Fig. 7), yet only 0.003 (PD-1/PD-L1) and 0.006 (B7-1/CTLA-4) for context-free representations (Fig. 3f and Extended Data Fig. 7). Incorporating information about biological contexts can help better distinguish protein interactions from noninteracting proteins in

specific cell types, suggesting that PINNACLE's contextualized representations can enhance protein representations derived from 3D protein structure modality. Modeling context-dependent interactions involving immune checkpoint proteins can deepen our understanding of how these proteins are used in cancer immunotherapies. Our benchmarking results further suggest that incorporating context can improve 3D structure prediction of protein interactions (Supplementary Note 2).

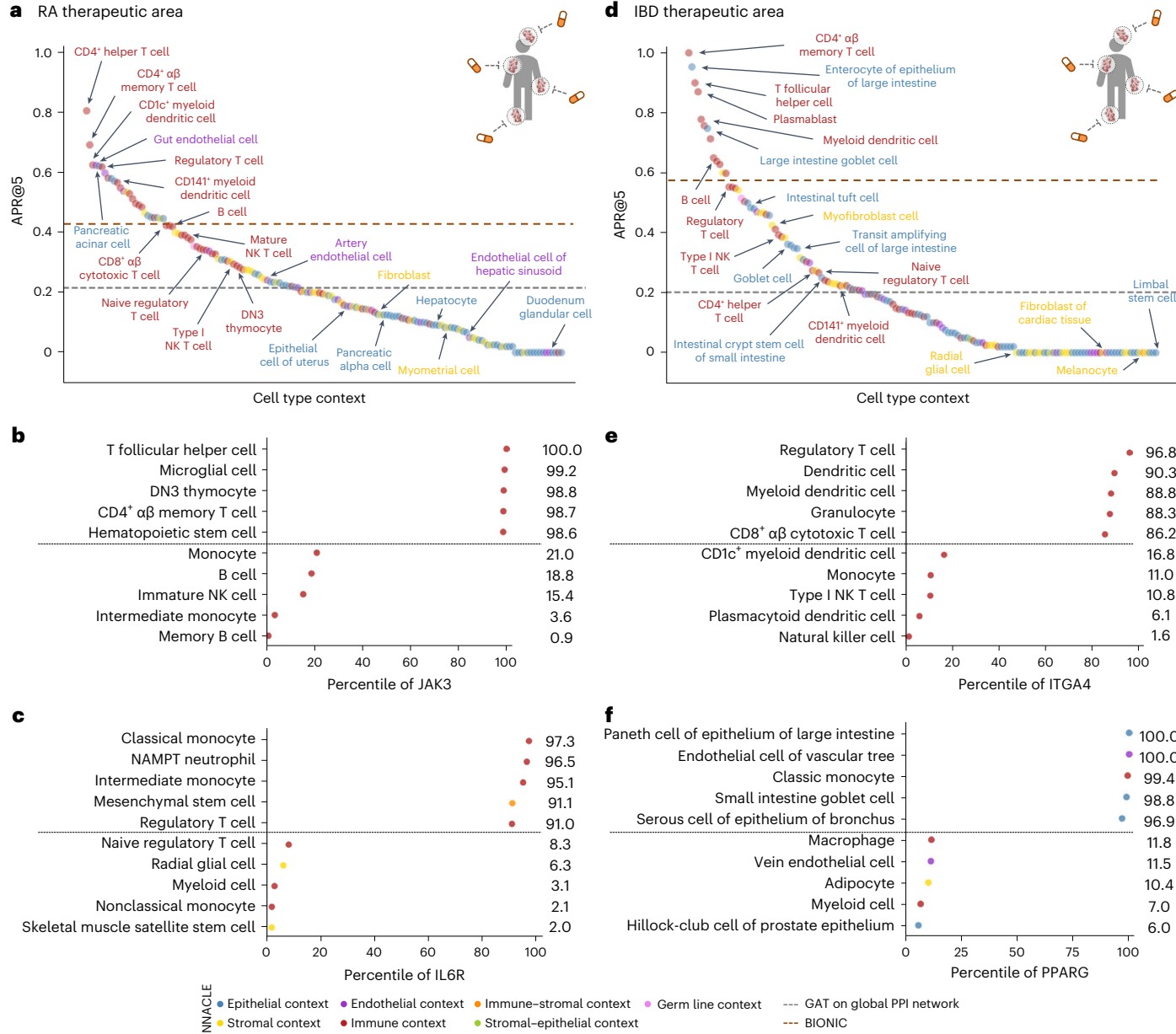

**Fig. 5 | Performance of contextualized target prioritization for RA and IBD therapeutic areas. a,d**, Model performance (measured by APR@5) for RA (**a**) and IBD (**d**) therapeutic areas, respectively. APR@K (or Average Precision and Recall at K) is a combination of Precision@K and Recall@K (refer to 'Metrics and statistical analyses' section in Methods for more details). Each dot is the performance (averaged across ten random seeds) of PINNACLE's protein representations from a specific cell type context. The gray and dark-orange lines are the performance of the GAT and BIONIC models, respectively. For each therapeutic area, 22 cell types are annotated and colored by their compartment category. Extended Data Fig. 8 contains model performance measured by APR@10, APR@15 and APR@20 for RA and IBD therapeutic areas. **b,c,e,f**, Selected proteins for RA and IBD therapeutic areas, where the horizontal solid line separates the top and bottom five cell types: two selected proteins, JAK3 (**b**) and IL6R (**c**), that are targeted by drugs that have completed phase IV of clinical trials for treating RA therapeutic area; two selected proteins, ITGA4 (**e**) and PPARG (**f**), that are targeted by drugs that have completed phase IV for treating IBD therapeutic area.

## Contextual models outperform context-free target prediction

With the representations from PINNACLE infused with cellular and tissue context, we can fine-tune them for downstream tasks (Fig. 1f–h). We hypothesize that PINNACLE's contextualized latent space can better differentiate between therapeutic targets and proteins with no therapeutic potential than a context-free latent space. Here, we focus on modeling the therapeutic potential of proteins across cell types for therapeutic areas with cell type-specific mechanisms of action (MoA) (Fig. 4). Certain cell types are known to play crucial and distinct roles in the disease pathogenesis of rheumatoid arthritis

(RA) and inflammatory bowel disease (IBD) therapeutic areas[24,37–40]. There is currently no cure for either type of condition, and the medications prescribed to mitigate the symptoms can lead to undesired side effects[41]. The new generation of therapeutics in development for RA and IBD conditions is designed to target specific cell types so that the drugs maximize efficacy and minimize adverse events (for example, by directly impacting the affected/responsible cells and avoiding off-target effects on other cells)[41,42]. We adopt PINNACLE models to predict the therapeutic potential of proteins in a cell type-specific manner.

We fine-tune PINNACLE to predict therapeutic targets for RA and IBD diseases. Specifically, we perform binary classification on each contextualized protein representation, where $y = 1$ indicates that the protein is a therapeutic candidate for the given therapeutic area and $y = 0$ otherwise. The ground truth positive examples (where $y = 1$) are proteins targeted by drugs that have at least completed one clinical trial of phase 2 or higher for indications under the therapeutic area of interest, indicating that the drugs are safe and potentially efficacious in an initial cohort of humans (Fig. 4a,b). The negative examples (where $y = 0$) are druggable proteins that have not been studied for the therapeutic area (Fig. 4b and 'Fine-tuning PINNACLE for context-specific target prioritization' section in Methods). The binary classification model can be of any architecture; our results for nominating RA and IBD therapeutic targets are generated by a multilayer perceptron (MLP) trained for each therapeutic area (Fig. 4c).

To evaluate PINNACLE's contextualized protein representations, we compare PINNACLE's fine-tuned models against three context-free models. We apply a random walk algorithm[43] and a graph attention network (GAT)[44] on the context-free reference protein interaction network. The BIONIC model is a graph convolutional neural network designed for (context-free) multimodal network integration[15].

We find that PINNACLE's protein representations for all cell type contexts outperform the random walk model for both RA (Fig. 4d) and IBD (Fig. 4e) diseases. Protein representations from 44.9% (70 out of 156) and 37.5% (57 out of 152) cell types outperform the GAT model for RA (Fig. 4d) and IBD (Fig. 4e) diseases, respectively. Although both PINNACLE and BIONIC can integrate the 156 cell type-specific protein interaction networks, PINNACLE's protein representations outperform BIONIC[15] in 18.6% of cell types (29 out of 156) and 8.6% of cell types (13 out of 152) for RA (Fig. 4d) and IBD diseases (Fig. 4e), respectively, highlighting the utility of contextualizing protein representations. PINNACLE outperforms these three context-free models via other metrics for both RA and IBD therapeutic areas (Extended Data Fig. 8). We have confirmed no significant correlation between the node degree of proteins in cell type-specific PPI networks and performance in RA and IBD models (Extended Data Fig. 9a). Additionally, there is only a moderate correlation between PINNACLE's performance and the enrichment of positive targets in these cell type-specific PPI networks (Extended Data Fig. 9b,c). These findings underscore that PINNACLE's predictions cannot be solely ascribed to the characteristics of the cell type-specific PPI networks. Benchmarking results indicate combining global reference networks with advanced deep graph representation learning techniques, such as GAT, can yield better predictors than network-based random walk methods alone. Integrative approaches, exemplified by methods such as BIONIC, enhance performance, a finding consistent with the established benefits of data integration. Contextualized learning approaches, such as PINNACLE, have the potential to enhance model performance and enable predictions tailored to specific contexts.

## PINNACLE can nominate targets across cell type contexts

There is existing evidence that drug effects vary with cell type depending on where therapeutic targets are expressed and where proteins act[45–49]. For instance, CD19-targeting chimeric antigen receptor T cell therapy has been highly effective in treating B cell malignancies yet causes a high incidence of neurotoxicity[47]. A recent study shows that chimeric antigen receptor T cells induce off-target effects by targeting the CD19 expressed in brain mural cells, probably causing the brain barrier leakiness responsible for neurotoxicity[47]. We hypothesize that the predicted protein druggability varies across cell types, and such variations can provide insights into the cell types' relevance for a therapeutic area.

Among the 156 biological contexts modeled by PINNACLE's protein representations, we examine the most predictive cell type contexts for nominating therapeutic targets of RA. We find that the most predictive

contexts consist of CD4+ helper T cells, CD4+ αβ memory T cells, CD1c+ myeloid dendritic cells, gut endothelial cells and pancreatic acinar cells (Fig. 5a). Immune cells play a significant role in the disease pathogenesis of RA[37,38]. Since CD4+ helper T cells (PINNACLE-predicted rank 1), CD4+ αβ memory T cells (PINNACLE-predicted rank 2) and CD1c+ myeloid dendritic cells (PINNACLE-predicted rank 3) are immune cells, it is expected that PINNACLE's protein representations in these contexts achieve high performance in our prediction task. Also, patients with RA often have gastrointestinal (GI) manifestations, whether concomitant GI autoimmune diseases or GI side effects of RA treatment[50]. Pancreatic acinar cells (PINNACLE-predicted rank 5) can behave like inflammatory cells during acute pancreatitis[51], one of the accompanying GI manifestations of RA[50]. In addition to GI manifestations, endothelial dysfunction is commonly detected in patients with RA[52]. While rare, rheumatoid vasculitis, which affects endothelial cells and is a serious complication of RA, has been found to manifest in the large and small intestines, liver and gallbladder[50,53]. Further, many of the implicated cell types for patients with RA (for example, T cells, B cells, natural killer cells, monocytes, myeloid cells and dendritic cells) are highly ranked by PINNACLE[24,25,39] (Supplementary Table 1). Our results suggest that injecting cell type context to protein representations can significantly improve performance in nominating therapeutic targets for RA diseases while potentially revealing the cell types underlying disease processes.

The most predictive cell type contexts for nominating therapeutic targets of IBD are CD4+ αβ memory T cells, enterocytes of epithelium of large intestine, T follicular helper cells, plasmablasts and myeloid dendritic cells (Fig. 5d). The intestinal barrier comprises a thick mucus layer with antimicrobial products, a layer of intestinal epithelial cells and a layer of mesenchymal cells, dendritic cells, lymphocytes and macrophages[54]. As such, these five cell types are expected to yield high predictive ability. Moreover, many of the implicated cell types for IBD (for example, T cells, fibroblasts, goblet cells, enterocytes, monocytes, natural killer cells, B cells and glial cells) are highly ranked by PINNACLE[26,27,55] (Supplementary Table 2). For example, CD4+ T cells are known to be the main drivers of IBD[56]. They have been found in the peripheral blood and intestinal mucosa of adult and pediatric patients with IBD[57]. Patients with IBD tend to develop uncontrolled inflammatory CD4+ T cell responses, resulting in tissue damage and chronic intestinal inflammation[58,59]. Due to the heterogeneity of CD4+ T cells in patients, treatment efficacy can depend on the patient's subtype of CD4+ T cells[58,59]. Thus, the highly predictive cell type contexts according to PINNACLE should be further investigated to design safe and efficacious therapies for RA and IBD diseases.

Conversely, we hypothesize that the cell type contexts of protein representations that yield worse performance than the cell type-agnostic protein representations may not have the predictive power (given the current list of targets from drugs that have at least completed phase 2 of clinical trials) for studying the therapeutic effects of candidate targets for RA and IBD therapeutic areas.

In the context-aware model trained to nominate therapeutic targets for RA diseases, the protein representations of duodenum glandular cells, endothelial cells of hepatic sinusoid, myometrial cells and hepatocytes perform worse than the cell type-agnostic protein representations (Fig. 5a). The RA therapeutic area is a group of inflammatory diseases in which immune cells attack the synovial lining cells of joints[37]. Since duodenum glandular cells (PINNACLE-predicted rank 153), endothelial cells of hepatic sinusoid (PINNACLE-predicted rank 126), myometrial cells (PINNACLE-predicted rank 119) and hepatocytes (PINNACLE-predicted rank 116) are neither immune cells nor found in the synovium, these cell type contexts' protein representations expectedly perform poorly. For IBD diseases, the protein representations of the limbal stem cells, melanocytes, fibroblasts of cardiac tissue, and radial glial cells have worse performance than the cell type-agnostic protein representations (Fig. 5d). The IBD therapeutic area is a group

of inflammatory diseases in which immune cells attack tissues in the digestive tract[40]. As limbal stem cells (PINNACLE-predicted rank 152), melanocytes (PINNACLE-predicted rank 147), fibroblasts of cardiac tissue (PINNACLE-predicted rank 135) and radial glial cells (PINNACLE-predicted rank 107) are neither immune cells nor found in the digestive tract, these cell type contexts' protein representations should also perform worse than context-free representations.

The least predictive cellular contexts in PINNACLE's models for RA and IBD have no known role in disease, indicating that protein representations from these cell type contexts are poor predictors of RA and IBD therapeutic targets. PINNACLE's overall improved predictive ability compared to context-free models indicates the importance of understanding cell type contexts where therapeutic targets are expressed and act.

### Predictive cell type contexts reflect MoAs in RA therapies

Recognizing and leveraging the most predictive cell type context for examining a therapeutic area can be beneficial for predicting candidate therapeutic targets[45–49]. We find that considering only the most predictive cell type contexts can yield significant performance improvements compared to context-free models (Extended Data Fig. 10). We examine cell type contexts selected by PINNACLE as the most predictive for JAK3 and IL6R, two protein targets of RA drugs.

Disease-modifying antirheumatic drugs, such as Janus kinase (JAK) inhibitors (that is, tofacitinib, upadacitinib and baricitinib), are commonly prescribed to patients with RA[60,61]. For JAK3, PINNACLE's five most predictive cell type contexts are T follicular helper cells, microglial cells, DN3 thymocytes, CD4+ αβ memory T cells and hematopoietic stem cells (Fig. 5b). Since the expression of JAK3 is limited to hematopoietic cells, mutations or deletions in JAK3 tend to cause defects in T cells, B cells and natural killer cells[62–65]. For instance, patients with JAK3 mutations tend to be depleted of T cells[63], and the abundance of T follicular helper cells is highly correlated with RA severity and progression[66]. JAK3 is also highly expressed in double negative (DN) T cells (early stage of thymocyte differentiation)[67], and the levels of DN T cells are higher in synovial fluid than peripheral blood, suggesting a possible role of DN T cell subsets in RA pathogenesis[68]. Lastly, dysregulation of the JAK/STAT pathway, which JAK3 participates in, has pathological implications for neuroinflammatory diseases, a significant component of disease pathophysiology in RA[69,70].

Tocilizumab and sarilumab are approved by the Food and Drug Administration for treating RA, and target the interleukin six receptor, IL6R[61]. For IL6R, PINNACLE's five most predictive cellular contexts are classical monocytes, NAMPT neutrophils, intermediate monocytes, mesenchymal stem cells and regulatory T cells (Fig. 5c). IL6R is predominantly expressed on neutrophils, monocytes, hepatocytes, macrophages and some lymphocytes[71]. IL6R simulates the movement of T cells and other immune cells to the site of infection or inflammation[72] and affects T cell and B cell differentiation[71,73]. IL6 acts directly on neutrophils, essential mediators of inflammation and joint destruction in RA, through membrane-bound IL6R[71]. Experiments on fibroblasts isolated from the synovium of patients with RA show that anti-IL6 antibodies prevented neutrophil adhesion, indicating a promising therapeutic direction for IL6R on neutrophils[71]. Lastly, mice studies have shown that pretreatment of mesenchymal stem/stromal cells with soluble IL6R can enhance the therapeutic effects of mesenchymal stem/stromal cells in arthritis inflammation[74].

PINNACLE's hypotheses to examine JAK3 and IL6R in the highly predictive cell type contexts, according to PINNACLE, to maximize therapeutic efficacy seem to be consistent with their roles in the cell types. It seems that targeting these proteins may directly impact the pathways contributing to the pathophysiology of RA therapeutic areas. Further, our results for IL6R suggest that PINNACLE's contextualized representations could be leveraged to evaluate potential enhancement in efficacy (for example, targeting multiple points in a pathway of interest).

### Predictive cell type contexts elucidate MoAs in IBD therapies

Like RA, we must understand the cells in which therapeutic targets are expressed and act to maximize the efficacy of targeted IBD therapies[75]. To support our hypothesis, we evaluate PINNACLE's predictions for two protein targets of commonly prescribed treatments for IBD diseases: ITGA4 and PPARG.

Vedolizumab and natalizumab target the integrin subunit alpha 4, ITGA4, to treat the symptoms of IBD therapeutic area[61]. PINNACLE's five most predictive cell type contexts for ITGA4 are regulatory T cells, dendritic cells, myeloid dendritic cells, granulocytes and CD8+ αβ cytotoxic T cells (Fig. 5e). Integrins mediate the trafficking and retention of immune cells to the GI tract; immune activation of integrin genes increases the risk of IBD[76]. For instance, ITGA4 is involved in homing memory and effector T cells to inflamed tissues, including intestinal and nonintestinal tissues, and imbalances in regulatory and effector T cells may lead to inflammation[77]. Circulating dendritic cells express the gut homing marker encoded by ITGA4; the migration of blood dendritic cells to the intestine allows these dendritic cells to become mature, which leads to gut inflammation and tissue damage, indicating that future studies are warranted to elucidate the functional properties of blood dendritic cells in IBD[78].

Balsalazide and mesalamine are aminosalicylate drugs (disease-modifying antirheumatic drugs) commonly used to treat ulcerative colitis by targeting peroxisome proliferator-activated receptor gamma (PPARG)[61,79]. PINNACLE's five most predictive cell types for PPARG are paneth cells of the epithelium of large intestines, endothelial cells of the vascular tree, classic monocytes, goblet cells of small intestines and serous cells of epithelium of bronchus (Fig. 5f). PPARG is highly expressed in the GI tract, higher in the large intestine (for example, colonic epithelial cells) than the small intestine[80–82]. In patients with ulcerative colitis, PPARG is often substantially downregulated in their colonic epithelial cells[82]. PPARG promotes enterocyte development[83] and intestinal mucus integrity by increasing the abundance of goblet cells[82]. Further, PPARG activation can inhibit endothelial inflammation in vascular endothelial cells[84,85], which is significant due to the importance of vascular involvement in IBD[86]. Additionally, PPARG agonists have been shown to act as negative regulators of monocytes and macrophages, which can inhibit the production of proinflammatory cytokines[87]. Intestinal mononuclear phagocytes, such as monocytes, play a major role in maintaining epithelial barrier integrity and fine-tuning mucosal immune system responsiveness[88]. Studies show that newly recruited monocytes in inflamed intestinal mucosa drive the immunopathogenesis of IBD, suggesting that blocking monocyte recruitment to the intestine could be one avenue for therapeutic development[88]. Lastly, PPARG is found to regulate mucin and inflammatory factors in bronchial epithelial cells[89]. Given the pulmonary complications of IBD, PPARG could be a promising target to investigate for treating IBD and pulmonary symptoms[90]. The predictive power of cell type contexts to examine ITGA4 and PPARG, according to PINNACLE, for IBD therapeutic development is thus well supported.

## Discussion

PINNACLE is a flexible geometric deep learning approach for contextualized prediction in user-defined biological contexts. Integrating single-cell transcriptomic atlases with the protein interactome, cell type interactions, and tissue hierarchy, PINNACLE produces latent protein representations specialized to biological contexts. PINNACLE's protein representations capture cellular and tissue organization spanning 156 cell types and 62 tissues of varying hierarchical scales. In addition to multimodal data integration, a pretrained PINNACLE model generates protein representations that can be used for downstream prediction on tasks where cell type dependencies and cell type-specific mechanisms are relevant.

One limitation of the study is the use of the human protein interactome, which is not measured in a cell type-specific manner[91].

No systematic measurements of protein interactions across cell types exist. We create cell type-specific protein interaction networks by overlaying single-cell measurements on the protein interaction network, leveraging previously validated techniques for the reconstruction of cell type-specific interactomes at single-cell resolution[14] and conducting sensitivity network analyses to confirm the validity of the networks used to train PINNACLE models (Extended Data Figs. 2 and 3). This approach enriches networks for cell type-relevant proteins (Extended Data Fig. 2). The resulting networks may contain false-positive protein interactions (for example, proteins that interact in the reference protein interaction network but do not interact in a specific cell type) and false-negative protein interactions (for example, proteins that interact only within a particular cell type context that has not yet been measured). PINNACLE does not currently model proteins that may play a role in the cell type yet are unaffected by cell type specificity. Nevertheless, strong performance gains of PINNACLE over context-free models indicate the importance of contextualized prediction and suggest a direction to enhance existing analyses on protein interaction networks[4,6,7].

We can leverage and extend PINNACLE in many ways. PINNACLE can accommodate and supplement diverse data modalities. We developed PINNACLE models using Tabula Sapiens[20], a molecular reference atlas comprising almost 500,000 cells from 24 distinct tissues and organs. However, since the tissues and cell types associated with specific diseases may not be entirely represented in the atlas of healthy human subjects, we anticipate that our predictive power may be limited. Tabula Sapiens does not include synovial tissues associated with RA disease progression[25,39], but these can be found in synovial RA atlases[92] and stromal cells obtained from individuals with chronic inflammatory diseases[93]. To enhance the predictive ability of PINNACLE models, they can be trained on disease-specific or perturbation-specific networks. In this study, PINNACLE representations capture physical interactions between proteins at the cell type level (Supplementary Note 3); PINNACLE can also be applied to cell type-specific protein networks created from other modalities, such as cell type-specific gene expression networks[94]. We show that PINNACLE's representations can supplement protein representations generated from other data modalities, including protein 3D structure surfaces[3,17]. While this study focuses on protein-coding genes, information on protein isoforms and differential information, such as alternative splicing or allosteric changes, can be used with PINNACLE when such data are broadly available. In addition to prioritizing candidate therapeutic targets, PINNACLE's representations can be fine-tuned to identify populations of cells with specific characteristics, such as drug resistance[95], adverse drug events[96] or disease progression biomarkers[97]. Lastly, to move toward a 'lab-in-the-loop' framework, where computational and experimental scientists can iteratively refine the machine learning model and validate hypotheses via experiments, recent techniques on conformal prediction[98] and evidential layers can be integrated with PINNACLE to quantify the uncertainty of model outputs.

Protein representation learning models are context-free and are limited in analyzing protein phenotypes that are resolved by contexts and vary with cell types and tissues. To address this limitation, we introduce PINNACLE that produces protein representations tailored to cell type contexts. We demonstrate that contextual learning can provide a more comprehensive understanding of protein roles across cell type contexts[99]. As experimental technologies advance, it is becoming feasible to generate adaptive protein representations across cell type contexts and leverage contextualized representations to predict cell type-specific protein functions and nominate therapeutic candidates at the cell type level. Looking to the future, understanding protein functions and developing molecular therapies will require a comprehensive understanding of the roles that proteins have in different cell types and the interactions between proteins across diverse cell type contexts[100].

Approaches like PINNACLE can help realize this potential by generating contextualized protein representations, which can then be used to predict cell type-specific protein functions and identify therapeutic targets at the cellular level.

## Online content

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

## Methods

The Methods describe (1) the curation of datasets, (2) the construction and representation of multiscale single-cell networks, (3) PINNACLE multiscale graph neural network, (4) the fine-tuning of PINNACLE for target prioritization and (5) the metrics and statistical analyses used.

### Datasets

**Reference human physical protein interaction network.** Our reference PPI network is the union of physical multivalidated interactions from BioGRID[101,102], the Human Reference Interactome (HuRI)[91] and Menche et al.[103] with 15,461 nodes and 207,641 edges. Different sources of PPI have their own methods of curating and validating physical interactions between proteins. BioGRID, HuRI and Menche et al. are PPI networks from three well-cited publications and databases regarding human protein interactions. By joining the three networks, we construct a comprehensive global PPI network for our analysis.

**Multiorgan, single-cell transcriptomic atlas of humans.** We leverage Tabula Sapiens[20] data source as our multiorgan, single-cell transcriptomic atlas of humans. The data consists of 15 donors, with 59 specimens total. There are 483,152 cells after quality control, of which 264,824 are immune cells, 104,148 are epithelial cells, 31,691 are endothelial cells and 82,478 are stromal cells. The cells correspond to 177 unique cell ontology classes.

### Construction of multiscale networks

Our multiscale networks comprises protein–protein physical interactions, cell type-to-cell type communication, cell type-to-tissue relationships and tissue–tissue hierarchy.

**Cell type-specific protein interaction networks.** For each cell type, we create a cell type-specific network that represents the physical interactions between proteins (or genes) that are probably expressed in the cell type. Intuitively, our approach identifies genes significantly expressed in a given cell type with respect to the rest of the cells in the dataset. Concretely, we use the processed Tabula Sapiens count matrix to calculate the average expression of each gene in a cell type of interest and the average expression of the corresponding gene in all other cells. Then, we use the Wilcoxon rank-sum test on the two sets of average gene expression. From the resulting ranked list of genes based on activation, we filter for the top $K$ most activated genes. We repeat these two steps $N$ times and filter for genes that appear in at least 90% of iterations. Finally, we extract these genes' corresponding proteins from the global protein interaction network and take only the largest connected component. To ensure high-quality representations of cell types in our networks, we keep networks with at least 1,000 proteins. We do not perform subsampling of cells (that is, sample the same number of cells per cell type) to minimize information loss for constructing protein interaction networks (Extended Data Fig. 2). Limitations are described in Discussion.

**Cell type and tissue relationships in the metagraph.** We identify interactions between cell types based on LR expression using the CellPhoneDB[104] tool and database. An edge between a pair of cell types indicates that CellphoneDB predicts at least one significantly expressed LR pair (with a $P$ value less than 0.001) between them. As recommended by CellPhoneDB, cells are subsampled before running the algorithm, which uses geometric sketching[105] to efficiently sample a small representative subset of cells from massive datasets while preserving biological complexity. We choose to subsample 25% of cells and run CellPhoneDB for 100 iterations. We determine cell type–tissue relationships and extract tissue–tissue relationships using Tabula Sapiens meta-data. For relationships between cell types and tissues, we draw edges between cell types and the tissue that the cells were taken from. For tissue–tissue relationships, we select the nodes corresponding to

the tissues where samples were taken from and all parent nodes up to the root of the BRENDA tissue ontology[106]. We perform sensitivity and ablation analyses on different components of the metagraph (Supplementary Tables 3–5).

**Final dataset.** We have 156 cell type-specific protein interaction networks, which have, on average, 2,530 ± 677 proteins per network. The number of unique proteins across all cell type-specific protein interaction networks is 13,643 of the 15,461 proteins in the global reference protein interaction network. In the metagraph, we have 62 tissues (nodes), and 24 are directly connected to cell types. There are 3,567 cell–cell interactions, 372 cell–tissue edges and 79 tissue–tissue edges.

### Multiscale graph neural network

**Overview.** PINNACLE performs biologically informed message passing through proteins, cell types and tissues to learn cell type-specific protein representations, cell type representations and tissue representations in a unified multiscale embedding space. Specifically, PINNACLE traverses through protein–protein physical interactions in each cell type-specific PPI network, cell type–cell type communication, cell type–tissue relationships and tissue–tissue hierarchy with an attention mechanism over individual nodes and edge types. Its objective function is designed and optimized for learning the topology across biological scales, from proteins to cell types to tissues. The resulting embeddings from PINNACLE can be visualized and manipulated for hypothesis-driven interrogation and fine-tuned for diverse downstream biomedical prediction tasks.

**Problem formulation.** Let $\mathcal{G} = \{G_1, \ldots, G_{|\mathcal{C}|}\}$ be a set of cell type-specific PPI networks, where $\mathcal{C}$ is a set of unique cell types. Each $G_{c_i} = (V_{c_i}, E_{c_i})$ consists of a set of nodes $V_{c_i}$ and edges $E_{c_i}$ for a given cell type $c_i \in \mathcal{C}$ specific PPI network. Their nodes $u, v \in V_{c_i}$ are proteins, and edges $e_{u,v}^{PP} \in E_{c_i}$ are physical PPIs (denoted with PP in superscript). Cell types and tissues form a network, referred to as a metagraph. The metagraph's set of nodes comprises cell types $c_i \in \mathcal{C}$ and tissues $t_i \in \mathcal{T}$. The types of edges are cell type-cell type interactions (denoted with CC in superscript) $e_{c_i,c_j}^{CC}$ between any pair of cell types $c_i, c_j \in \mathcal{C}$; cell type-tissue associations (denoted with CT in superscript) $e_{c_i,t_i}^{CT}$ between any pair of cell type $c_i \in \mathcal{C}$ and tissue $t_i \in \mathcal{T}$; and tissue–tissue relationships (denoted with TT in superscript) $e_{t_i,t_j}^{TT}$ between any pair of tissues $t_i, t_j \in \mathcal{T}$.

**Protein-level attention with cell type specificity.** For each cell type-specific PPI network $\mathcal{G}_{c_i}$, we leverage protein-level attention to learn cell type-specific embeddings of proteins. Intuitively, protein-level attention learns which neighboring nodes are probably most important for characterizing a particular cell type's protein. As such, each cell type-specific protein interaction network has its own cell type-specific set of learnable parameters. Concretely, at each layer-wise update of layer $l$, the node-level attention learns the importance $\alpha_{u,v}$ of protein $u$ to its neighboring protein $v$ in a given cell type $c_i \in \mathcal{C}$:

$$\mathbf{h}_u^{PP} \leftarrow \text{AGG}\left(\sigma\left(\sum_{v \in \mathcal{N}_u} \alpha_{u,v} W^{PP} \mathbf{h}_v^{PP}\right)\right) \tag{1}$$

where AGG is an aggregation function (that is, concatenation across $K$ attention heads), $\sigma$ is the nonlinear activation function (that is, ReLU), $\mathcal{N}_u$ is the set of neighbors for $u$ (including itself via self-attention), $\alpha_{u,v}$ is an attention mechanism defined as $\alpha_{u,v} = \frac{\exp(\sigma(\mathbf{a}^T \cdot [\mathbf{h}_u \| \mathbf{h}_v]))}{\sum_{v \in \mathcal{N}_u} \exp(\sigma(\mathbf{a}^T \cdot [\mathbf{h}_u \| \mathbf{h}_v]))}$ between a pair of interacting proteins from a specific cell type, $W^{PP}$ is a PP-specific transformation matrix to project the features of protein $u$ in its cell type-specific protein interaction network, and $\mathbf{h}_v^{PP}$ is the previous layer's

cell type-specific embedding for protein $v$. Practically, we leverage the attention function in graph attention neural networks (that is, GATv2)[44]. Proteins of the same identity are initialized with the same random Gaussian vector to maintain their identity during training.

**Metagraph-level attention on cellular interactions and tissue hierarchy.** For the metagraph, we use node-level and edge-level attention to learn which neighboring nodes and edge types are probably most important for characterizing the target node (that is, the node of interest). Intuitively, to learn an embedding for a specific cell type or tissue, we evaluate the informativeness of each direct cell type or tissue neighbor, as well as the relationship type between the cell type or tissue and their neighbors (for example, parent–child tissue relationship, tissue from which a cell type is found, and cell type with which the cell type of interest communicates with).

Concretely, at each layer $l$ of PINNACLE, the embeddings of a cell type $c_i \in \mathcal{C}$ are the result of aggregating (via function AGG) the embeddings ($\mathbf{h}_c^{CC}$ and $\mathbf{h}_t^{CT}$) of its direct cell type neighbor $c$ and tissue neighbor $t$ that are projected via edge-type-specific transformation matrices ($W^{CC}$ and $W^{CT}$) and weighted by learned attention weights ($\alpha_{c_i,c}$ and $\alpha_{c_i,t}$ respectively):

$$\mathbf{h}_{c_i}^{CC} \leftarrow \text{AGG}\left(\sigma\left(\sum_{c \in \mathcal{N}_{c_i}} \alpha_{c_i,c} W^{CC} \mathbf{h}_c^{CC}\right)\right) \quad (2)$$

$$\mathbf{h}_{c_i}^{CT} \leftarrow \text{AGG}\left(\sigma\left(\sum_{t \in \mathcal{N}_{c_i}} \alpha_{c_i,t} W^{CT} \mathbf{h}_t^{CT}\right)\right) \quad (3)$$

The embeddings generated from separately propagating messages through cell type–cell type edges or cell type–tissue edges are combined using learned attention weights $\beta^{CC}$ and $\beta^{CT}$, respectively.

$$\mathbf{h}_{c_i} = \beta^{CC} \mathbf{h}_{c_i}^{CC} + \beta^{CT} \mathbf{h}_{c_i}^{CT} \quad (4)$$

Similarly, the embeddings of a tissue $t_i \in \mathcal{T}$ are the result of aggregating (via function AGG) the embeddings ($\mathbf{h}_t^{TT}$ and $\mathbf{h}_c^{TC}$) of its direct tissue neighbor $t$ and cell type neighbor $c$ that are projected via edge-type-specific transformation matrices ($W^{TT}$ and $W^{TC}$) and weighted by learned attention weights ($\alpha_{t_i,t}$ and $\alpha_{t_i,c}$ respectively).

$$\mathbf{h}_{t_i}^{TT} \leftarrow \text{AGG}\left(\sigma\left(\sum_{t \in \mathcal{N}_{t_i}} \alpha_{t_i,t} W^{TT} \mathbf{h}_t^{TT}\right)\right) \quad (5)$$

$$\mathbf{h}_{t_i}^{TC} \leftarrow \text{AGG}\left(\sigma\left(\sum_{c \in \mathcal{N}_{t_i}} \alpha_{t_i,c} W^{TC} \mathbf{h}_c^{TC}\right)\right) \quad (6)$$

The embeddings generated from separately propagating messages through tissue–tissue edges or tissue–cell type edges are combined using learned attention weights $\beta^{TT}$ and $\beta^{TC}$, respectively.

$$\mathbf{h}_{t_i} = \beta^{TT} \mathbf{h}_{t_i}^{TT} + \beta^{TC} \mathbf{h}_{t_i}^{TC} \quad (7)$$

For the node-level attention mechanisms (equations (2), (3), (5) and (6)), AGG is an aggregation function (that is, concatenation across $K$ attention heads), $\sigma$ is the nonlinear activation function (that is, ReLU), $\mathcal{N}_{c_i}$ and $\mathcal{N}_{t_i}$ are the sets of neighbors for $c_i$ and $t_i$ respectively (includes itself via self-attention), $W^{CC}$, $W^{CT}$, $W^{TC}$ and $W^{TT}$ are edge-type-specific transformation matrices to project the features of a given target node, $\mathbf{h}_c^{CC}$, $\mathbf{h}_t^{CT}$, $\mathbf{h}_t^{TT}$ and $\mathbf{h}_c^{TC}$ are the previous layer's embedding for $c$ given the edge type CC, $t$ given the edge type CT, $t$ given the edge type TT, and $c$ given the edge type TC, respectively. Practically, we leverage the attention function in graph attention neural networks (that is, GATv2)[44]. Finally, the node-level attention mechanism for a given source node $u$

and edge type $r$ is $\alpha_{u,v}^r = \frac{\exp(\sigma(\mathbf{a}_r^T \cdot [\mathbf{h}_u \| \mathbf{h}_v]))}{\sum_{v \in \mathcal{N}_u} \exp(\sigma(\mathbf{a}_r^T \cdot [\mathbf{h}_u \| \mathbf{h}_v]))}$. For the attention mechanisms over edge types (equations (4) and (7)), $\beta^r = \frac{\exp(m_r)}{\sum_{r \in R} \exp(m_r)}$ such that $m_r = \sum_{u \in V_q} \mathbf{s}^T \cdot \tanh(M \cdot \mathbf{h}_u^r + \mathbf{b})$ where $V_q$ is the set of nodes in the metagraph, $\mathbf{s}$ is the attention vector, $M$ is the weight matrix and $\mathbf{b}$ is the bias vector. These parameters are shared for all edge types in the metagraph.

**Bridge between protein and cell type embeddings.** Using a pooling mechanism, we bridge cell type-specific protein embeddings with their corresponding cell type embeddings. We initialize cell type embeddings by taking the average of their proteins' embeddings: $\mathbf{h}_{c_i} = \frac{1}{|V_{c_i}|} \sum_{u \in V_{c_i}} \mathbf{h}_u$, where $\mathbf{h}_u$ is the embedding of protein node $u \in V_{c_i}$ in the PPI subnetwork for cell type $c_i$. Similarly, we initialize tissue embeddings by taking the average of their neighbors: $\mathbf{h}_{t_i} = \frac{1}{|\mathcal{N}_{t_i}|} \left(\sum_{t \in \mathcal{N}_{t_i}} \mathbf{h}_t + \sum_{c \in \mathcal{N}_{t_i}} \mathbf{h}_c\right)$, where $\mathbf{h}_t$ and $\mathbf{h}_c$ are the embeddings of tissue node $t$ and cell type node $c$, respectively, in the immediate neighborhood of source tissue node $t_i$. At each layer $l > 0$, we learn the importance $\gamma_{c_i,u}$ of node $u \in V_{c_i}$ to cell type $c_i$ such that

$$\mathbf{h}_{c_i} \leftarrow \mathbf{h}_{c_i} + \text{AGG}\left(\sigma\left(\sum_{u \in V_{c_i}} \gamma_{c_i,u} \mathbf{h}_u\right)\right). \quad (8)$$

After propagating cell type and tissue information in the metagraph (namely equations (2)–(6)), we apply $\gamma_{c_i,u}$ to the cell type embedding of $c_i$ such that

$$\mathbf{h}_u \leftarrow \mathbf{h}_u + \gamma_{c_i,u} \mathbf{h}_{c_i}. \quad (9)$$

Intuitively, we are imposing the structure of the metagraph onto the PPI subnetworks based on a protein's importance to its corresponding cell type's identity.

**Overall objective function of PINNACLE.** PINNACLE is optimized for three biological scales: protein, cell type and tissue level. Concretely, the loss function $\mathcal{L}$ has three components corresponding to each biological scale:

$$\mathcal{L} = \mathcal{L}_{\text{protein}} + (1 - \theta)(\mathcal{L}_{\text{celltype}} + \mathcal{L}_{\text{tissue}}), \quad (10)$$

where $\mathcal{L}_{\text{protein}}$, $\mathcal{L}_{\text{celltype}}$ and $\mathcal{L}_{\text{tissue}}$ minimize the loss from protein-level predictions, cell type-level predictions and tissue-level predictions, respectively. $\theta$ is a tunable parameter with a range of 0 and 1 that scales the contribution of the link prediction loss of the metagraph relative to that of the PPIs. At the protein level, we consider two aspects: prediction of PPIs at each cell type-specific PPI network ($\mathcal{L}_{\text{ppi}}$) and prediction of cell type identity of each protein ($\mathcal{L}_{\text{celltypeid}}$). The contribution of the latter is scaled by $\lambda$, which is a tunable parameter with a range of 0 and 1.

$$\mathcal{L}_{\text{protein}} = \theta \mathcal{L}_{\text{ppi}} + \lambda \mathcal{L}_{\text{celltypeid}} \quad (11)$$

Intuitively, we aim to simultaneously learn the topology of each cell type-specific PPI network (that is, $\mathcal{L}_{\text{ppi}}$) and the nuanced differences between proteins activated in different cell types. Specifically, we use binary cross-entropy to minimize the error of predicting positive and negative PPIs in each cell type-specific PPI network

$$\mathcal{L}_{\text{ppi}} = \sum_{c_i \in \mathcal{C}} \sum_{u,v \in V_{c_i}} y_{u,v} \log(\hat{y}_{u,v}) + (1 - y_{u,v}) \log(1 - \hat{y}_{u,v}) \quad (12)$$

and center loss[107] for discriminating between protein embeddings $\mathbf{z}_u$ from different cell types, represented by embeddings denoted as $\mathbf{z}_{c_i}$.

$$\mathcal{L}_{\text{celltypeid}} = \sum_{c_i \in \mathcal{C}} \sum_{u \in V_{c_i}} ||\mathbf{z}_u - \mathbf{z}_{c_i}||_2^2 \tag{13}$$

At the cell type level, we use binary cross-entropy to minimize the error of predicting cell type–cell type interactions and cell type–tissue relationships:

$$\mathcal{L}_{\text{celltype}} = \mathcal{L}_{\text{celltype}}^{\text{CC}} + \mathcal{L}_{\text{celltype}}^{\text{CT}} \tag{14}$$

such that

$$\mathcal{L}_{\text{celltype}}^{\text{CC}} = \sum_{c_i, c_j \in \mathcal{C}} y_{c_i, c_j} \log(\hat{y}_{c_i, c_j}) + (1 - y_{c_i, c_j}) \log(1 - \hat{y}_{c_i, c_j}) \tag{15}$$

$$\mathcal{L}_{\text{celltype}}^{\text{CT}} = \sum_{c_i \in \mathcal{C}} \sum_{t_k \in \mathcal{T}} y_{c_i, t_k} \log(\hat{y}_{c_i, t_k}) + (1 - y_{c_i, t_k}) \log(1 - \hat{y}_{c_i, t_k}). \tag{16}$$

Similarly, at the tissue level, we use binary cross-entropy to minimize the error of predicting tissue–tissue and tissue–cell type relationships:

$$\mathcal{L}_{\text{tissue}} = \mathcal{L}_{\text{tissue}}^{\text{TT}} + \mathcal{L}_{\text{tissue}}^{\text{TC}} \tag{17}$$

such that

$$\mathcal{L}_{\text{tissue}}^{\text{TT}} = \sum_{t_k, t_q \in \mathcal{T}} y_{t_k, t_q} \log(\hat{y}_{t_k, t_q}) + (1 - y_{t_k, t_q}) \log(1 - \hat{y}_{t_k, t_q}) \tag{18}$$

$$\mathcal{L}_{\text{tissue}}^{\text{TC}} = \sum_{t_k \in \mathcal{T}} \sum_{c_i \in \mathcal{C}} y_{t_k, c_i} \log(\hat{y}_{t_k, c_i}) + (1 - y_{t_k, c_i}) \log(1 - \hat{y}_{t_k, c_i}). \tag{19}$$

The probability of an edge of type $i$ between nodes $u$ and $v$ is calculated using a bilinear decoder:

$$y_{u,v} = \mathbf{z}_u \cdot \mathbf{r}_i \cdot \mathbf{z}_v, \tag{20}$$

where $\mathbf{z}_u$ and $\mathbf{z}_v$ are embeddings of nodes $u$ and $v$, and $\mathbf{r}_i$ is the embedding for edge type $i$. Note that any decoder can be used for link prediction in PINNACLE.

**Training details for PINNACLE.** *Overview.* PINNACLE is trained using the cell type identity of the protein interaction networks and the graph connectivity of the cell type-specific protein interaction networks and metagraph. To learn cell type identity, PINNACLE predicts the cell type(s) that the node(s) corresponding to each protein are activated in. For capturing graph connectivity, PINNACLE performs self-supervised link prediction; it predicts whether an edge (and its type) exists between a pair of nodes. For link prediction, a randomly selected subset of edges is masked (or hidden) from the model, and the model must be able to predict that such edges exist (and that the randomly generated false edges do not exist). Practically, this means that the graphs being fed as input into PINNACLE during train, validation, or test do not contain the masked edges.

*Data split.* Protein–protein edges are randomly split into train (80%), validation (10%) and test (10%) sets. The metagraph edges are not split into train, validation and test sets because there are relatively few of them, and they are all critical for injecting cell type and tissue organization to the model. The proteins involved in the train edges are considered in the cell type identification term of the loss function ($\mathcal{L}_{\text{celltypeid}}$).

*Sampling negative edges.* For link prediction, false (or negative) edges have the label of 0 and are randomly generated (via structured_negative_sampling function in Pytorch Geometric). The ratio of positive to negative edges is 1:1.

*Hyperparameter tuning.* We leverage Weights and Biases[108] to select optimal hyperparameters via a random search over the hyperparameter space. The best-performing hyperparameters for PINNACLE are selected by optimizing the ROC and Calinski–Harabasz score[109] on the validation set. The hyperparameter space on which we perform a random search to choose the optimal set of hyperparameters is: the dimension of the nodes' feature matrix $\in$ [1,024, 2,048], dimension of the output layer $\in$ [4, 8, 16, 32], lambda $\in$ [0.1, 0.01, 0.001], learning rate for link prediction task $\in$ [0.01, 0.001], learning rate for protein's cell type classification task $\in$ [0.1, 0.01, 0.001], number of attention heads $\in$ [4, 8], weight decay rate $\in$ [0.0001, 0.00001], dropout rate $\in$ [0.3, 0.4, 0.5, 0.6, 0.7] and normalization layer $\in$ [layernorm, batchnorm, graphnorm, none]. The best hyperparameters are as follows: the dimension of the nodes' feature matrix = 1,024, dimension of the output layer = 16, lambda = 0.1, learning rate for link prediction task = 0.01, learning rate for protein's cell type classification task = 0.1, number of attention heads = 8, weight decay rate = 0.00001, dropout rate = 0.6, and normalization layers are layernorm and batchnorm. Further, PINNACLE consists of two custom graph attention neural network layers ('Protein-level attention with cell type specificity' and 'Metagraph-level attention on cellular interactions and tissue hierarchy' sections in Methods) per cell type-specific PPI network and metagraph and is trained for 250 epochs.

*Implementation.* We implement PINNACLE using Pytorch (Version 1.12.1)[110] and Pytorch Geometric (Version 2.1.0)[111]. We leverage Weights and Biases[108] for hyperparameter tuning and model training visualization, and we create interactive demos of the model using Gradio[112]. Models are trained on a single NVIDIA Tesla V100-SXM2-16GB GPU.

## Generating contextualized 3D protein representations
After pretraining PINNACLE, we can leverage the output protein representations for diverse downstream tasks. Here, we demonstrate PINNACLE's ability to improve the prediction of PPIs by injecting context into 3D molecular structures of proteins.

**Overview.** Given a protein of interest, we generate both the context-free structure-based representation via MaSIF[3,17] and a contextualized PPI network-based representation via PINNACLE. We calculate the binding score of a pair of proteins based on either context-free representations or contextualized representations of the proteins. To quantify the added value, if any, provided by contextualizing protein representations with cell type context, we compare the size of the gap between the average binding scores of binding and nonbinding proteins in the two approaches.

**Dataset.** The proteins being compared are PD-1, PD-L1, B7-1, CTLA-4, RalB, RalBP1, EPO, EPOR, C3 and CFH. The pairs of binding proteins are PD-1/PD-L1 (PDB ID: 4ZQK) and B7-1/CTLA-4 (PDB ID: 1I8L). The nonbinding proteins are any of the four proteins paired with any of the remaining six proteins (for example, PD-1/RalB, PD-1/RalBP1 and PD-L1/RalBP1). The PDB IDs for the other six proteins are 2KWI for RalB/RalBP1, 1CN4 for EPO/EPOR, and 3OXU for C3/CFH.

**Structure-based protein representation learning.** We directly apply the pretrained model for MaSIF[3,17] to generate the 3D structure-based protein representations. We use the model pretrained for MaSIF-site task, named all_feat_3l_seed_benchmark. The output of the pretrained model for a given protein is $P \times d$, where $P$ is the number of patches (precomputed by the authors of MaSIF[3,17]) and $d = 4$ is the dimension of the pretrained model's output layer. As proteins vary in size (that is, the number of patches to cover the surface of the protein), we select a fixed $k$ number of patches that are most likely to be part of the binding site (according to the pretrained MaSIF model). For each protein, we select $k = 200$ patches, which is the average number of patches for PD-1,

PD-L1, B7-1 and CTLA-4, resulting in a matrix of size 200 × 4. Finally, we take the element-wise median on the 200 × 4 matrix to transform it into a vector of length 200. This vector becomes the structure-based protein representation for a given protein.

**Experimental setup.** For each cell type context of a given protein, we concatenate the 3D structure-based protein representation (from MaSIF) with the cell type-specific protein representation (from PINNACLE) to generate a contextualized structure-based protein representation. To create the context-free protein representation, we concatenate the structure-based protein representation with an element-wise average of PINNACLE's protein representations. This is to maintain consistent dimensionality and latent space between context-free and contextualized protein representations. Given a pair of proteins, we calculate a score via cosine similarity (a function provided by sklearn[113]) using the context-free or contextualized protein representations. Lastly, we quantify the gap between the scores of binding and nonbinding proteins using context-free or contextualized protein representations to evaluate the added value (if any) of contextual AI.

### Fine-tuning PINNACLE for context-specific target prioritization

After pretraining PINNACLE, we can fine-tune the output protein representations for diverse biomedical downstream tasks. Here, we demonstrate PINNACLE's ability to enhance the performance of predicting a protein's therapeutic potential for a specific therapeutic area.

For each protein of interest, we feed its PINNACLE-generated embedding into an MLP. The model outputs a score between 0 and 1, where 1 indicates strong candidacy to target (that is, by a compound/drug) for treating the therapeutic area and 0 otherwise. Since a protein has multiple representations corresponding to the cell types it is activated in, the MLP model generates a score for each of the protein's cell type-specific representations (Fig. 4a). For example, Protein 1's representation from Cell type 1 is scored independently of its representation from Cell type 2. The output scores can be examined to identify the most predictive cell types and the strongest candidates for therapeutic targets in any specific cell type.

**Therapeutic targets dataset.** We obtain labels for therapeutic targets from the Open Targets Platform[61].

*Therapeutic area selection.* To curate target information for a therapeutic area, we examine the drugs indicated for the therapeutic area of interest and its descendants. The two therapeutic areas examined are RA and IBD. For RA, we collected therapeutic data (that is, targets of drugs indicated for the therapeutic area) from OpenTargets[61] for RA (EFO_0000685), ankylosing spondylitis (EFO_0003898) and psoriatic arthritis (EFO_0003778). For IBD, we collected therapeutic data for ulcerative colitis (EFO_0000729), collagenous colitis (EFO_1001293), colitis (EFO_0003872), proctitis (EFO_0005628), Crohn's colitis (EFO_0005622), lymphocytic colitis (EFO_1001294), Crohn's disease (EFO_0000384), microscopic colitis (EFO_1001295), IBD (EFO_0003767), appendicitis (EFO_0007149), ulcerative proctosigmoiditis (EFO_1001223) and small bowel Crohn's disease (EFO_0005629).

*Positive training examples.* We define positive examples (that is, where the label $y = 1$) as proteins targeted by drugs that have at least completed phase 2 of clinical trials for treating a certain therapeutic area. As such, a protein is a promising candidate if a compound that targets the protein is safe for humans and effective for treating the disease. We retain positive training examples that are activated in at least one cell type-specific protein interaction network. The final number of positive training examples for RA and IBD is 152 and 114, respectively.

*Negative training examples.* We define negative examples (that is, where the label $y = 0$) as druggable proteins that do not have any known association with the therapeutic area of interest according to OpenTargets. A protein is deemed druggable if it is targeted by at least one existing drug[114]. We extract drugs and their nominal targets from DrugBank[79]. We retain negative training examples that are activated in at least one cell type-specific protein interaction network. The final number of negative training examples for RA and IBD is 1,465 and 1,377, respectively.

*Data processing workflow.* For a therapeutic area of interest, we identify its descendants. With the list of disease terms for the therapeutic area, we curate its positive and negative training examples. We split the dataset such that about 60%, 20% and 20% of the proteins are in the train, validation and test sets, respectively. We additionally apply two criteria to avoid data leakage and ensure that all cell types are represented during training/inference: Proteins are assigned to train (60%), validation (20%) and test (20%) datasets based on their identity; this is to prevent data leakage where cell type-specific representations of a single protein are observed in multiple data splits. We also ensure that there are sufficient numbers of train, validation and test positive samples per cell type; proteins may be reassigned to a different data split so that each cell type is represented during training, validating and testing stages. With these criteria, the train, validation and test dataset splits may not necessarily consist of approximately 60%, 20% and 20% of the total protein representations (Supplementary Table 6).

**Fine-tuning model details.** *Model architecture.* Our MLP comprises an input feedforward neural network, one hidden feedforward neural network layer and an output feedforward neural network layer. In between each layer, we have a nonlinear activation layer. In addition, we use dropout and normalization layers between the input and hidden layer (see 'Implementation' section for more information). Our objective function is binary cross-entropy loss.

*Hyperparameter tuning.* We leverage Weights and Biases[108] to select optimal hyperparameters via a random search over the hyperparameter space. The best-performing hyperparameters are selected by optimizing the AUPRC on the validation set. The hyperparameter space on which we perform a random search to choose the optimal set of hyperparameters is the dimension of the first hidden layer ∈ [8, 16, 32], dimension of the second hidden layer ∈ [8, 16, 32], learning rate ∈ [0.01, 0.001, 0.0001], weight decay rate ∈ [0.001, 0.0001, 0.00001, 0.000001], dropout rate ∈ [0.2, 0.3, 0.4, 0.5, 0.6, 0.7, 0.8], normalization layer ∈ [layernorm, batchnorm, none] and the ordering of dropout and normalization layer (that is, normalization before dropout or vice versa).

*Implementation.* We implement the MLP using Pytorch (Version 1.12.1)[110]. In addition, we use Weights and Biases[108] for hyperparameter tuning and model training visualization. Models are trained on a single NVIDIA Tesla M40 GPU.

### Metrics and statistical analyses

Here, we describe metrics, visualization methods and statistical tests used in our analysis.

**Visualization of embeddings.** We visualize PINNACLE's embeddings using a uniform manifold approximation and projection for dimension reduction (UMAP)[115] and seaborn. Using the Python package, umap, we transform PINNACLE's embeddings to two-dimensional vectors via the parameters: n_neighbors = 10, min_dist = 0.9, n_components = 2 and the euclidean distance metric. The plots are created using the seaborn package's scatterplot function.

**Visualization of cell type embedding similarity.** The pairwise similarity of PINNACLE's cell type embeddings is calculated using cosine similarity (a function provided by sklearn[113]). Then, these similarity scores are visualized using the seaborn package's clustermap function. For visualization purposes, similarity scores are mapped to colors after being raised to the 20th power.

**Spatial enrichment analysis of PINNACLE's protein embeddings.** To quantify the spatial enrichment for PINNACLE's protein embedding regions, we apply a systematic approach, SAFE[31], that identifies regions that are overrepresented for a feature of interest (Extended Data Figs. 3 and 4). The required input data for SAFE are networks and annotations of each node. We first construct an unweighted similarity network on PINNACLE protein embeddings: (1) calculate pairwise cosine similarity, (2) apply a similarity threshold on the similarity matrix to generate an adjacency matrix and (3) extract the largest connected component. The protein nodes are labeled as 1 if they belong to a given cell type context and 0 otherwise. We then apply SAFE to each network using the recommended settings: neighborhoods are defined using the short-path_weighted_layout metric for node distance and neighborhood radius of 0.15, and *P* values are computed using the hypergeometric test, adjusted using the Benjamini–Hochberg false discovery rate correction (significance cutoff $\alpha = 0.05$).

Due to computation and memory constraints, we sample 50 protein embeddings from a cell type context of interest and 10 protein embeddings from each of the other 155 cell type contexts. We use a threshold of 0.3 in our evaluation of PINNACLE's protein embedding regions (Fig. 2 and Extended Data Fig. 3). We also evaluate the spatial enrichment analysis on networks constructed from different thresholds to ensure that the enrichment is not sensitive to our network construction method: [0.1, 0.2, 0.3, 0.4, 0.5, 0.6, 0.7, 0.8, 0.9] (Extended Data Fig. 4). We use the Python implementation of SAFE (https://github.com/baryshnikova-lab/safepy).

**Statistical significance of tissue embedding distance.** Tissue embedding distance between a given pair of tissue nodes is calculated using cosine distance (a function provided by sklearn[113]). Tissue ontology distance between a given pair of tissue nodes is calculated by taking the sum of the nodes' shortest path lengths to the lowest common ancestor (functions provided by networkx[116]. We use the two-sample Kolmogorov–Smirnov test (a function provided by scipy[117]) to compare PINNACLE embedding distances against randomly generated vectors (via the randn function in numpy to sample an equal number of vectors from a standard normal distribution). We also use the Spearman correlation (a function provided by scipy[117]) to correlate PINNACLE embedding distance to tissue ontology distance. We additionally generate a null distribution of tissue ontology distance by calculating tissue ontology distance on a shuffled tissue hierarchy (repeated ten times). Concretely, we shuffle the node identities of the Brenda Tissue Ontology[106] and compute the pairwise tissue ontology distances.

**Statistical significance of binding and nonbinding proteins' score gaps.** We perform a one-sided nonparametric permutation test. First, we concatenate the scores for the *N* binding pairs and *M* nonbinding pairs. Next, for 100,000 iterations, we randomly sample *N* scores as the new set of binding protein scores and *M* scores as the new set of nonbinding protein scores, calculate the mean $\mu_N$ of the *N* binding protein scores and the mean $\mu_M$ of the *M* nonbinding protein scores, calculate the score gap by taking the difference of the means as $\mu_N - \mu_M$, and keep track of the score gaps that are greater than or equal to the true score gap calculated from the real data. Lastly, we calculate the *P* value, defined as the fraction of 100,000 iterations in which the permuted score gap is greater than or equal to the true score gap (that is, one-sided nonparametric permutation test).

**Performance metric for therapeutic target prioritization.** For our downstream therapeutic target prioritization task ('Fine-tuning PINNACLE for context-specific target prioritization' section in Methods), we use a metric called Average Precision and Recall at K (APR@K) to evaluate model performance. APR@K leverages a combination of Precision@K and Recall@K to measure the ability to rank the most relevant items (in our case, proteins) among the top *K* predictions. In essence, APR@K calculates Precision@K for each $k \in [1, ..., K]$, multiplying each Precision@*k* by whether the *k*th item is relevant, and divides by the total number of relevant items *r* at *K*:

$$\text{APR@}K = \frac{1}{r} \sum_{k=1}^{K} \text{Precision@}k \times \text{rel}(k),$$

where

$$\text{rel}(k) = \begin{cases} 1, & \text{if item at } k \text{ is relevant} \\ 0, & \text{otherwise} \end{cases}.$$

Given the nature of our target prioritization task, some key advantages of using APR@K include robustness to (1) varied numbers of protein targets activated across cell type-specific protein interaction networks and (2) varied sizes of cell type-specific protein interaction networks.

### Reporting summary

Further information on research design is available in the Nature Portfolio Reporting Summary linked to this article.

## Data availability

All data used in the paper, including the cell type-specific protein interaction networks, the metagraph of cell type and tissue relationships, PINNACLE's contextualized representations, the therapeutic targets of RA and IBD diseases, and the final and intermediate results of the analyses, are shared via the project website at https://zitniklab.hms.harvard.edu/projects/PINNACLE. Datasets are available via figshare at https://doi.org/10.6084/m9.figshare.22708126 (ref. 118).

## Code availability

Python implementation of the methodology developed and used in the study is available via the project website at https://zitniklab.hms.harvard.edu/projects/PINNACLE. The code to reproduce results, together with documentation and examples of usage, is available on GitHub at https://github.com/mims-harvard/PINNACLE. We provide an interactive demo via HuggingFace to explore PINNACLE's contextualized protein representations.

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

## Acknowledgements

We thank A. Xenos for his valuable feedback on analyses of cell type-specific and tissue-agnostic protein functions. M.M.L. is supported by T32HG002295 from the National Human Genome Research Institute and a National Science Foundation Graduate Research Fellowship. M.M.L. and M.Z. gratefully acknowledge the support of NIH R01HD108794, NSF CAREER 2339524, US DoD FA8702-15-D-0001, awards from Harvard Data Science Initiative, Amazon Faculty Research, Google Research Scholar Program, AstraZeneca Research, Roche Alliance with Distinguished Scientists, Sanofi iDEA-iTECH Award, Pfizer Research, Chan Zuckerberg Initiative, John and Virginia Kaneb Fellowship award at Harvard Medical School, Aligning Science Across Parkinson's (ASAP) Initiative, Biswas Computational Biology Initiative in partnership with the Milken Institute, Harvard Medical School Dean's Innovation Awards for the Use of Artificial Intelligence, and Kempner Institute for the Study of Natural and Artificial Intelligence at Harvard University. A.N.A. gratefully acknowledges the support of NIH R01DK127171. K.L. gratefully acknowledges the support of NIH P30 AR072577. The content is solely the responsibility of the authors. The funders had no role in study design, data collection and analysis, decision to publish or preparation of the manuscript.

## Author contributions

M.M.L. retrieved and processed Tabula Sapiens, the global reference protein interaction network, CellPhoneDB repository of LR interactions, and the tissue hierarchy to construct the cell type-specific protein interaction networks and metagraph of cell type and tissue relationships. M.M.L. and M.S. performed the network analysis. M.M.L. and Y.H. retrieved and processed the OpenTargets data. M.M.L. developed, implemented and benchmarked PINNACLE, Y.H. improved the scalability of PINNACLE, and M.M.L. and Y.H. performed detailed analyses of PINNACLE's algorithm. A.V. and D.M. advised the network construction and the analysis of PINNACLE's outputs. M.Q.L., K.L. and A.N.A. provided clinical expertise on using PINNACLE for predicting therapeutic targets in a cell type-specific manner and interpreting the resulting cell type contexts for RA and IBD. M.M.L. and M.Z. designed the study. All authors contributed to writing the manuscript.

## Competing interests

D.M. and A.V. are currently employed by F. Hoffmann-La Roche Ltd. The other authors declare no competing interests.

## Additional information

**Extended data** is available for this paper at https://doi.org/10.1038/s41592-024-02341-3.

**Correspondence and requests for materials** should be addressed to Marinka Zitnik.

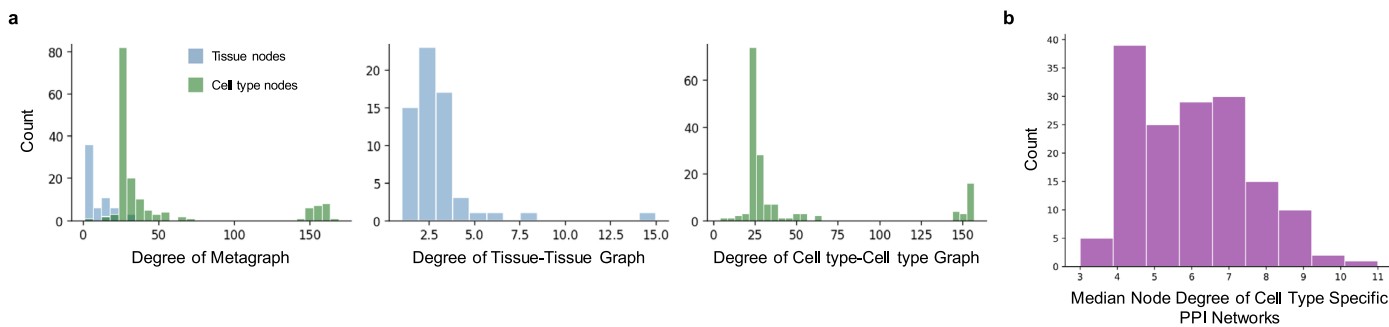

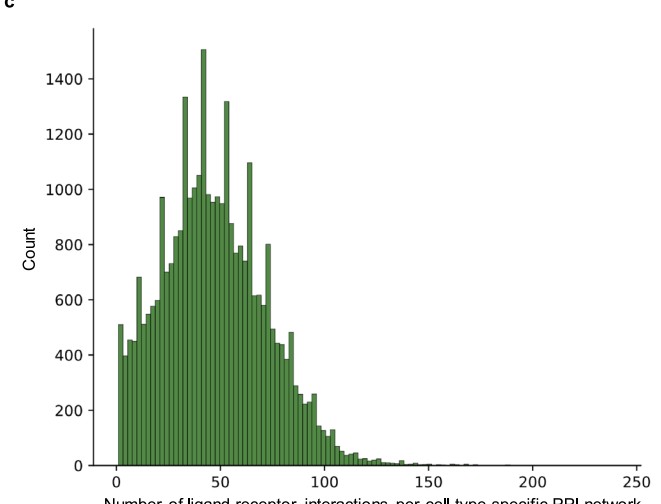

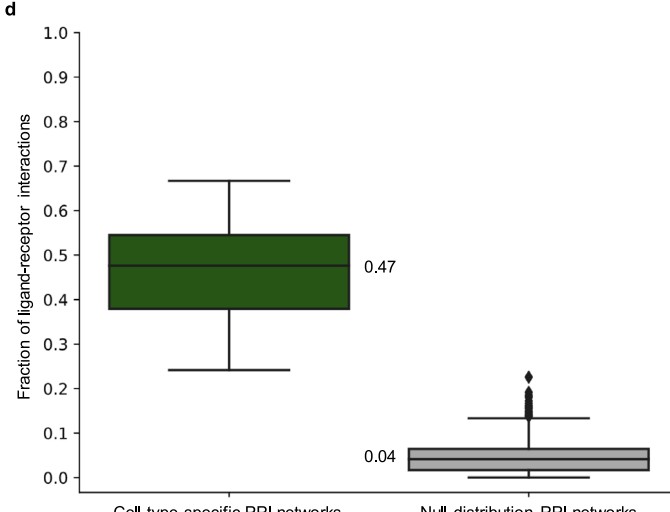

**Extended Data Fig. 1 | Network properties of the metagraph and cell type specific protein interaction networks. (a-b)** Degree distributions of the metagraph and cell type specific protein interaction (PPI) networks. **(a)** Degree distributions of the metagraph (composed of cell type-cell type, cell type-tissue, and tissue-tissue edges), tissue-tissue graph, and cell type-cell type graph. The median, maximum, and minimum degrees for the metagraph are 24, 169, 1; for the tissue-tissue graph are 2, 15, 1; and for the cell type-cell type graph are 24, 157, 4. **(b)** Distribution of the median node degree of each cell type specific PPI network. The median, maximum, and minimum of median node degree across cell type specific PPI networks are 6, 11, and 3, respectively. **(c-d)** Enrichment analysis of ligand-receptor interactions in the cell type specific PPI networks. We utilize CellPhoneDB[103] to predict interactions between cell types in our metagraph by identifying significantly expressed ligand-receptor (LR) interactions between pairs of cell types in our dataset. **(c)** Shown is a histogram of the number of significant LR interactions per cell type specific PPI network predicted by CellPhoneDB. **(d)** We hypothesize that the predicted LR interactions are enriched in our cell type specific PPI networks. To quantify the enrichment of LR interactions, we calculate the fraction of LR interactions where the corresponding ligand and receptor proteins are activated in the cell

type pair (that is, for a LR interaction identified between cell types A and B, the ligand protein is activated in cell type A's PPI network and the receptor protein is activated in cell type B's PPI network). We compare the fraction of LR pairs that are activated in our cell type specific PPI networks against the fraction of LR pairs that are activated in null distribution PPI networks. For each cell type specific PPI network, we generate 100 null distribution PPI networks by sampling the same number of nodes with a similar degree distribution. Degree distribution is preserved by binning nodes such that there are at least 100 nodes in each bin, and nodes are then randomly sampled within the appropriate degree interval. We find that our cell type specific PPI networks have a significantly higher fraction of ligand-receptor pairs activated (0.47 +/- 0.12) than the null distribution PPI networks (0.04 +/- 0.04); n = 2,020 pairs of cell type specific PPI networks, of which 20 are pairs of real cell type specific PPI networks and 2,000 are pairs of null cell type specific PPI networks. Note that the ligand-receptor interactions considered in both analyses are those where the genes corresponding to the ligands and receptors are known. However, this does not factor into our construction of the edges/interactions between cell types (CCI). The bounds of the box show the quartiles of the data, the center indicates the median value of the data, and the whiskers represent the farthest data point within 1.5 x IQR.

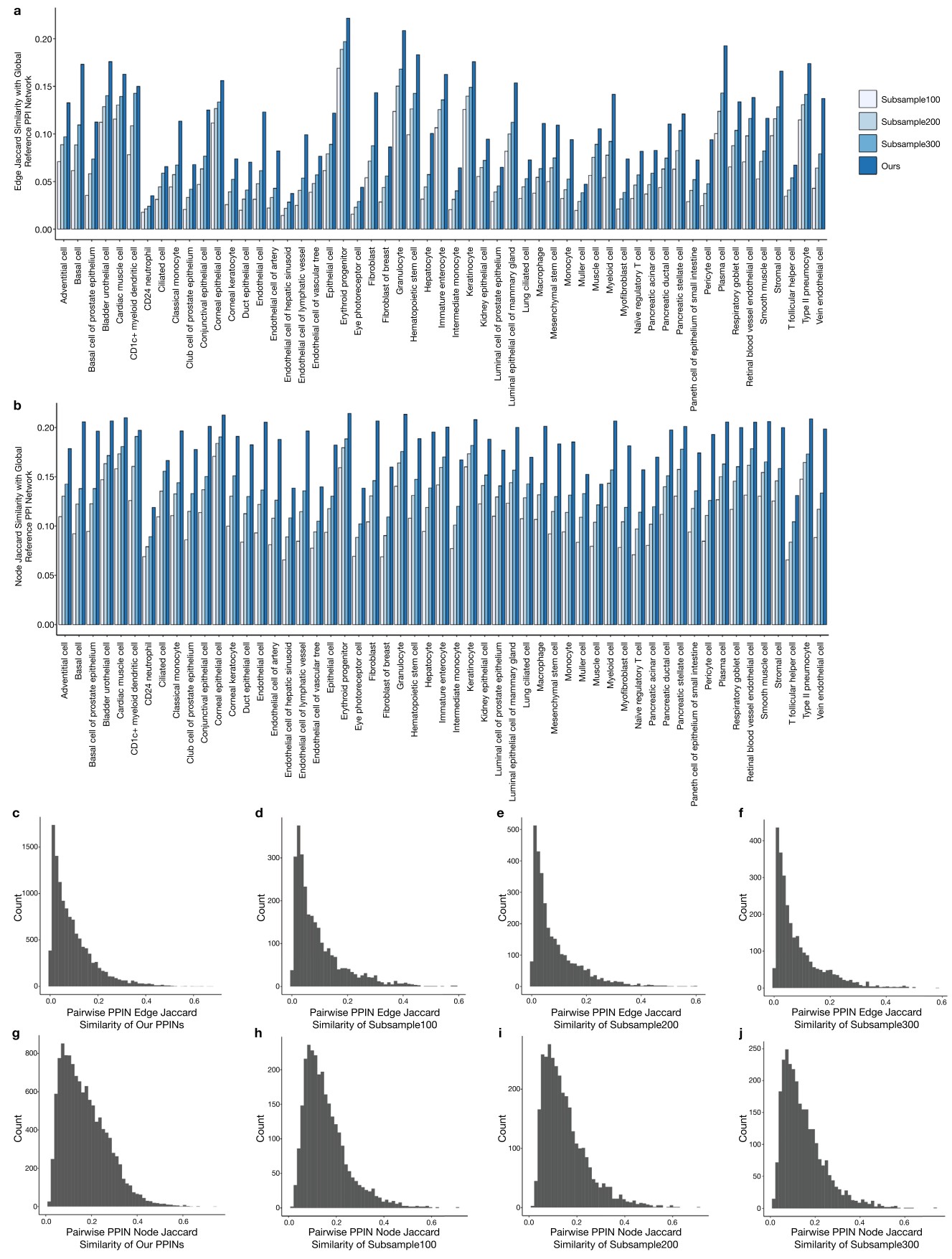

**Extended Data Fig. 2 | See next page for caption.**

**Extended Data Fig. 2 | Sensitivity analysis of network construction.** To examine whether cell types with fewer cells are poorly represented in our networks, we construct networks after subsampling equal numbers of cells per cell type. We compare our finalized networks (no subsampling of cells) against approaches that subsample 100, 200, and 300 cells. We find that our approach yields networks that are maximally similar to the global reference network yet maintain specificity to cell type context. **(a)** Edge and **(b)** node Jaccard similarity of a cell type specific PPIN to the global reference PPIN. **(c-j)** Distribution of edge Jaccard similarity between PPINs constructed by **(c)** our finalized approach and subsampling **(d)** 100, **(e)** 200, and **(f)** 300 cells. **(g-j)** Distribution of node Jaccard similarity between PPINs constructed by **(g)** our finalized approach and subsampling **(h)** 100, **(i)** 200, and **(j)** 300 cells.

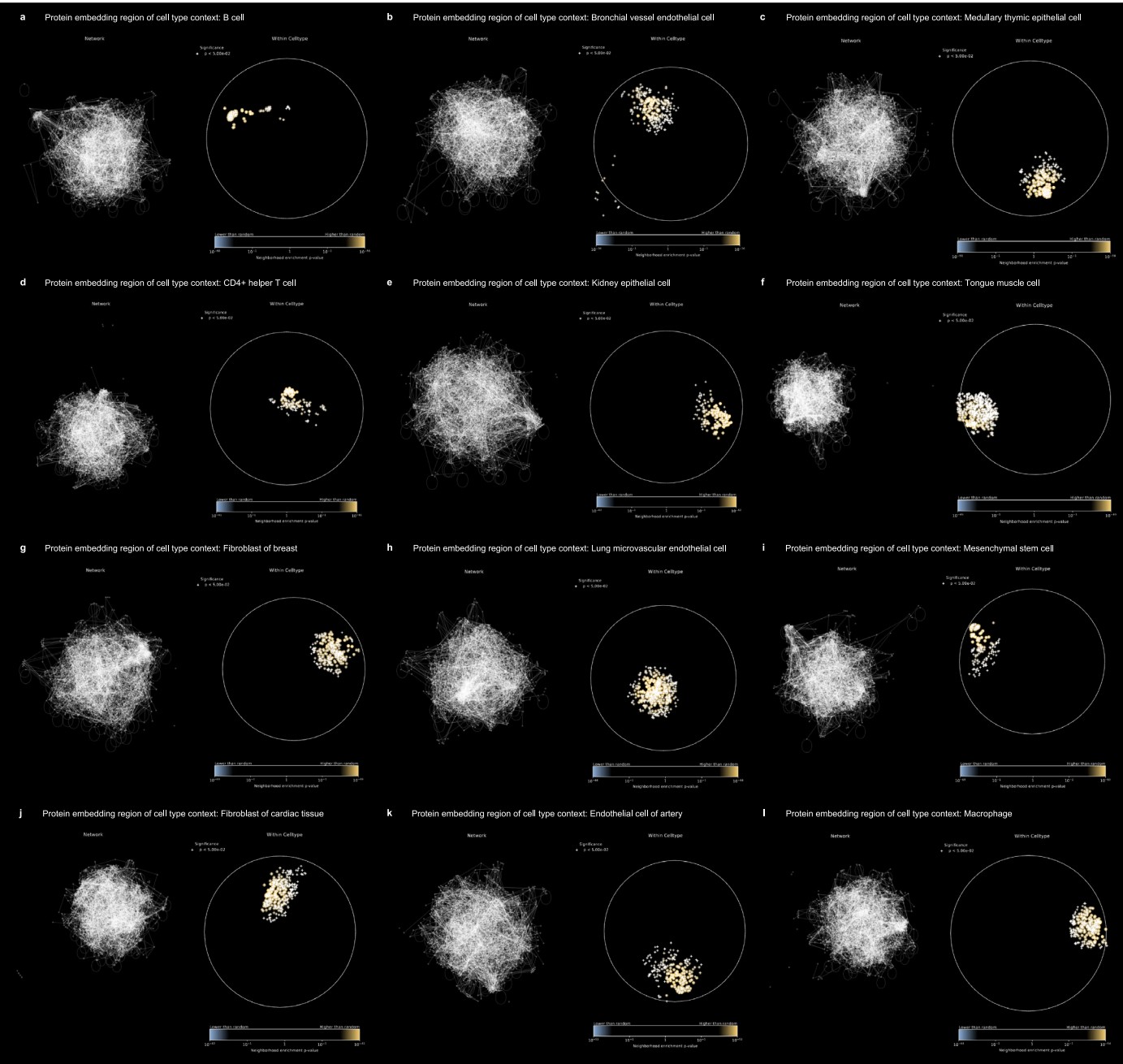

**Extended Data Fig. 3 | Spatial enrichment analysis of PINNACLE's protein embedding regions. (a-l)** For each cell type specific set of protein embeddings generated by PINNACLE, we sample a subset to construct a similarity network and perform spatial enrichment analysis using SAFE[31]. Shown for each cell type context is the network (left) and enrichment landscape (right). Dots represent the neighborhood enrichment p-value; crosses indicate a significant p-value < 0.05; hypergeometric test, adjusted using the Benjamin-Hochberg false discovery rate correction with significance cutoff $\alpha$ = 0.05.

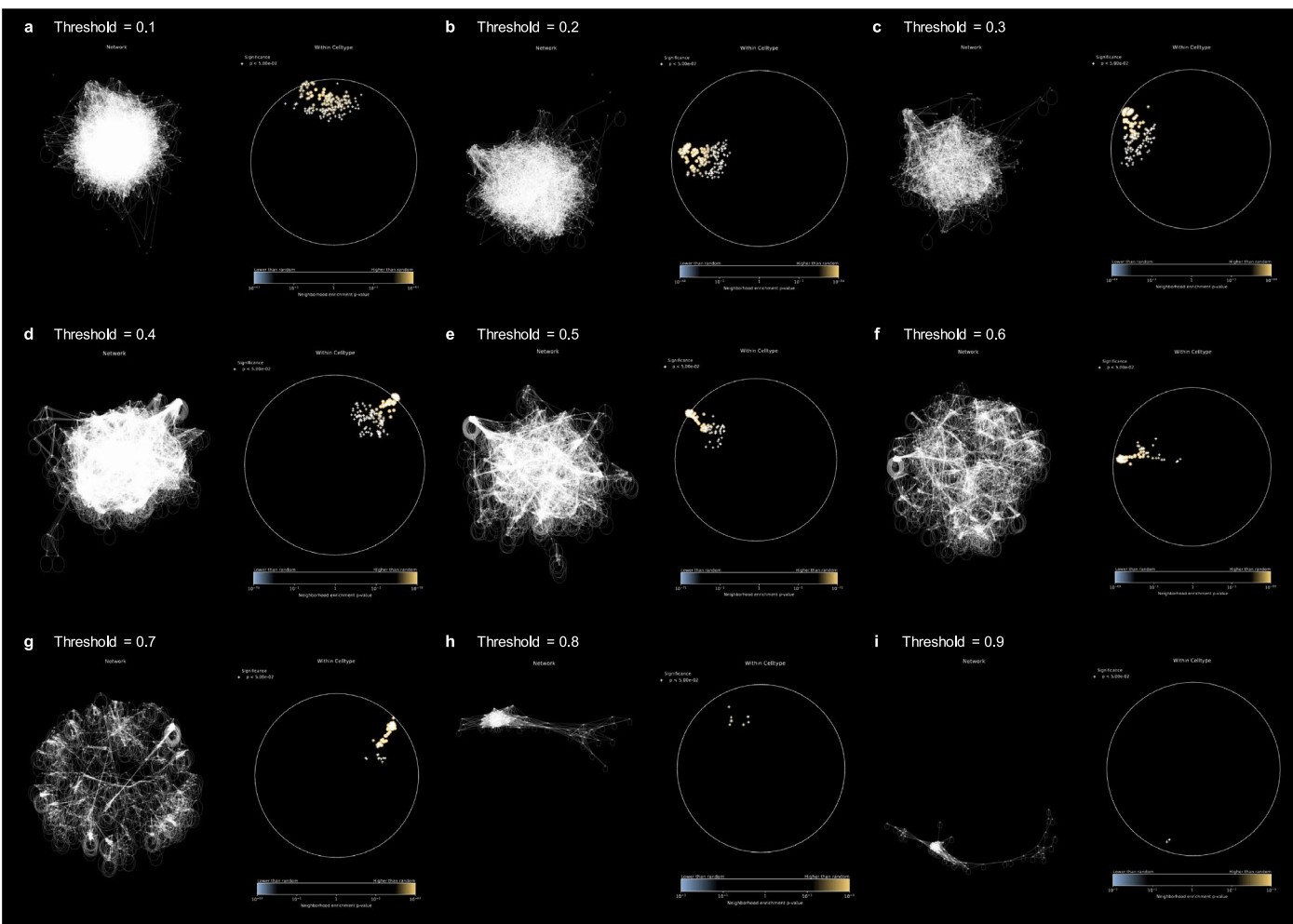

**Extended Data Fig. 4 | Spatial enrichment analysis of PINNACLE's protein embedding regions across thresholds. (a-i)** From the mesenchymal stem cell type specific protein embeddings generated by PINNACLE, we sample a subset to construct a similarity network and perform spatial enrichment analysis using SAFE[31]. Networks are constructed using a similarity threshold t ∈ [0.1, 0.2, 0.3,

0.4, 0.5, 0.6, 0.7, 0.8, 0.9]. Shown for each threshold is the network (left) and enrichment landscape (right). Dots represent the neighborhood enrichment p-value; crosses indicate a significant p-value < 0.05; hypergeometric test, adjusted using the Benjamin-Hochberg false discovery rate correction with significance cutoff α = 0.05.

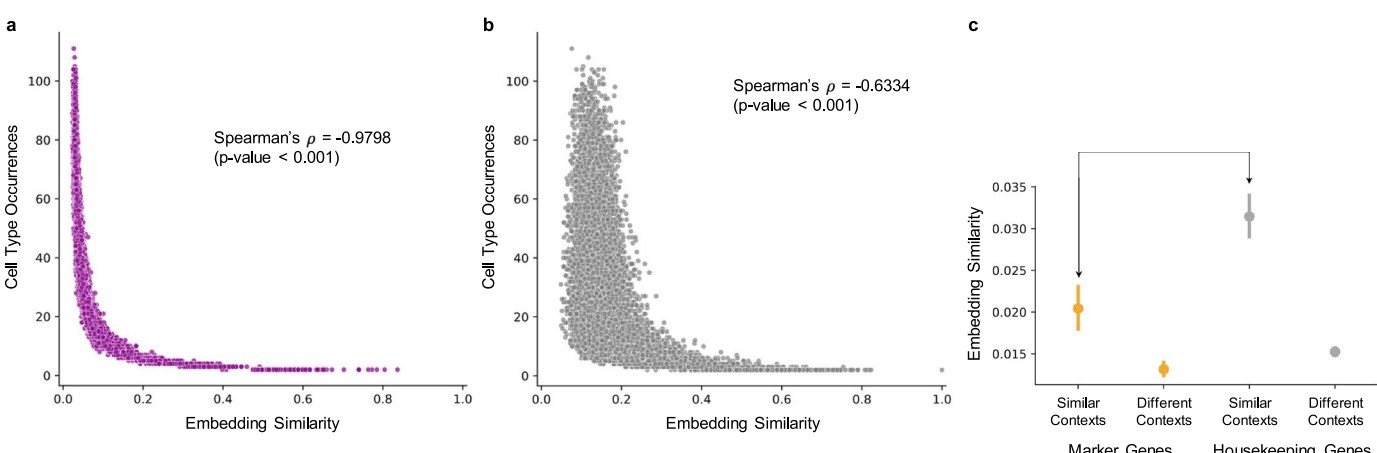

**Extended Data Fig. 5 | Embedding similarity based on proteins' cell type activation and function. (a-b)** Each dot represents a protein that is activated in at least two cell types. Shown is the average cosine similarity of embeddings for each protein as a function of the number of cell types that it is activated in **(a)** with (p-value < 0.001) and **(b)** without (p-value < 0.001) cellular and tissue context. Both Spearman correlation statistical tests for **(a)** and **(b)** are two-sided. **(c)** Comparison of embedding similarities of a marker (orange) or housekeeping (gray) gene's contextualized protein representation (from PINNACLE) across different cell type contexts. The marker genes are specific to cell types in the family of T lymphocytes (a total of 10 T lymphocyte cell types). For each marker/housekeeping gene, its cell type specific protein representations are compared in similar contexts (that is, between different T lymphocyte cell types) or different contexts (that is, between a T lymphocyte cell type and a non-immune cell type; a total of 115 non-immune cell types). All comparisons between these four groups shown are statistically significant. Cosine embedding similarity is used to compare contextualized protein representations. Data are represented as mean values with error bars indicating a 95% confidence interval.

**a**

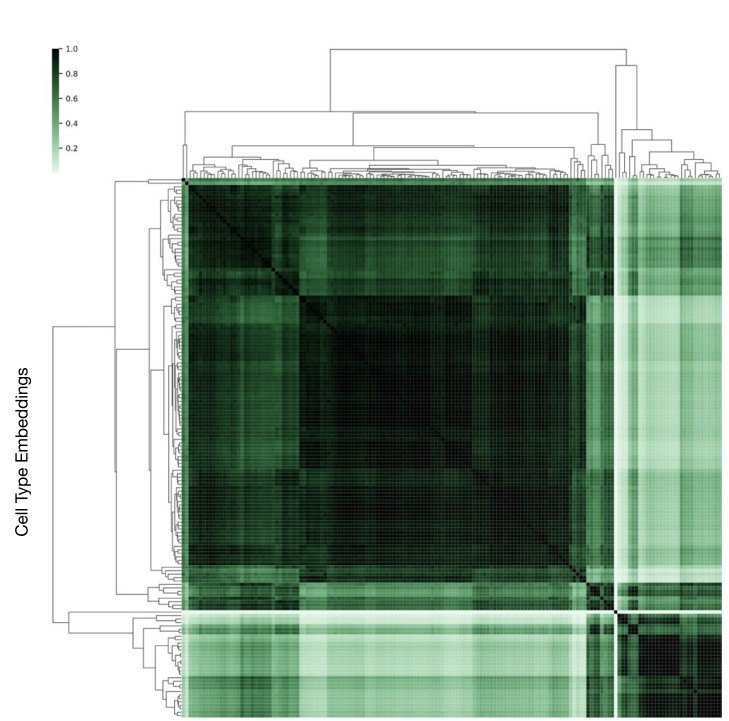

**b**

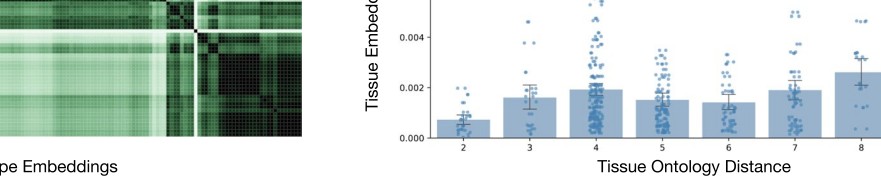

Spearmanr $\rho = -0.46$
(*p*-value = $8.01 \times 10^{-30}$)

**c**

Spearmanr $\rho = 0.11$
(*p*-value = 0.01)

**Extended Data Fig. 6 | Evaluation of PINNACLE's cell type and tissue representations. (a)** We quantify the quality of PINNACLE's cell type representations by calculating pairwise similarities of cell type representations. Pairwise similarities are computed via cosine similarity. We expect several major groups of cell type representations that are organized according to cellular and tissue hierarchy and acting as anchors for our complete set of cell type representations. This implies that the contextual information being transferred between the representations of cell types and proteins reflects the tissue hierarchy. Our results show that the local organization of PINNACLE's cell type representations (that is, identity of cell types in each group) reflects cellular communication, and the global organization of cell type representations

(that is, proximity of groups to each other) reflects tissue organization. Since PINNACLE's protein representations are embedded near their corresponding cell type representation, such organization is enforced among the contextualized protein representations as well. **(b)** Correlation between cosine distance of tissue representations and the fraction of overlapping cell types neighbors between the tissue pair. Spearman $\rho = -0.46$ with p-value = $8.01 \times 10^{-30}$. **(c)** Correlation between PINNACLE's tissue embedding distance to tissue ontology distance for leaf nodes in the metagraph. Spearman $\rho = 0.11$ with p-value = 0.01. All Spearman correlation statistical tests are two-sided. Data are represented as mean values with error bars indicating a 95% confidence interval. Both panels show n = 548 pairwise comparison calculations.

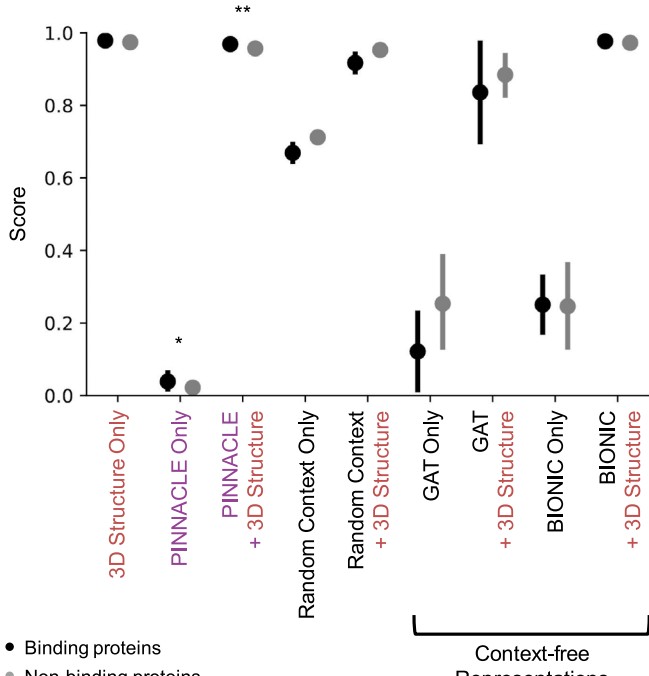

| Representation Type | Binding Proteins | Non-binding Proteins | Gap (p-value) |
|---|---|---|---|
| 3D Structure Only | 0.9789 ± 0.0004 | 0.9742 ± 0.0078 | 0.0047 (0.2121) |
| **PINNACLE Only** | **0.0385 ± 0.1531** | **0.0218 ± 0.1081** | **0.0167 (0.0299)** |
| **PINNACLE + 3D Structure** | **0.9690 ± 0.0049** | **0.9571 ± 0.0127** | **0.0119 ($< 10^{-5}$)** |
| Random Context Only | 0.6691 ± 0.1642 | 0.7122 ± 0.0445 | −0.0431 (1.0) |
| Random Context + 3D Structure | 0.9172 ± 0.1732 | 0.9529 ± 0.0277 | −0.0356 (1.0) |
| GAT Only | 0.1214 ± 0.1056 | 0.2533 ± 0.2898 | −0.1319 (0.6939) |
| GAT + 3D Structure | 0.8360 ± 0.1360 | 0.8847 ± 0.1255 | −0.0486 (0.5706) |
| BIONIC Only | 0.2506 ± 0.0760 | 0.2460 ± 0.2536 | 0.0046 (0.4556) |
| BIONIC + 3D Structure | 0.9769 ± 0.0018 | 0.9725 ± 0.0087 | 0.0043 (0.2797) |

**Extended Data Fig. 7 | Benchmarking context-free and contextualized 3D structure protein representations.** Shown are binding and non-binding scores (that is, cosine similarity) of proteins when using only 3D structure-based protein representations (p-value = 0.2121; n = 22 pairwise comparisons between 2 binding and 20 non-binding pairs), PINNACLE's contextualized protein representations (without 3D structural information; p-value = 0.0299; n = 7,956 pairwise computations between 180 binding and 7,776 non-binding pairs), contextualized structure-based protein representations (p-value < 10⁻⁵; n = 7,956 pairwise computations between 180 binding and 7,776 non-binding pairs), and baseline models. The baseline models are random context only (that is, randomly sampling pairs of PINNACLE's protein representations from different cell type contexts; p-value = 1.0; n = 7,956 pairwise computations between 180 'binding' and 7,776 'non-binding' pairs), concatenating random context protein representations with 3D structure-based protein representations (p-value = 1.0; n = 7,956 pairwise computations between 180 'binding' and 7,776 'non-binding' pairs), GAT only (that is, context-free protein representations generated by a graph attention neural network[44] on the global reference interactome; p-value

= 0.6939; n = 22 pairwise comparisons between 2 binding and 20 non-binding pairs), concatenating GAT protein representations with 3D structure-based protein representations (p-value = 0.5706; n = 22 pairwise comparisons between 2 binding and 20 non-binding pairs), BIONIC only (that is, context-free protein representations generated by BIONIC[15], a graph convolutional neural network designed for multi-modal network integration; p-value = 0.4556; n = 22 pairwise comparisons between 2 binding and 20 non-binding pairs), and concatenating BIONIC protein representations with 3D structure-based protein representations (p-value = 0.2797; n = 22 pairwise comparisons between 2 binding and 20 non-binding pairs). Note that all protein representations have consistent dimensions (328 = 200 structure-based protein representation + 128 context-aware/-free protein representation) to ensure that they are comparable. The protein representations without 3D structure are padded with 0's (that is, null 3D structure-based protein representation). The significance of the score gaps between binding and non-binding proteins is measured using a one-sided non-parametric permutation test. Data are represented as mean values with error bars indicating a 95% confidence interval.

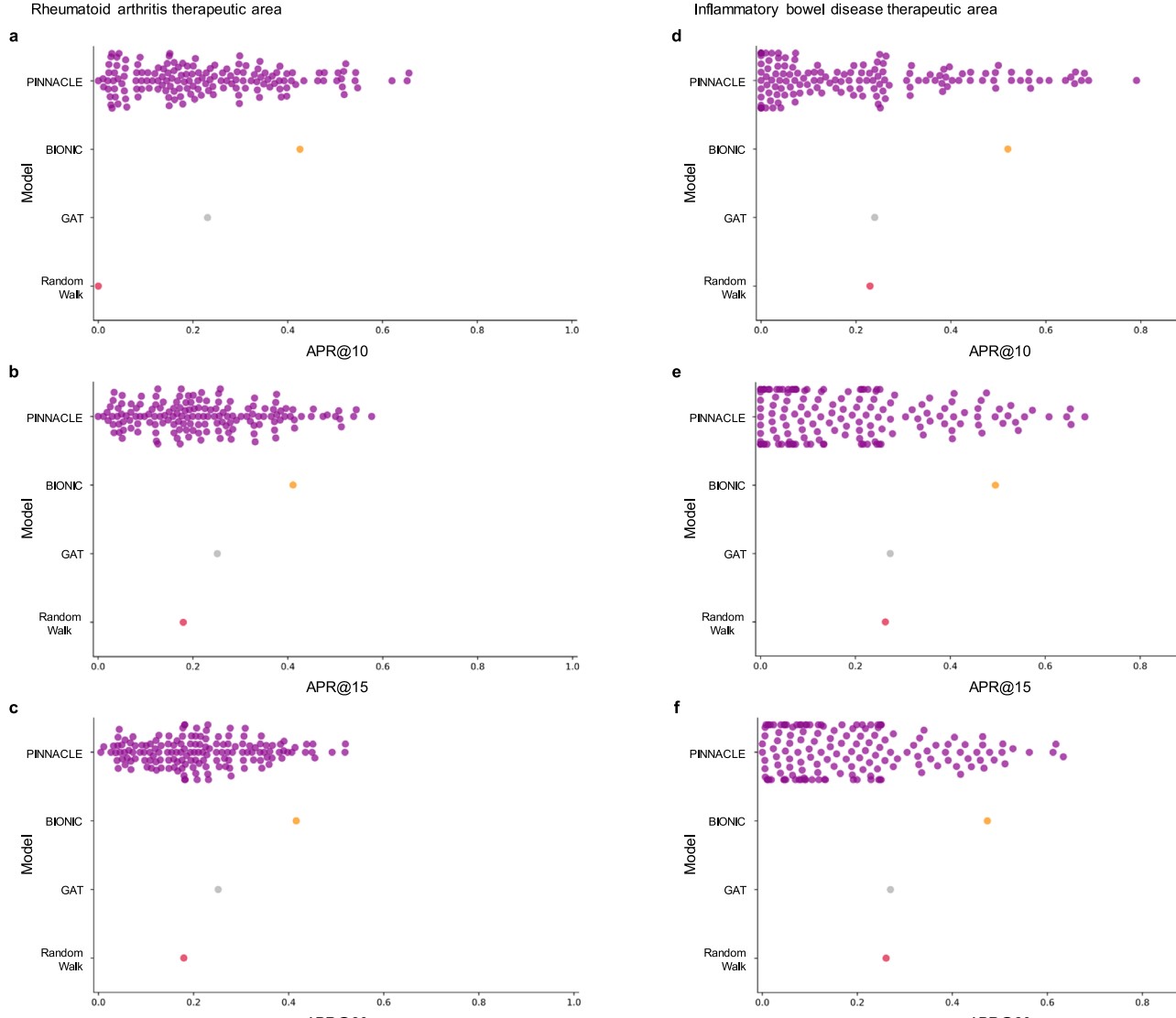

**Extended Data Fig. 8 | Performance of therapeutic target prioritization models for rheumatoid arthritis and inflammatory bowel diseases.** Benchmarking of context-aware and context-free approaches for **(a-c)** RA and **(d-f)** IBD therapeutic areas. Each dot is the performance (averaged across 10 random seeds) of protein representations from a given context (that is, cell type context for PINNACLE, context-free global reference protein interaction network for random walk[43] and GAT[44], and context-free multi-modal protein interaction network for BIONIC[15]). In the model for the RA therapeutic area: **(a)** at APR@10, 100% of cell types (156 out of 156) outperform the random walk model, 44.2% of cell types (69 out of 156) outperform GAT, and 11.5% of cell types (18 out of 156) outperform BIONIC. **(b)** At APR@15, 58.3% (91 out of 156) outperform the

random walk model, 38.5% of cell types (60 out of 156) outperform GAT, and 9.0% of cell types (14 out of 156) outperform BIONIC. **(c)** At APR@20, 59.0 (92 out of 156) outperform the random walk model, 34.6% of cell types (54 out of 156) outperform GAT, and 5.1% of cell types (8 out of 156) outperform BIONIC. In the model for the IBD therapeutic area: **(d)** at APR@10, 39.5% (60 out of 152) outperform the random walk model, 38.2% of cell types (58 out of 152) outperform GAT, and 10.5% of cell type (16 out of 152) outperform BIONIC. **(e)** At APR@15, 28.3% (43 out of 152) outperform the random walk model and GAT, and 8.6% of cell types (13 out of 152) outperform BIONIC. **(f)** At APR@20, 26.3% (40 out of 152) outperform the random walk model and GAT, and 6.6% of cell types (10 out of 152) outperform BIONIC.

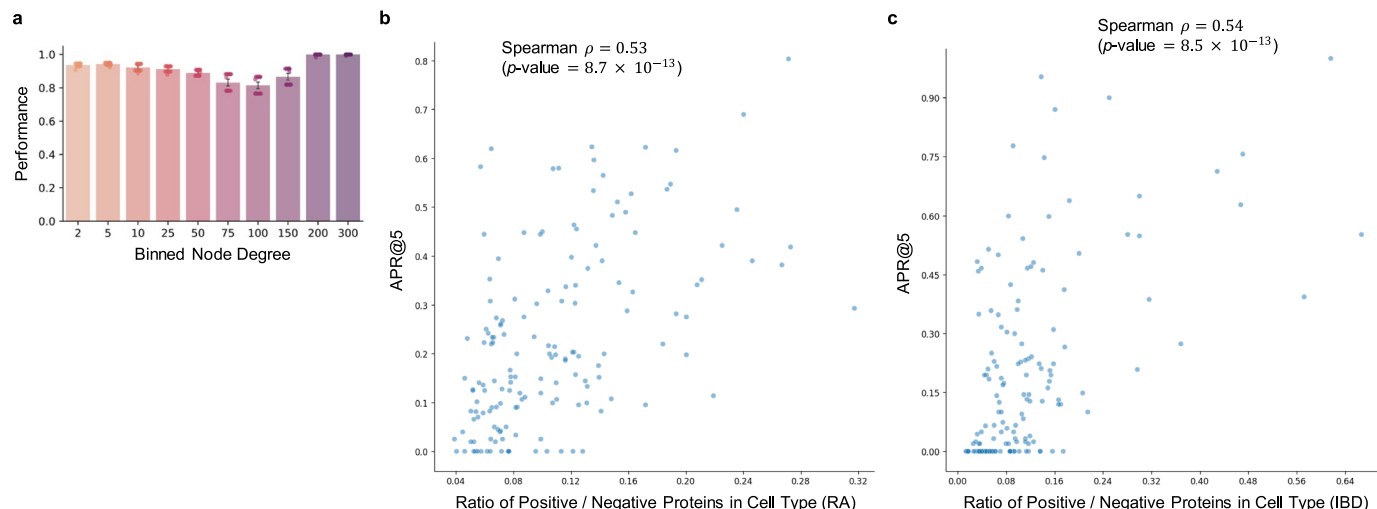

**Extended Data Fig. 9 | Correlating downstream performance on rheumatoid arthritis and inflammatory bowel diseases with protein degree and network enrichment. (a)** Correlation between the node degrees of proteins (in the cell type specific protein interaction networks) and the downstream performance of their learned representations. Combining the RA and IBD prediction results, the Spearman $\rho = 0.087$ with p-value = 0.223 (n = 36,229, consisting of 3,165 positive protein examples with label y = 1 and 33,064 negative protein examples with label y = 0). For RA only, the Spearman $\rho = 0.205$ with p-value = 0.041 (n = 26,773, consisting of 2,382 positive protein examples with label y = 1 and 24,391 negative protein examples with label y = 0). For IBD only, the Spearman $\rho = 0.024$ with p-value = 0.810 (n = 9,456, consisting of 783 positive protein examples

with label y = 1 and 8,673 negative protein examples with label y = 0). Data are represented as mean values with error bars indicating a 95% confidence interval. **(b-c)** Correlation between PINNACLE's performance and network enrichment. **(b)** Comparing PINNACLE's predicted performance (APR@5) and the ratio of positive to negative proteins in each cell type for RA (Spearman $\rho = 0.53$ with p-value = $8.7 \times 10^{-13}$; n = 26,773, consisting of 2,382 positive proteins with label y = 1 and 24,391 negative proteins with label y = 0). **(c)** Comparing PINNACLE's predicted performance (APR@5) and the ratio of positive to negative proteins in each cell type for IBD (Spearman $\rho = 0.54$ with p-value = $8.5 \times 10^{-13}$; n = 9,456, consisting of 783 positive proteins with label y = 1 and 8,673 negative proteins with label y = 0). All Spearman correlation statistical tests are two-sided.

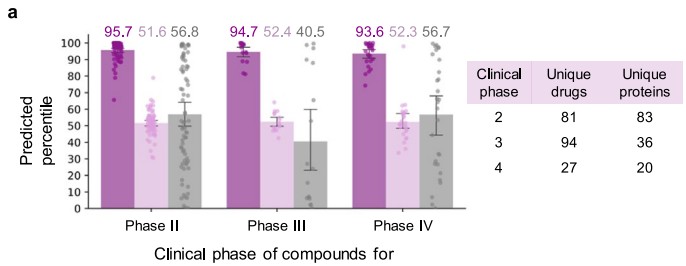

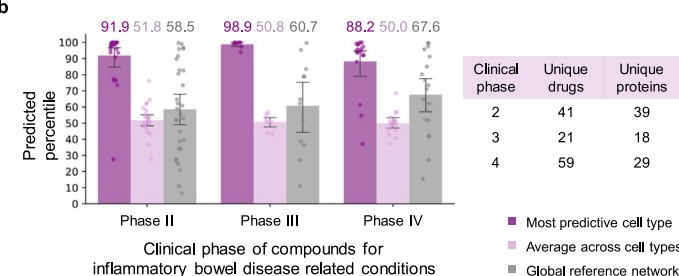

**Extended Data Fig. 10 | Performance of therapeutic target prioritization models for rheumatoid arthritis and inflammatory bowel diseases stratified by clinical trials.** Comparison of the percentiles of drug targets across cell types, in their best-performing cell types, and in the context-free global reference model, stratified by clinical phase of compounds for **(a)** RA and **(b)** IBD. The table shows the number of unique drugs in each clinical phase, as well as the numbers of unique proteins targeted by those drugs. Data are represented as mean values with error bars indicating a 95% confidence interval.

# Reporting Summary

## Statistics

For all statistical analyses, confirm that the following items are present in the figure legend, table legend, main text, or Methods section.

| n/a | Confirmed | |
|---|---|---|
| ☐ | ☒ | The exact sample size (*n*) for each experimental group/condition, given as a discrete number and unit of measurement |
| ☒ | ☐ | A statement on whether measurements were taken from distinct samples or whether the same sample was measured repeatedly |
| ☐ | ☒ | The statistical test(s) used AND whether they are one- or two-sided <br> *Only common tests should be described solely by name; describe more complex techniques in the Methods section.* |
| ☐ | ☒ | A description of all covariates tested |
| ☐ | ☒ | A description of any assumptions or corrections, such as tests of normality and adjustment for multiple comparisons |
| ☐ | ☒ | A full description of the statistical parameters including central tendency (e.g. means) or other basic estimates (e.g. regression coefficient) AND variation (e.g. standard deviation) or associated estimates of uncertainty (e.g. confidence intervals) |
| ☐ | ☒ | For null hypothesis testing, the test statistic (e.g. *F*, *t*, *r*) with confidence intervals, effect sizes, degrees of freedom and *P* value noted <br> *Give P values as exact values whenever suitable.* |
| ☒ | ☐ | For Bayesian analysis, information on the choice of priors and Markov chain Monte Carlo settings |
| ☒ | ☐ | For hierarchical and complex designs, identification of the appropriate level for tests and full reporting of outcomes |
| ☐ | ☒ | Estimates of effect sizes (e.g. Cohen's *d*, Pearson's *r*), indicating how they were calculated |

*Our web collection on statistics for biologists contains articles on many of the points above.*

## Software and code

Policy information about availability of computer code

| Data collection | Python implementation of the methodology developed and used in the study is available via the project website at https://zitniklab.hms.harvard.edu/projects/PINNACLE. The code to reproduce results, together with documentation and examples of usage, are available on GitHub at https://github.com/mims-harvard/PINNACLE. We provide an interactive demo via HuggingFace to explore PINNACLE's contextualized protein representations |
|---|---|
| Data analysis | Python implementation of the methodology developed and used in the study is available via the project website at https://zitniklab.hms.harvard.edu/projects/PINNACLE. The code to reproduce results, together with documentation and examples of usage, are available on GitHub at https://github.com/mims-harvard/PINNACLE. We provide an interactive demo via HuggingFace to explore PINNACLE's contextualized protein representations. <br><br> We visualize PINNACLE embeddings using a uniform manifold approximation and projection for dimension reduction (UMAP package version 0.5, https://umap-learn.readthedocs.io/en/latest/) and Seaborn (seaborn package version 0.13, https://seaborn.pydata.org/index.html). Additionally, we use the Python implementation of SAFE version 1, https://github.com/baryshnikova-lab/safepy. <br><br> We implement PINNACLE using Pytorch (Version 1.12.1) (Paszke et al., 2019) and Pytorch Geometric (Version 2.1.0) (Fey et al., 2019). We leverage Weights and Biases (Biewald 2020) for hyperparameter tuning and model training visualization, and we create interactive demos of the model using Gradio (Abid et al. 2019). |

For manuscripts utilizing custom algorithms or software that are central to the research but not yet described in published literature, software must be made available to editors and reviewers. We strongly encourage code deposition in a community repository (e.g. GitHub). See the Nature Portfolio guidelines for submitting code & software for further information.

## Data

Policy information about [availability of data](availability%20of%20data)

All manuscripts must include a [data availability statement](data%20availability%20statement). This statement should provide the following information, where applicable:

- Accession codes, unique identifiers, or web links for publicly available datasets
- A description of any restrictions on data availability
- For clinical datasets or third party data, please ensure that the statement adheres to our [policy](policy)

---

All data used in the paper are shared via the project website at https://zitniklab.hms.harvard.edu/projects/PINNACLE.

Our global reference protein-protein interaction (PPI) network is the union of physical multi-validated interactions from BioGRID (Oughtred et al., 2019), the Human Reference Interactome (HuRI) (Luck et al., 2020), and Menche et al., 2015 with 15,461 nodes and 207,641 edges. Different sources of PPI have their own methods of curating and validating physical interactions between proteins. BioGRID, HuRI, and Menche et al. are PPI networks from three well-cited publications and databases regarding human protein interactions. By joining the three networks, we construct a comprehensive global PPI network for our analysis.

We leverage Tabula Sapiens (Tabula Sapiens Consortium, 2022) data source as our multi-organ, single-cell transcriptomic atlas of humans. The data consists of 15 donors, with 59 specimens total. There are 483,152 cells after quality control, of which 264,824 are immune cells, 104,148 are epithelial cells, 31,691 are endothelial cells, and 82,478 are stromal cells. The cells correspond to 177 unique cell ontology classes.

For 3D structural analyses, the proteins being compared are PD-1, PD-L1, B7-1, CTLA-4, RalB, RalBP1, EPO, EPOR, C3, and CFH. The pairs of binding proteins are PD-1/PD-L1 (PDB ID: 4ZQK) and B7-1/CTLA-4 (PDB ID: 1I8L). The non-binding proteins are any of the four proteins paired with any of the remaining six proteins (e.g., PD-1/RalB, PD-1/RalBP1, PD-L1/RalBP1). The PDB IDs for the other six proteins are 2KWI for RalB/RalBP1, 1CN4 for EPO/EPOR, and 3OXU for C3/CFH.

We obtain labels for therapeutic targets from the Open Targets Platform (Ochoa et al., 2020).

---

## Human research participants

Policy information about [studies involving human research participants and Sex and Gender in Research.](studies%20involving%20human%20research%20participants)

| | |
|---|---|
| Reporting on sex and gender | This study did not involve human research participants. |
| Population characteristics | This study did not involve human research participants. |
| Recruitment | This study did not involve human research participants. |
| Ethics oversight | This study did not involve human research participants. |

Note that full information on the approval of the study protocol must also be provided in the manuscript.

# Field-specific reporting

Please select the one below that is the best fit for your research. If you are not sure, read the appropriate sections before making your selection.

☒ Life sciences  ☐ Behavioural & social sciences  ☐ Ecological, evolutionary & environmental sciences

For a reference copy of the document with all sections, see [nature.com/documents/nr-reporting-summary-flat.pdf](nature.com/documents/nr-reporting-summary-flat.pdf)

# Life sciences study design

All studies must disclose on these points even when the disclosure is negative.

| | |
|---|---|
| Sample size | No statistical methods were used to determine sample sizes. Small 95% confidence intervals or standard deviations indicate the chosen sample sizes were sufficient.<br><br>1) Cell type-specific protein interaction network samples. To ensure high-quality representations of cell types in our networks, we keep networks with at least 1,000 proteins. We do not perform subsampling of cells (i.e., sample the same number of cells per cell type) to minimize information loss for constructing protein interaction networks (Supplementary Figure S2).<br><br>2) Cell type and tissue relationship samples in the metagraph. As recommended by CellPhoneDB, cells are subsampled prior to running the algorithm, which uses geometric sketching [105] to efficiently sample a small representative subset of cells from massive datasets while preserving biological complexity. We choose to subsample 25% of cells and run CellPhoneDB for 100 iterations. We determine cell type-tissue relationships and extract tissue-tissue relationships using Tabula Sapiens meta-data. For relationships between cell types and tissues, we draw edges between cell types and the tissue that the cells were taken from. For tissue-tissue relationships, we select the nodes corresponding to the tissues where samples were taken from and all parent nodes up to the root of the BRENDA tissue ontology. We perform sensitivity and ablation analyses on different components of the metagraph (Supplementary Table S3-S5). |

3) Therapeutic area selection. To curate target information for a therapeutic area, we examine the drugs indicated for the therapeutic area of interest and its descendants. The two therapeutic areas examined are rheumatoid arthritis (RA) and inflammatory bowel disease. For rheumatoid arthritis, we collected therapeutic data (i.e., targets of drugs indicated for the therapeutic area) from OpenTargets for rheumatoid arthritis (EFO_0000685), ankylosing spondylitis (EFO_0003898), and psoriatic arthritis (EFO_0003778). For inflammatory bowel disease, we collected therapeutic data for ulcerative colitis (EFO_0000729), collagenous colitis (EFO_1001293), colitis (EFO_0003872), proctitis (EFO_0005628), Crohn's colitis (EFO_0005622), lymphocytic colitis (EFO_1001294), Crohn's disease (EFO_0000384), microscopic colitis (EFO_1001295), inflammatory bowel disease (EFO_0003767), appendicitis (EFO_0007149), ulcerative proctosigmoiditis (EFO_1001223), and small bowel Crohn's disease (EFO_0005629).

We define positive examples (i.e., where the label y = 1$ as proteins targeted by drugs that have at least completed phase 2 of clinical trials for treating a certain therapeutic area. As such, a protein is a promising candidate if a compound that targets the protein is safe for humans and effective for treating the disease. We retain positive training examples that are activated in at least one cell type specific protein interaction network. The final number of positive training examples for RA and IBD are 152 and 114, respectively.

We define negative examples (i.e., where the label y = 0) as druggable proteins that do not have any known association with the therapeutic area of interest according to OpenTargets. A protein is deemed druggable if it is targeted by at least one existing drug. We extract drugs and their nominal targets from DrugBank. We retain negative training examples that are activated in at least one cell type specific protein interaction network. The final number of negative training examples for RA and IBD are 1,465 and 1,377, respectively.

**Data exclusions**  No data was excluded from the analysis.

**Replication**  Data on biological replicates were subject to statistical tests to ensure effects were significant. All replication attempts were successful. For the analyses of tissue hierarcy (Figure 3c), tissue ontology was shuffled 10 times to produce gray distribution. For analyses of RA/IBD models, ll the experiments were performed independently 10 times using different randomly-selected seeds.

**Randomization**  Randomization was performed using unbiased, non-parametric, random sampling and perturbation-based null hypothesis testing.

**Blinding**  Any group allocations were programmatically randomly generated and not assigned by the investigators, so blinding is not relevant to this study.

# Reporting for specific materials, systems and methods

We require information from authors about some types of materials, experimental systems and methods used in many studies. Here, indicate whether each material, system or method listed is relevant to your study. If you are not sure if a list item applies to your research, read the appropriate section before selecting a response.

## Materials & experimental systems

| n/a | Involved in the study |
|---|---|
| ☒ | Antibodies |
| ☒ | Eukaryotic cell lines |
| ☒ | Palaeontology and archaeology |
| ☒ | Animals and other organisms |
| ☒ | Clinical data |
| ☒ | Dual use research of concern |

## Methods

| n/a | Involved in the study |
|---|---|
| ☒ | ChIP-seq |
| ☒ | Flow cytometry |
| ☒ | MRI-based neuroimaging |

