## [Peer Review File · Nature Methods]

Peer Review Information

Manuscript Title: Contextual AI models for single-cell protein biology

Corresponding author name(s): Marinka Zitnik

Editorial Notes: None

Reviewer Comments & Decisions:

Decision Letter, initial version:

Dear Professor Zitnik,

Your Article, "Contextualizing protein representations using deep learning on protein networks and single-cell data", has now been seen by 2 reviewers. As you will see from their comments below, although the reviewers find your work of considerable potential interest, they have raised a number of concerns. We are interested in the possibility of publishing your paper in Nature Methods, but would like to consider your response to these concerns before we reach a final decision on publication.

We therefore invite you to revise your manuscript to address these concerns.

[Redacted]

We hope to receive your revised paper within 6 weeks. If you cannot send it within this time, please let us know. In this event, we will still be happy to reconsider your paper at a later date so long as nothing similar has been accepted for publication at Nature Methods or published elsewhere.

OPEN SCIENCE REQUIREMENTS

REPORTING SUMMARY AND EDITORIAL POLICY CHECKLISTS

IMAGE INTEGRITY

When submitting the revised version of your manuscript, please pay close attention to our Digital Image Integrity Guidelines and to the following points below:

Finally, please ensure that you retain unprocessed data and metadata files after publication, ideally

archiving data in perpetuity, as these may be requested during the peer review and production process or after publication if any issues arise.

DATA AVAILABILITY

All novel DNA and RNA sequencing data, protein sequences, genetic polymorphisms, linked genotype and phenotype data, gene expression data, macromolecular structures, and proteomics data must be deposited in a publicly accessible database, and accession codes and associated hyperlinks must be provided in the "Data Availability" section.

CODE AVAILABILITY

Please include a "Code Availability" subsection in the Online Methods which details how your custom code is made available. Only in rare cases (where code is not central to the main conclusions of the paper) is the statement "available upon request" allowed (and reasons should be specified).

For more information on our code sharing policy and requirements, please see:
<https://www.nature.com/nature-research/editorial-policies/reporting-standards#availability-of-computer-code>

MATERIALS AVAILABILITY

SUPPLEMENTARY PROTOCOL

To help facilitate reproducibility and uptake of your method, we ask you to prepare a step-by-step Supplementary Protocol for the method described in this paper. We encourage authors to share their step-by-step experimental protocols on a protocol sharing platform of their choice and report the protocol DOI in the reference list. Nature Portfolio 's Protocol Exchange is a free-to-use and open resource for protocols; protocols deposited in Protocol Exchange are citable and can be linked from the published article. More details can found at www.nature.com/protocolexchange/about.

ORCID

Nature Methods is committed to improving transparency in authorship. As part of our efforts in this direction, we are now requesting that all authors identified as 'corresponding author' on published papers create and link their Open Researcher and Contributor Identifier (ORCID) with their account on the Manuscript Tracking System (MTS), prior to acceptance. This applies to primary research papers only. ORCID helps the scientific community achieve unambiguous attribution of all scholarly contributions. You can create and link your ORCID from the home page of the MTS by clicking on 'Modify my Springer Nature account'. For more information please visit please visit www.springernature.com/orcid.

Sincerely,
Arunima

Arunima Singh, Ph.D.
Senior Editor

Nature Methods

Reviewers' Comments:

Reviewer #1:

Remarks to the Author:

Summary:

The manuscript introduces PINNACLE, a self-supervised geometric deep learning model designed to generate contextualized protein representations from protein interaction networks under different cell types. The model produces a unified embedding space, effectively capturing cell type and tissue contexts of proteins. Spatial enrichment analysis indicates that the generated protein embeddings retain information about protein localization. The application of zero-shot retrieval of the tissue hierarchy further suggests distances between tissues were embedded in the protein representations. In addition, the authors demonstrate that incorporating the context-aware protein representations with protein 3D structures better differentiates binding and non-binding proteins. The authors further extend the utility of these contextualized protein representations by incorporating them into a therapeutic target prediction model for Rheumatoid Arthritis (RA) and Inflammatory Bowel Disease (IBD), showing that this model outperforms existing models that lack context-specific information.

The manuscript is well-structured and addresses a significant gap in the existing literature by introducing a model that generates contextualized protein representations. The model has potential for applications in drug target identification for specific cell types. While this paper concludes with robust findings, there are some gaps in illustration and clarity noted in the major and minor comments that follow. Additionally, the absence of certain statistical analyses in some of the presented results limits full assessment of the findings (see specific comments below). Overall, the manuscript is relatively strong with some limitations and issues with clarity.

Major Comments:

1. Lack of clarity and consistency between Figure 1 and main text. There is a lack of clarity in the description of the model outputs and the applications of the PINNACLE model:

a) The main text suggests that the PINNACLE model results in a unified embedding space of proteins that captures both cell type and tissue information (lines 120-121). Figure 1d implies that the model produces separate embeddings, however.

b) Figure 1f-h illustrates the application of PINNACLE including multi-modal deep learning and transfer learning, and contextualized predictions. However, transfer learning is mentioned in the introduction section without explicit linkage to the figure, and multi-modal deep learning is not explicitly mentioned in the overview of PINNACLE. It is clear, in the later text, that contextualized predictions is a separate application, but It's also not clear whether the other techniques (multimodal deep learning and transfer learning) are part of the PINNACLE model or separate entities. To address these issues, I suggest a major revision of Figure 1, especially d, f-h along with corresponding changes in the text. The revised figure should explicitly indicate the output of the PINNACLE model (one embedding space or separate), and distinguish from the applications. Additionally, the text should clarify the role and integration of multi-modal deep learning and transfer learning techniques within the PINNACLE model, or as its applications.

2. Insufficient details on the therapeutic target prediction model. The main text currently describes the drug-target prediction model as the binary classification, while it is actually regarding an underlying Multi-Layer Perceptron (MLP) architecture. Although Figure 4c attempts to visualize this model architecture, it is not referenced or elaborated upon in the main text. In addition, authors did not include a table or figure panel detailing the training, validation, and test sets used for model evaluation, although mentioned in the Method. Additionally, the distribution of protein cell types in each data set shall be informative as a supplementary figure showing the model is not biased. To enhance clarity and completeness, the authors should: 1) Provide a more descriptive overview of the binary classification model, specifically stating its MLP-based architecture, and referring explicitly to Figure 4c in the text. 2) Include a table or figure panel to show the training, validation, and test data sets used, along with the distributions of cell types in each set.

Minor Comments:

Last paragraph of the Introduction. The number of details describing the evaluation of the model is very high as an introductory paragraph. Suggest being more concise, and elaborate the details in later Methods or Results sections.

Line 88 "As a result, we have 156 context-aware protein interaction networks, each with $2,530 \pm 5676.8$ proteins": Precision here is unreasonably high as there is no possible way the authors can compute such a number to that accuracy. At the very least suggest rounding the average number of proteins to the nearest integer. Same comment applies to all other instances where this specific number is mentioned throughout the paper.

The second paragraph of Results reads like a Discussion paragraph and is actually also covered in the Discussion. Suggest removing from the Result and keep in the Discussion.

Fig 1a depicts how the protein interaction networks were generated, while it leaves a gap in explaining how edges between different cell types or tissues are determined. Recommend an additional panel to explicitly illustrate how cell-cell interactions are inferred, along with a simplified representation of the tissue hierarchy.

Fig 1c: Please add labels to the dashed arrow lines in the figure legend for clarification. The term 'repelled edge' is misleading. Especially if cell type C1 is interacting with cell type C2, why are the proteins in C1 repelling proteins in C2? Based on the described loss function, the model aims to separate unrelated entities, but the authors need to be careful with related but not directly interactive entities. I recommend that the authors clarify this point either in the text or by adjusting the terminology or illustration associated with 'repelled edges.'

Line 145-line 150: The repetitive numbers in these two sentences create confusion. Suggest combining them into a single sentence focusing on what is in the unified embedding space (output of PINNACLE).

Line 153-154: Please make the significant cutoff explicit in both the main text and the Methods section. And apply multiple testing corrections when conducting a hypergeometric test.

Line 158: Clarify what is meant by "pairwise similarity" in this context.

Fig 3a-b: For clarity, either keep the legend separate or place the two figures side by side. Additionally, conduct statistical analysis when comparing protein embeddings across the same and

different cell types.

Fig 3c: The two-sample KS test results for comparing the distributions of tissue ontology and random categories should be included in the figure legend as well.

Fig 3c: An abrupt drop in embedding distance is observed when the ontology distance reaches 10. Please provide an explanation for this in the main text.

Line 215-217: The result of using PINNACLE representations without incorporating 3D structures is mentioned but is not shown in any main figures/ supplemental figures. Suggest including one figure (supplementary is fine) as a reference. Additionally, consider reordering the presentation of results to first show the impact of incorporating both 3D structures and context-aware PPI networks, followed by a comparison with context-free representations and the context-aware representation alone.

Figure 3f: While differences between context-free and context-aware protein representations are shown, there is no statistical analysis of these differences, particularly in binding vs non-binding proteins. Please include statistical tests to validate these differences. In addition, the magnitude of the differences in scores between binding vs non-binding are small. Without infusing 3D structure information, the average difference between interacting vs non-interacting proteins also reaches 0.016 (mentioned in Line 217). Why 3D structure needs to be integrated with the contextualized protein representations. Please comment on this in the manuscript.

Ordering of the text and figure: Fig4 d-e were mentioned after Fig5 a and d. The model benchmarking should flow better after the overview of the drug-target prediction model. Consider the move paragraphs between line 296-314 to follow the paragraph that ends on line 257.

Reviewer #2:
Remarks to the Author:
Summary

This manuscript describes a method to obtain contextual protein embedding, by considering not only protein-protein interactions (PPI) but also between-cell and tissue-interaction as a meta-graph using graph attention networks for each context. The authors created cell-type specific PPIs based on highly-expressed genes within a single-cell dataset, a cell-cell graph based on a ligand-receptor database, and a tissue-tissue graph based on tissue ontology. The authors validated the model with quantitative geometric measurement of protein embeddings and also performance gain of PD/PDL1 protein interaction prediction with or without contextual protein embedding. Lastly, they fine-tuned the model to predict a protein's 'druggability' based on how often the protein is used or registered as the target for a clinical trial. They chose RA and IBD as the target diseases for these druggability predictions. Using the predictiveness of such druggability of each protein in a cell-type specific PPI, they showed that the model can capture the meaningful cell type that is related to the protein's mechanism of action.

The paper is very timely, technically interesting and addresses an important and challenging problem. The approach used by the authors -- making a more cell-type-specific contextualized protein embedding is interesting -- but we feel that some key technical details are missing and the technical

validation of the method is lacking. In particular, we believe the authors should conduct a more thorough ablation study, to identify which contextual information is critical for the method's performance, as well as include a more extensive ('fair') comparison with a baseline without contextual information. We elaborate these points below and look forward to a revised manuscript that addresses them, which could be an important paper.

Major Points

1. What is the overall architecture of the neural networks employed and their implementation details? Although the authors gave the hyperparameter space they browsed, the authors didn't share the final choice for the model (e.g., How many graph neural network layers are used for each graph? How many epochs the model is trained?). These details are crucial to include.
2. How is the model trained in a self-supervised manner? To perform a link prediction task, is the dropped link selected randomly? Also, how was train/valid/test split done for pre-training?
3. What are the critical design choices for the model? Does the performance change if the cell-cell graph or tissue-tissue graph is dropped? Does the performance change if the link prediction loss for the cell-cell graph or tissue-tissue graph is dropped?
4. How similar are the protein embeddings for proteins with similar functions across cell or tissue types? Is the same protein initialized with the same embedding across cell-type specific PPI? Although the authors show that proteins within the same cell type have more similar embedding, this is a rather easy 'validation bar', considering the nature of the message-passing method and that each cell-type specific protein-networks are trained as subgraph and the only connection between the other cell-types is cell-node which is done after pooling of whole proteins within the subgraph. Because the protein embeddings are adjusted by the cell embedding that harbors the information of the neighborhood cell, if the protein representation is meaningful enough then the proteins that have a similar role across cells types (eg housekeeping-like) should have similar embedding and the proteins that have a distinctively different role (cell type marker-like) should have different embedding.
5. What are the properties of each cell-type specific PPI graph and cell-cell, tissue-tissue graph? The variations of the number of nodes are large and should affect how dense the graph is. What is the variation in the graph degrees and how does it affect the protein embedding? The threshold for cell-cell connection seems low (at least one shared ligand-receptor pair); if the connection is sparser after setting up a high bar for connection, does it affect the performance? How does the cell-type specific network handle a node without an edge?
6. The score-gap between binding and non-binding proteins should be compared with another baseline, such as the BIONIC and GAT models. For the GAT model, is it trained with the same self-supervised link prediction loss? If not, then the results of Fig. 3f and Fig. 4d-e should also be compared with the GAT model trained with link prediction loss.
7. The score defined for binding and non-binding proteins is cosine similarity. Is it a good metric to measure protein embedding quality for binding and non-binding proteins? Since the message-passing method makes protein embeddings that frequently connect with each other more similar, that gap is enforced by the model. MaSIF tried to measure the prediction accuracy, so if the contextualized protein representation brings the performance increase to a binding prediction accuracy then it would

prove that the embedding quality of the protein is good.

8. What is the exact definition of the predictiveness of protein representation? The output of the MLP is the prediction for the supervised target and does not represent the confidence score of the model. In our understanding, it's better to measure the uncertainty of the model using conformal prediction techniques to statistically measure the predictiveness of the model. The authors should at least show if the range of the output of MLP is well aligned with the predictiveness of the model.

9. All the contexts for RA and IBD in Fig. 5a-b are cell types. How much correlation do these cell types have with the ratio of protein within that cell type? Is a complex representation required to obtain these cell type lists? Similarly, for Fig. 5b-f, if we measure the gene set enrichment score that includes JAK3, IL6R, or others, are we able to obtain the same cell type lists?

10. Do all results in Fig 4d-e and Fig 5. a-f. come from test datasets?

Minor Points

1. The authors seem to use GAT for each context level. It would be important to clarify this point to ensure the reader understands the model.

Author Rebuttal to Initial comments

Response to Reviewers

Contextualizing protein representations using deep learning on protein networks and single-cell data

Michelle M. Li¹, Yepeng Huang¹, Marissa Sumathipala¹, Man Qing Liang¹, Alberto Valdeolivas², Ashwin N. Ananthakrishnan^{1,3}, Katherine Liao^{1,4}, Daniel Marbach², and Marinka Zitnik^{1,5,6,7}

¹Harvard Medical School, Boston, MA, USA; ²Roche Pharma Research and Early Development, Pharmaceutical Sciences, Roche Innovation Center Basel, Basel, Switzerland; ³Division of Gastroenterology, Massachusetts General Hospital, Boston, MA, USA; ⁴Division of Rheumatology, Inflammation, and Immunity, Brigham and Women's Hospital, Boston, MA, USA; ⁵Kempner Institute for the Study of Natural and Artificial Intelligence, Harvard University, MA, USA; ⁶Broad Institute of MIT and Harvard, Cambridge, MA, USA; ⁷Harvard Data Science Initiative, Cambridge, MA, USA.

We thank the Reviewers for their valuable feedback and positive evaluation of this article. In response to the Reviewers' feedback, we have substantially changed the manuscript to improve the robustness and clarity of our analyses on contextualized protein representations. **In addition to revising the main and supplementary manuscript and answering the Reviewers' queries, we include nine new Response Figures and eight new Response Tables in this document.**

Our response is structured as follows:

- **Response to Reviewer 1 (page 3)**
- **Response to Reviewer 2 (page 12)**
- **Response Figures (page 24)**
- **Response Tables (page 32)**
- **Response References (page 44)**

Comments from Reviewers are in boxes, and our responses to each comment follow. Changes are marked in blue in the revised manuscript.

Our new analyses demonstrate the utility and robustness of PINNACLE through additional evaluation of PINNACLE's contextualized protein representations, as recommended by the Reviewers. We perform statistical tests and multiple hypothesis testing (if applicable) for analyses in the submitted paper, confirming that our findings are robust and statistically significant. We also conduct new analyses using PINNACLE's protein representations to show that: (1) despite not being optimized such that proteins with similar functions are embedded nearby, PINNACLE's protein representations can already capture functional similarity; (2) network properties, such as node degrees, do not influence PINNACLE's downstream performance on RA and IBD.

We perform ablation and sensitivity analyses on PINNACLE's metagraph to evaluate the contribution of each component. By removing the cell type edges, tissue edges, and the entire metagraph itself, we demonstrate that PINNACLE cannot capture cell type and tissue

organization in its embeddings without any of these components, clearly establishing that all elements of the PINNACLE model are necessary for overall strong performance. Additionally, we evaluate PINNACLE's contextualized representations with different cutoff values for the minimum number of significant ligand-receptor interactions between a pair of cell types to create an edge, and find that PINNACLE maintains a strong performance across a range of cutoff values, suggesting that the model behavior is robust.

We strengthen case studies and analyses examining contextualized 3D structure-based protein representations to improve the prediction of binding and non-binding proteins (i.e., PD-1/PD-L1 and CTLA-4/B7-1) by conducting additional statistical tests and benchmarks. Specifically, we investigate six new benchmarks and perform one-sided permutation tests to demonstrate that the score gaps observed using our contextualized structure-based protein representations are both significant and, importantly, they cannot be explained by various confounders. Our results comparing PINNACLE against context-free methods, such as GAT and BIONIC, suggest that context-free protein representations cannot predict intercellular communication (i.e., protein interactions between different cell types).

Further, we have revised the figures, introduction, results, discussion, and methods sections to improve their clarity and impact on the readers. We also mention the importance of uncertainty quantification, especially as the role of AI models in scientific discovery grows.

We believe that this revision, which carefully addresses the reviewers' valuable feedback, has confirmed the potential and impact of contextual AI models for biology. Since our manuscript submission, PINNACLE has been featured in Chan-Zuckerberg Initiative's CELLxGENE projects page (<https://cellxgene.cziscience.com/census-models>), and we have established new collaborations to leverage PINNACLE's pretrained contextualized models to interrogate biomarkers of neurological diseases and cancers. These follow-ups highlight the potential of PINNACLE to become a broadly used AI tool in biological research.

Response to Reviewer #1

The manuscript introduces PINNACLE, a self-supervised geometric deep learning model designed to generate contextualized protein representations from protein interaction networks under different cell types. The model produces a unified embedding space, effectively capturing cell type and tissue contexts of proteins. Spatial enrichment analysis indicates that the generated protein embeddings retain information about protein localization. The application of zero-shot retrieval of the tissue hierarchy further suggests distances between tissues were embedded in the protein representations. In addition, the authors demonstrate that incorporating the context-aware protein representations with protein 3D structures better differentiates binding and non-binding proteins. The authors further extend the utility of these contextualized protein representations by incorporating them into a therapeutic target prediction model for Rheumatoid Arthritis (RA) and Inflammatory Bowel Disease (IBD), showing that this model outperforms existing models that lack context-specific information.

The manuscript is well-structured and addresses a significant gap in the existing literature by introducing a model that generates contextualized protein representations. The model has potential for applications in drug target identification for specific cell types.

While this paper concludes with robust findings, there are some gaps in illustration and clarity noted in the major and minor comments that follow. Additionally, the absence of certain statistical analyses in some of the presented results limits full assessment of the findings (see specific comments below). Overall, the manuscript is relatively strong with some limitations and issues with clarity.

We thank the Reviewer for recognizing our work and providing valuable feedback that has helped us further improve the manuscript. Please refer to our responses below, where we thoroughly address each Reviewer's comments and suggestions.

Major Comments:

1. Lack of clarity and consistency between Figure 1 and main text. There is a lack of clarity in the description of the model outputs and the applications of the PINNACLE model:

a) The main text suggests that the PINNACLE model results in a unified embedding space of proteins that captures both cell type and tissue information (lines 120-121). **Figure 1d implies that the model produces separate embeddings**, however.

b) Figure 1f-h illustrates the application of PINNACLE including multi-modal deep learning and transfer learning, and contextualized predictions. However, transfer learning is mentioned in the introduction section without explicit linkage to the figure, and multi-modal deep learning is not explicitly mentioned in the overview of PINNACLE. It is clear, in the later text, that contextualized predictions is a separate application, but it's also not clear whether the other techniques (multimodal deep learning and transfer learning) are part of the PINNACLE model or separate entities. To address these issues, I suggest a major revision of Figure 1, especially d, f-h along with corresponding changes in the text. **The revised figure should explicitly indicate the output of the PINNACLE model (one embedding space or**

separate), and distinguish from the applications. Additionally, the text should clarify the role and integration of multi-modal deep learning and transfer learning techniques within the PINNACLE model, or as its applications.

We thank the Reviewer for these great suggestions to improve our illustration and description of PINNACLE in Figure 1. We have made the following modifications:

- To address point a), we add a gray rectangle around the Figure 1C panel and annotate it as a “unified embedding space.” As clarified in revised Figure 1C, the PINNACLE model produces a single unified embedding space, where regions of the embedding space correspond to cell types (ovals in revised Figure 1C) and points in each region represent a protein embedding in a given cell type context.
- To address point b) about clarifying model outputs versus model applications, we:
 - Explicitly label PINNACLE’s outputs in Figure 1 by adding a gray box with the text “Model outputs” on top of the illustrations of PINNACLE’s outputs (panel D) and prior methods’ outputs (panel E).
 - Add a gray box with the text “Applications of PINNACLE” on top of the illustrations in panels F, G, and H.
- Finally, we add details in the main text (under the section “Overview of PINNACLE model”) about multimodal and transfer learning so that the text reflects the figure better. The revised figure clearly communicates that PINNACLE is a self-supervised representation learning model. Once the model is pre-trained, the learned context-specific protein representations (embeddings) can be used for downstream applications, including multimodal deep learning, contextualized prediction, and transfer learning across contexts.

Further, we make the following minor changes to improve the clarity of Figure 1:

- We add gray boxes with panel titles to better differentiate between panels (e.g., model outputs versus model applications).
- We fix the inconsistent dashed lines in Figures 1C and 1D.
- The legend clarifies that the red line in Figure 1C indicates “increase distance in embedding space” and the orange line signifies “decrease distance in embedding space.” We also remove the dashed lines in Figure 1D because the dashed lines do not add any information nor have a consistent meaning, as in Figure 1C.

2. Insufficient details on the therapeutic target prediction model.

The main text currently describes the drug-target prediction model as the binary classification, while it is actually regarding an underlying Multi-Layer Perceptron (MLP) architecture. Although Figure 4C attempts to visualize this model architecture, it is not referenced or elaborated upon in the main text.

In addition, authors did not include a table or figure panel detailing the training, validation, and test sets used for model evaluation, although mentioned in the Methods. Additionally, the distribution of protein cell types in each data set shall be informative as a

supplementary figure showing the model is not biased.

To enhance clarity and completeness, the authors should: 1) Provide a more descriptive overview of the binary classification model, specifically stating its MLP-based architecture, and referring explicitly to Figure 4c in the text. 2) Include a table or figure panel to show the training, validation, and test data sets used, along with the distributions of cell types in each set.

We thank the Reviewer for the valuable comment. We have made the following changes and additions to the manuscript.

Firstly, we have added a sentence (and reference to Figure 4C) at the end of the second paragraph introducing the therapeutic prediction task setup. For your convenience, here is the added sentence:

The binary classification model can be of any architecture; our results for nominating RA and IBD therapeutic targets are generated by independently training a multi-layer perceptron for each therapeutic area (Figure 4C).

We also slightly modified Figure 4C to clarify that our prediction task is a binary classification: given a cell type-specific protein representation, output a probability (between 0 and 1) that the protein is a target for a given therapeutic area.

Next, we have included three tables (**Response Tables 1, 2, 3**) in this document that provide statistics about the dataset splits for the RA and IBD models:

- **Response Table 1** (Supplementary Table S6 in the revised manuscript) summarizes the sizes of the train, validation, and test datasets for the RA and IBD models.
- **Response Tables 2 and 3** show the distribution of cell types in the train, validation, and test datasets for the RA and IBD models, respectively.

Due to the number of cell types in our datasets, we provide the data splits of a subset of cell types. This subset of cell types is already examined in Supplementary Tables S1 and S2 because of their involvement in RA and IBD, respectively, according to the literature.

Finally, we have added the following details in the Methods section on how the dataset splits are created. While we aim to create train, validation, and test dataset splits such that they contain approximately 60%, 20%, and 20% of the total protein representations, respectively, there are two important criteria that we consider when splitting the proteins:

1. Proteins are assigned to train (60%), validation (20%), and test (20%) datasets based on their identity. This is to prevent data leakage where cell type-specific representations of a single protein are observed in multiple data splits.
2. We ensure sufficient numbers of train, validation, and test positive samples per cell type (number of test positive samples > 5). Proteins may be reassigned to a different data split so that each cell type is represented during training, validating, and testing stages.

With these criteria, the train, validation, and test dataset splits may not necessarily consist of approximately 60%, 20%, and 20% of the total protein representations.

Minor Comments:

Last paragraph of the Introduction. The number of details describing the evaluation of the model is very high as an introductory paragraph. Suggest being more concise, and elaborate the details in later Methods or Results sections.

Thank you for your suggestion! We have removed the sentences that explain the specific tasks (and their takeaways) from the last paragraph of the introduction. These details are already thoroughly described in the Results section.

Line 88 "As a result, we have 156 context-aware protein interaction networks, each with $2,530 \pm 676.8$ proteins": Precision here is unreasonably high as there is no possible way the authors can compute such a number to that accuracy. At the very least suggest rounding the average number of proteins to the nearest integer. Same comment applies to all other instances where this specific number is mentioned throughout the paper.

Thank you for your comment! We have incorporated your feedback: we round to the nearest integer for all mentions of this number.

The second paragraph of Results reads like a Discussion paragraph and is actually also covered in the Discussion. Suggest removing from the Result and keep in the Discussion.

This is a good point, and we agree. Per your suggestion, we have removed this paragraph from the Results section. The content is now only covered in the Discussion section.

Fig 1a depicts how protein interaction networks were generated, while it leaves a gap in explaining how edges between different cell types or tissues are determined. Recommend an additional panel to explicitly illustrate how cell-cell interactions are inferred, along with a simplified representation of the tissue hierarchy.

Thank you for your suggestion! You make an excellent point. We have revised Figure 1A to include a new panel that illustrates the schema of the metagraph with cell type and tissue nodes, how cell-cell interactions are inferred, and a simplified representation of the tissue hierarchy.

Fig 1c: Please add labels to the dashed arrow lines in the figure legend for clarification. The term 'repelled edge' is misleading. Especially if cell type C1 is interacting with cell type C2, why are the proteins in C1 repelling proteins in C2? Based on the described loss function, the model aims to separate unrelated entities, but the authors need to be careful with related but not directly interactive entities. I recommend that the authors clarify this point either in the text or by adjusting the terminology or illustration associated with 'repelled edges.'

Thanks for your valuable suggestion. We have changed the legend of the "attract" and "repel" edges in Figure 1c to "decrease embedding distance" and "increase embedding distance,"

respectively. We have also removed the dashed lines, as they did not provide any additional/helpful information.

Line 145-line150: The repetitive numbers in these two sentences create confusion. Suggest combining them into a single sentence focusing on what is in the unified embedding space (output of PINNACLE).

Thank you for your suggestion! Since the statistic on the average number of proteins per cell type-specific PPI network is mentioned earlier in the text, we decided to remove the first of the two sentences and focus on the representations in the unified embedding space.

Line 153-154: Please make the significant cutoff explicit in both the main text and the Methods section. And apply multiple testing corrections when conducting a hypergeometric test.

Thank you for your suggestion!

- We explicitly provide the significance cutoff ($\alpha = 0.05$) in the results (last sentence of the first paragraph under "PINNACLE's representations capture cellular and tissue organization") and Methods section 6.3.
- We apply Benjamin-Hochberg FDR correction to the hypergeometric tests. The figures (Figure 2 and Supplementary Figures S4 and S5) are updated to show the corrected scores. The Methods section 6.3 is updated to include the name of the multiple hypothesis testing procedure (Benjamin-Hochberg FDR correction).

Line 158: Clarify what is meant by "pairwise similarity" in this context.

Thank you for your suggestion! We have rephrased the sentence for clarity: "To quantify the quality of PINNACLE's protein embedding regions, we calculate **the similarities between protein** representations across cell type contexts." The following sentences in the text describe the different types of similarity calculations we consider in the analysis. To summarize the text:

- For Figure 3A-B, in dark purple (panel A) and dark gray (panel B), given two proteins from the same cell type, we calculate cosine similarity between the protein embeddings from that cell type. In light purple (panel A) and light gray (panel B), given two proteins from different cell types, we calculate cosine similarity between the protein embeddings from different cell types. We hypothesized that high-quality embeddings would be such that protein embeddings from the same cell type were more similar to each other than protein embeddings from different cell types. Analyses in Figure 3A-B confirm this hypothesis, providing another piece of evidence that PINNACLE's context-aware embeddings are meaningful.
- For Supplementary Figure S6, we calculate the cosine similarity of protein embeddings from different cell types.

Fig 3A-B: For clarity, either keep the legend separate or place the two figures side by side. Additionally, conduct statistical analysis when comparing protein embeddings across the same and different cell types.

Thank you for your suggestion!

- We have split up the legend for clarity.
- We conduct a statistical analysis comparing protein embeddings across the same and different cell types. We perform a two-sample Kolmogorov-Smirnov test (via the `ks_2samp` function in the `scipy.stats` library) and include the p-values in the figures.

Fig 3C: The two-sample KS test results for comparing the distributions of tissue ontology and random categories should be included in the figure legend as well.

Thank you for your comment. We have included the following in the revised Figure 3C:

- Spearman's rho and p-value for the null distribution based on the non-parametric permutation test (shuffled tissue ontology).
- Two-sample KS test results for comparing the distribution of tissue embedding distance to a standard normal distribution.

Fig 3C: An abrupt drop in embedding distance is observed when the ontology distance reaches 10. Please provide an explanation for this in the main text.

We thank you for your observation. We hypothesize (and confirm) that the drop in tissue embedding distance (at the ontology distance of 10) is due to:

1. The pairs of tissue nodes with the ontology distance of 10 are leaf nodes (i.e., have no children nodes that are tissues).
2. Leaf tissue nodes capture information from cell type and tissue neighbors (i.e., cell type identity is important, as well as parents of the tissues in the ontology).

Regarding point #1, we confirm that the pairs of tissue nodes with ontology distance of 10 are leaf nodes. The only comparisons are: small intestine (BTO:0000651) - uterus (BTO:0001424) and large intestine (BTO:0000706) - uterus (BTO:0001424).

Regarding point 2, we calculate the number of overlapping cell types and correlate it with the cosine distance of learned embeddings (**Response Figure 1** in this document / Supplementary Figure S9a in the revised manuscript). We apply Spearman rho's correlation and find that $\rho = -0.45$ with p-value = 1.91×10^{-20} . This result suggests that the embeddings of tissue leaves (in the metagraph) contain information about both cell type identity and tissue neighborhood. Additionally, we correlate tissue embedding distance with tissue ontology distance of the leaf nodes (**Response Figure 2** in this document / Supplementary Figure S9b in the revised manuscript), finding that the Spearman $\rho = 0.100$ with p-value = 0.018.

Due to the small number of tissue pairs with the ontology distance of 10 in our metagraph, we omit them from Figure 3C. It is not possible to perform statistically robust analyses on them while the tissue atlases and ontology continue to grow. The BRENDA Tissue Ontology is an evolving tissue ontology; it is continuously updated as new cellular and tissue relationships are established. As such, it is not surprising that the leaves of the ontology (i.e., the more specific terms) are those with less information and less established. Thank you for bringing this point to our attention.

Line 215-217: The result of using PINNACLE representations without incorporating 3D structures is mentioned but is not shown in any main figures/ supplemental figures. Suggest including one figure (supplementary is fine) as a reference.

Additionally, consider reordering the presentation of results to first show the impact of incorporating both 3D structures and context-aware PPI networks, followed by a comparison with context-free representations and the context-aware representation alone.

We appreciate your suggestions to improve the clarity and cohesiveness of our manuscript. We have made the following modifications:

- We have added a supplementary figure (**Response Figure 3** in this document / Supplementary Figure S10 in the revised manuscript) to show the score gap of binding and non-binding proteins using PINNACLE's representations alone (i.e., no 3D structures). We also reference the figure in the main text when discussing this score gap.
- We have reordered the text to discuss the contextualized 3D embedding results first and then the context-aware and context-free-only results.

Figure 3F: While differences between context-free and context-aware protein representations are shown, there is no statistical analysis of these differences, particularly in binding vs non-binding proteins. **Please include statistical tests to validate these differences.**

In addition, the magnitude of the differences in scores between binding vs. non-binding are small. Without infusing 3D structure information, the average difference between interacting vs non-interacting proteins also reaches 0.016 (mentioned in Line 217). Why does 3D structure need to be integrated with the contextualized protein representations? **Please comment on this in the manuscript.**

We thank you for this great suggestion. We have added a supplementary figure (**Response Figure 3** in this document / Supplementary Figure S10 in the revised manuscript) to provide the results of the statistical tests that validate the score gaps between binding and non-binding proteins. For the statistical test, we perform a one-sided non-parametric permutation test:

1. Concatenate the scores for the N binding pairs and M non-binding pairs
2. For 100,000 iterations:
 - a. Randomly sample N scores as the new set of binding protein scores and M scores as the new set of non-binding protein scores

- b. Calculate the mean μ_N of the N binding protein scores and the mean μ_M of the M non-binding protein scores
 - c. Calculate the score gap by taking the difference of the means as $\mu_N - \mu_M$
 - d. Keep track of the score gaps that are greater than or equal to the true score gap calculated from the real data
3. Calculate the p-value, defined as the fraction of 100,000 iterations in which the permuted score gap is greater than or equal to the true score gap (i.e., one-sided non-parametric permutation test).

Further, we benchmark scores between binding vs non-binding proteins against two null distributions (**Response Figure 3**):

- **Random selection of cell type contexts**: Randomly sample pairs of PINNACLE's protein representations from different cell type contexts. The score gap between binding and non-binding proteins is -0.0431 (statistically insignificant), indicating that randomly sampling pairs of proteins from different cell type contexts cannot produce the score gap observed in the contextualized protein representations (PINNACLE without 3D structure) nor contextualized structure-based protein representations (PINNACLE with 3D structure).
- **Random cell type context + 3D structure**: Concatenate protein representations from a random cell type context together with 3D structure-based protein representations. The score gap between binding and non-binding proteins is -0.0356 (statistically insignificant), indicating that concatenating randomly sampled pairs of proteins from different cell type contexts cannot produce the score gap observed in the contextualized structure-based protein representations (nor the contextualized protein representations without 3D structure).

Additionally, we benchmark scores between binding vs. non-binding proteins against four context-free approaches (**Response Figure 3**):

- **GAT only**: Context-free protein representations generated by a graph attention neural network (GAT) pretrained via self-supervised link prediction loss on the global reference interactome. The score gap between binding and non-binding proteins is -0.1319 (statistically insignificant).
- **GAT + 3D structure**: Concatenate GAT protein representations with 3D structure-based protein representations. The score gap between binding and non-binding proteins is -0.0486 (statistically insignificant).
- **BIONIC only**: Context-free protein representations pretrained via self-supervised link prediction loss on the cell type-specific protein interaction networks by BIONIC, a graph convolutional neural network designed for multi-modal network integration. The score gap between binding and non-binding proteins is 0.0046 (statistically insignificant).
- **BIONIC + 3D structure**: Concatenate BIONIC protein representations with 3D structure-based protein representations. The score gap between binding and non-binding proteins is 0.0043 (statistically insignificant).

These results indicate that context-free protein representations cannot predict intercellular communication (i.e., protein interactions between different cell types).

Note that all protein representations have consistent dimensions (328 = 200 structure-based protein representation + 128 context-aware/-free protein representation) to ensure they are comparable. The "Random context only," "GAT only," and "BIONIC only" protein representations are padded with 0's (i.e., null 3D structure-based protein representation).

Finally, thank you for the great suggestion to improve our motivation in the manuscript for adding context to 3D structure-based protein representations. We have included a sentence at the end of the first paragraph of the subsection, "Context improves 3D structure prediction of PD-1/PD-L1 and B7-1/CTLA-4 protein interactions," to better motivate the task/analysis and our hypothesis. For your convenience, here is the added text:

Because 3D structures of molecules, which contain precise atom or residue level contact information, provide complementary knowledge to protein-protein interaction networks, which summarize binary interactions between proteins, we expect that context-aware protein interaction networks can further improve the ability to differentiate between binding and non-binding proteins across different cell types (Braberg et al. 2022).

We cite a review paper published in *Nature Reviews Genetics* titled "From systems to structure—using genetic data to model protein structures" (Braberg et al. 2022) to reaffirm our point. In short, we emphasize that 3D structure and protein interaction networks are complementary, and often, both are necessary to elucidate protein mechanisms and design drugs to bind to and disrupt target protein-protein interactions. Our analysis is a proof-of-concept to demonstrate the value of adding cell type context to 3D structures that are otherwise captured in a single context. **Response Figure 3** further shows that our contextualized structure-based protein representations yield meaningful score gaps between binding and non-binding proteins across different cell types.

Ordering of the text and figure: Fig 4 D-E were mentioned after Fig 5 A and D. The model benchmarking should flow better after the overview of the drug-target prediction model. Consider the move paragraphs between line 296-314 to follow the paragraph that ends on line 257.

Thank you for your helpful suggestion. We have reordered the paragraphs as per your recommendation. Now, we discuss the experimental setup for cell type-specific therapeutic target prediction (Figure 4A-C), the benchmarking setup, and then the results (Figure 4D-E). Then, the detailed results and case studies are shown in Figure 5.

Response to Reviewer #2

This manuscript describes a method to obtain contextual protein embeddings by considering not only protein-protein interactions (PPI) but also between-cell and tissue-interactions as a meta-graph using graph attention networks for each context. The authors created cell-type specific PPIs based on highly-expressed genes within a single-cell dataset, a cell-cell graph based on a ligand-receptor database, and a tissue-tissue graph based on tissue ontology.

The authors validated the model with quantitative geometric measurement of protein embeddings and also performance gain of PD/PDL1 protein interaction prediction with or without contextual protein embedding. Lastly, they fine-tuned the model to predict a protein's 'druggability' based on how often the protein is used or registered as the target for a clinical trial. They chose RA and IBD as the target diseases for these druggability predictions. Using the predictiveness of such druggability of each protein in a cell-type specific PPI, they showed that the model can capture the meaningful cell type that is related to the protein's mechanism of action.

The paper is very timely, technically interesting and addresses an important and challenging problem. The approach used by the authors -- making a more cell-type-specific contextualized protein embedding is interesting -- but we feel that some key technical details are missing and the technical validation of the method is lacking. In particular, we believe the authors should conduct a more thorough ablation study to identify which contextual information is critical for the method's performance, as well as include a more extensive ('fair') comparison with a baseline without contextual information. We elaborate these points below and **look forward to a revised manuscript that addresses them, which could be an important paper.**

We thank the Reviewer for recognizing the algorithmic innovation and impact of our work and providing valuable feedback to improve this manuscript. We provide a detailed point-by-point response below, where we thoroughly address each of the Reviewer's comments.

Major Points

1. What is the overall architecture of the neural networks employed and their implementation details? Although the authors gave the hyperparameter space they browsed, the authors didn't share the final choice for the model (e.g., How many graph neural network layers are used for each graph? How many epochs the model is trained?). These details are crucial to include.

Thank you for your question! We have included the requested details in the Online Methods section 3.5 ("Hyperparameter tuning"). For your convenience, here is the added text:

The best hyperparameters are as follows: the dimension of the nodes' feature matrix = 1024, the dimension of the output layer = 16, lambda = 0.1, the learning rate for link prediction task = 0.01, the learning rate for the protein's cell type classification task = 0.1, number of attention heads = 8, weight decay rate = 0.00001, dropout rate = 0.6, and

normalization layers are layer norm and batch norm. Further, PINNACLE has two custom graph attention neural network layers (as specified in Section 3) for each cell type-specific PPI network and another one for the metagraph. The model is trained for 250 epochs until it converges.

2. How is the model trained in a self-supervised manner? To perform a link prediction task, is the dropped link selected randomly? Also, how was train/valid/test split done for pre-training?

We thank you for your questions, which have helped us clarify the manuscript. To answer your questions:

1. PINNACLE is trained in a self-supervised manner using cell type identity and graph connectivity (i.e., cell type-specific protein interaction networks and metagraph) as a supervised signal to define positive vs. negative links in self-supervised link prediction. For the edge type specific link prediction tasks, PINNACLE predicts whether an edge (and its type, i.e., protein-protein, cell type-cell type, tissue-tissue) exists between a pair of nodes. For the cell type identification term of the loss function, PINNACLE is predicting the cell type(s) that the protein is activated in.
2. For each link prediction task, a randomly selected subset of edges is masked from the model during each batch update of PINNACLE model training. This edge masking approach is an established strategy for pre-training graph neural networks (Hu et al., Strategies for Pre-training Graph Neural Networks, ICLR 2020). Practically, this means that the graphs being fed as input into PINNACLE during train, validation, or test do not contain the masked edges.
3. For pretraining, protein-protein edges are randomly split into train, validation, and test sets. The metagraph edges are not split into train, validation, and test sets because there are relatively few of them and they are critical for inducing cell type and tissue organization in the model (as shown in **Response Table 4** in this document / Supplementary Table S5 in the revised manuscript). The proteins involved in the train edges are considered in the cell type identification term of the loss function.

We have revised the Methods section by adding a subsection to Methods section 3.5 that describes self-supervised learning and data split creation. For your convenience, here is the added text:

PINNACLE is trained in a self-supervised manner using cell type identity and graph connectivity (i.e., cell type-specific protein interaction networks and metagraph) as a supervised signal to define positive vs. negative links in self-supervised link prediction. Specifically, PINNACLE predicts whether an edge (and its type) exists between a pair of nodes and the cell type(s) in which the protein is activated. For the link prediction tasks, a randomly selected subset of edges is masked from the model. Practically, this means that the graphs being fed as input into PINNACLE during train, validation, or test do not contain the masked edges. Protein-protein edges are randomly split into train, validation, and test sets. The metagraph edges are not split into train, validation, and test sets because there are only a few of them, and they are critical for injecting cell type and

tissue organization into the model. The proteins involved in the train edges are considered in the cell type identification term of the loss function.

3. What are the critical design choices for the model? Does the performance change if the cell-cell graph or tissue-tissue graph is dropped? Does the performance change if the link prediction loss for the cell-cell graph or tissue-tissue graph is dropped?

Thank you for this great suggestion! We perform several ablation studies, summarized in **Response Table 4**, and find that all components of the metagraph are critical for full performance of PINNACLE models.

As per your request, we investigate three ablated versions of PINNACLE models:

- Removing cell-type-to-cell-type relationships (i.e., excluding the effect of link prediction loss for cell-cell graph): Since the metagraph is required to propagate neural messages between cell type-specific protein interaction networks, it is not possible to drop the cell type nodes nor their edges without breaking the model. So, we “remove” cell type nodes/edges by shuffling the cell type nodes’ identities.
- Removing tissue-to-tissue relationships (i.e., excluding the effect of link prediction loss for tissue-tissue graph): Similar to the previous ablation model, we “remove” tissue nodes/edges by shuffling the tissue nodes’ identities.
- Removing the metagraph (i.e., dropping cell-cell graph and tissue-tissue graph): Similar to the previous two ablation models, we “remove” the metagraph by setting the weight of the metagraph-related terms in the objective function to zero (i.e., set $\theta = 1$ so that the weight of the protein-related terms, θ , is 1 and the weight of the metagraph-related terms, $(1 - \theta)$, is 0). Note that θ is a tunable hyperparameter; in the complete model, $\theta = 0.3$.

We calculate three performance metrics to evaluate how each ablated version of the PINNACLE model captures cell type and tissue organization in the embedding space:

- Spearman’s correlation between tissue embedding distance (computed using the model’s tissue representations) and tissue ontology distance. In a well-performing model, we expect a positive correlation.
- Spearman’s correlation between tissue embedding distance (computed using the model’s tissue representations) and tissue ontology distance among the tissue leaf nodes of the metagraph. In a well-performing model, we expect a positive correlation (see **Response Figures 1 and 2** for more details).
- Spearman’s correlation between tissue embedding distance and fraction of overlapping cell types. In a well-performing model, we expect a strong negative correlation (see **Response Figures 1 and 2** for more details).

Our results show that:

- Removing cell-type-to-cell-type relationships yields a stronger correlation (compared to the complete model) between tissue embedding distance and tissue ontology distance among nodes in the entire metagraph (**Response Table 4**, column 2). At the same time,

the correlation between tissue embedding distance and fraction of overlapping cell type nodes is ~2X weaker (**Response Table 4**, column 4). Such a result is expected because the composition of the tissues' cell type neighbors is no longer meaningful. In other words, information about tissues from which a cell type is extracted is removed. Due to the propagation of neural messages from the metagraph to the cell type-specific protein interaction networks, the cell type-specific protein representations also lose cell type identity and organization.

- Removing tissue-to-tissue relationships yields a negative correlation between tissue embedding distance vs. tissue ontology distance (**Response Table 4**, columns 2 and 3). At the same time, the correlation between tissue embedding distance and fraction of overlapping cell type nodes is ~3X weaker (**Response Table 4**, column 4). This finding is expected because the tissue nodes have a random structure. Due to the propagation of neural messages from the metagraph to the cell type-specific protein interaction networks, the cell type-specific protein representations also lose tissue organization.
- Removing the metagraph yields a slightly weaker correlation between tissue embedding distance and tissue ontology among nodes in the entire metagraph (**Response Table 4**, column 2). Interestingly, the correlation between tissue embedding distance and tissue ontology distance among the tissue leaf nodes of the metagraph is negative (**Response Table 4**, column 3). Similar to the other two ablation models, the correlation between tissue embedding distance and fraction of overlapping cell types is weaker (**Response Table 4**, column 4). This finding suggests that cell type organization can be significantly affected by the loss of the metagraph. Due to the propagation of neural messages from the metagraph to the cell type-specific protein interaction networks, the cell type-specific protein representations also lose cell type identity and organization.

4. How similar are protein embeddings for proteins with similar functions across cell or tissue types? Is the same protein initialized with the same embedding across cell-type specific PPI?

Although the authors show that proteins within the same cell type have more similar embedding, this is a rather easy 'validation bar', considering the nature of the message-passing method and that each cell-type specific protein-networks are trained as subgraph and the only connection between the other cell-types is cell-node which is done after pooling of whole proteins within the subgraph. Because the protein embeddings are adjusted by the cell embedding that harbors the information of the neighborhood cell, if the protein representation is meaningful enough then **the proteins that have a similar role across cells types (eg housekeeping-like) should have similar embedding and the proteins that have a distinctively different role (cell type marker-like) should have different embedding.**

Thank you for the insightful comments.

First, to answer your questions:

- Yes, proteins of the same identity are initialized with the same random Gaussian vector to maintain their identity during training (we have included this detail in the final sentence of Methods section 3.1).

- To clarify, the cell type-specific protein interaction networks are a set of networks that are “connected” indirectly via the metagraph. In other words, no edges exist between the cell type-specific protein interaction networks.

Next, as per your suggestion, we have analyzed how similar protein representations are in different cell type contexts:

- **Data:**
 - We extract human housekeeping genes from the Housekeeping and Reference Transcript Atlas (<https://housekeeping.unicamp.br/>, Hounkpe et al. 2021)
 - We extract marker genes from the human gold standard T lymphocyte-specific protein functional networks from HumanBase (<https://hb.flatironinstitute.org/>, Greene et al., 2015) (accessed on November 20th, 2023). Only edges of level C1 (i.e., tissue-specific) are kept. The nodes corresponding to these edges are considered to be marker genes for cell types in the family of T lymphocytes.
 - The lists of marker and housekeeping genes do not overlap, as we remove any overlapping housekeeping genes from the list of marker genes.
 - The 10 T lymphocyte cell types in this analysis are:
 - [CD4+ helper T cell, CD4+ alpha-beta memory T cell, CD8+ alpha-beta cytokine secreting effector T cell, CD8+ alpha-beta cytotoxic T cell, DN1 thymic pro-T cell, Mature natural killer T cell, Naive regulatory T cell, Naive thymus-derived CD4+ alpha-beta T cell, Regulatory T cell, Type I natural killer T cell]
 - The 115 non-immune cells in this analysis are:
 - [endothelial cell of hepatic sinusoid, fibroblast, hepatocyte, intrahepatic cholangiocyte, tracheal goblet cell, endothelial cell, smooth muscle cell, ciliated cell, ionocyte, secretory cell, basal cell, mucus secreting cell, serous cell of epithelium of trachea, stromal cell, acinar cell of salivary gland, pericyte cell, endothelial cell of lymphatic vessel, adventitial cell, duct epithelial cell, myoepithelial cell, epithelial cell, vein endothelial cell, endothelial cell of artery, capillary endothelial cell, tongue muscle cell, keratinocyte, fibroblast of breast, luminal epithelial cell of mammary gland, vascular associated smooth muscle cell, epithelial cell of uterus, ciliated epithelial cell, myometrial cell, conjunctival epithelial cell, eye photoreceptor cell, muller cell, limbal stem cell, retinal blood vessel endothelial cell, epithelial cell of lacrimal sac, corneal keratocyte, retinal pigment epithelial cell, corneal epithelial cell, limbal stromal cell, melanocyte, retinal bipolar neuron, lacrimal gland functional unit cell, radial glial cell, ocular surface cell, retina horizontal cell, ciliary body, myofibroblast cell, muscle cell, cardiac endothelial cell, cardiac muscle cell, fibroblast of cardiac tissue, pancreatic acinar cell, pancreatic stellate cell, pancreatic ductal cell, pancreatic beta cell, basal cell of prostate epithelium, hillock-club cell of prostate epithelium, luminal cell of prostate epithelium, endothelial cell of vascular tree, skeletal muscle satellite stem cell, tendon cell, fast muscle cell, slow muscle cell, mesothelial cell,

salivary gland cell, medullary thymic epithelial cell, adipocyte, retinal ganglion cell, bladder urothelial cell, enterocyte of epithelium of large intestine, large intestine goblet cell, paneth cell of epithelium of large intestine, transit amplifying cell of large intestine, gut endothelial cell, intestinal crypt stem cell of large intestine, intestinal enteroendocrine cell, intestinal tuft cell, type ii pneumocyte, bronchial vessel endothelial cell, respiratory mucous cell, club cell of prostate epithelium, hillock cell of prostate epithelium, sperm, enterocyte of epithelium of small intestine, transit amplifying cell of small intestine, small intestine goblet cell, paneth cell of epithelium of small intestine, intestinal crypt stem cell of small intestine, cell of skeletal muscle, schwann cell, artery endothelial cell, lymphatic endothelial cell, kidney epithelial cell, immature enterocyte, intestinal crypt stem cell, mature enterocyte, goblet cell, lung ciliated cell, respiratory goblet cell, serous cell of epithelium of bronchus, capillary aerocyte, type i pneumocyte, alveolar fibroblast, bronchial smooth muscle cell, club cell, lung microvascular endothelial cell, pulmonary ionocyte, pancreatic pp cell, pancreatic alpha cell, pancreatic delta cell, duodenum glandular cell, connective tissue cell]

- **Analysis:** We calculate the embedding similarities of a marker (orange in **Response Figure 4**) or housekeeping (gray in **Response Figure 4**) gene's contextualized protein representation (from PINNACLE) across different cell type contexts. For each marker/housekeeping gene, its cell type-specific protein representations are compared in similar contexts (i.e., between different T lymphocyte cell types) or different contexts (i.e., between a T lymphocyte cell type and a non-immune cell type).

We observe that all comparisons between the four groups shown are statistically significant (**Response Figure 4** in this document / Supplementary Figure S7 in the revised manuscript).

We note the following key results:

- Housekeeping genes in similar contexts have higher embedding similarity than marker genes in similar contexts (two-sample KS test, p-value = 3.2×10^{-14}). This is expected, as housekeeping genes have shared functions across these cell types. Because housekeeping genes have shared functions across non-immune cell types as well, housekeeping genes in different contexts expectedly have higher embedding similarity than marker genes in different contexts (two-sample KS test, p-value = 1.0×10^{-91}).
- Marker genes in similar contexts have higher embedding similarity than marker genes in different contexts (two-sample KS test, p-value = 3.1×10^{-26}). Because marker genes are specific to the T lymphocyte cell types, their protein representations should be more similar than comparing marker genes in the context of T lymphocyte cell types to marker genes in the context of non-immune cell types.

Given that PINNACLE is not optimized such that the contextualized representations of proteins with the same protein identity/function are more similar, this is an exciting finding. We expect

that including an additional term in PINNACLE's objective function to optimize for this would yield even stronger results.

5. What are the properties of each cell-type specific PPI graph and cell-cell, tissue-tissue graph?

The variations of the number of nodes are large and should affect how dense the graph is. What is the variation in the graph degrees and how does it affect the protein embedding?

The threshold for cell-cell connection seems low (at least one shared ligand-receptor pair); if the connection is sparser after setting up a high bar for connection, does it affect the performance?

How does the cell-type specific network handle a node without an edge?

Thank you for these important questions!

First, we provide histograms (**Response Figures 5 and 6** in this document / Supplementary Figure S1 in the revised manuscript) to show the properties of each cell type-specific PPI, cell type-cell type, tissue-tissue, and metagraph networks:

- **Response Figure 5** shows the degree distributions of the metagraph (consisting of the cell type-cell type and tissue-tissue graphs, and further cell type-tissue edges), and the cell type-cell type and tissue-tissue graphs. The median, maximum, and minimum degrees for the metagraph are 24, 169, 1; for the tissue-tissue graph, are 2, 15, 1; and for the cell type-cell type graph, are 24, 157, 4.
- **Response Figure 6** shows the distribution of the median node degree of each cell type-specific PPI network. The median, maximum, and minimum of median node degree across cell type-specific PPI networks are 6, 11, and 3, respectively.

Evaluating the effects of node degree on protein representations is not a straightforward task. So, we investigate the effect of node degree on our downstream prediction task, nominating therapeutic targets for RA/IBD (**Response Figure 7** in this document / Supplementary Figure S12 in the revised manuscript).

- We expect that performance is not influenced by the amount of information content. In other words, there is no correlation between the degree of a protein node and the performance of the protein representation.
- **Response Figure 7** shows no significant correlation between degree and performance. We bin the protein degrees (x-axis) and synthesize their performance (y-axis) by calculating the fraction of correct predictions (for both positive and negative proteins). We compute Spearman correlation on the binned protein degrees and performance. Combining the RA and IBD prediction results, the Spearman $\rho = 0.087$ with p -value = 0.223. For RA only, the Spearman $\rho = 0.205$ with p -value = 0.041. For IBD only, the Spearman $\rho = 0.024$ with p -value = 0.810.

Yes, at least one shared ligand-receptor pair is a low threshold. We set this threshold because we subsample cells to make CellPhoneDB more computationally feasible. As recommended by CellPhoneDB (documentation), cells are subsampled prior to running the algorithm, which uses geometric sketching (Hie et al. 2019) to efficiently sample a small representative subset of cells from massive datasets while preserving biological complexity. For 100 iterations, we subsample 25% of cells from our dataset and run CellPhoneDB.

- **Response Table 5** (Supplementary Table S3 in the revised manuscript) shows the data statistics of the metagraph with different cutoffs for the minimum number of significant ligand-receptor interactions (p -value < 0.001) between a pair of cell types to create an edge.
- We evaluate the models' ability to capture cell type and tissue organization in the embedding space (**Response Table 6** in this document / Supplementary Table S4 in the revised manuscript):
 - Spearman's correlation between tissue embedding distance (computed using the model's tissue representations) and tissue ontology distance. In a well-performing model, we expect a positive correlation.
 - Spearman's correlation between tissue embedding distance and fraction of overlapping cell types. In a well-performing model, we expect a strong negative correlation (see Response Figure 1 for more details).
- Our results (**Response Table 6**) suggest that higher cutoffs, which yield sparser CCI edges in the metagraph, do not affect the ability of PINNACLE's contextualized representations to capture cell type and tissue organization. It is also worth noting that we did not perform a hyperparameter sweep on these modified metagraphs; the hyperparameters are optimized for our original metagraph. Moving forward, we suggest potentially including the cutoff value as a hyperparameter to optimize for the user's desired utility and characteristics.

Finally, to answer your last question, we take the largest connected component (LCC) for our cell type-specific PPI networks and the metagraph, a standard graph pre-processing step in network science. As a result, none of the networks have a node without an edge.

6. The score-gap between binding and non-binding proteins should be compared with another baseline, such as the BIONIC and GAT models. For the GAT model, is it trained with the same self-supervised link prediction loss? If not, then the results of Fig. 3f and Fig. 4d-e should also be compared with the GAT model trained with link prediction loss.

We thank you for your great question and suggestion.

First, we want to clarify that Figure 3f does not require any fine-tuning. We concatenate PINNACLE's contextualized protein representations with the 3D structure-based protein representations and directly calculate their cosine similarities. On the other hand, Figure 4 does require fine-tuning of PINNACLE's protein representations to nominate therapeutic targets for RA/IBD. We benchmark these results against context-free approaches, such as GAT and

BIONIC. For GAT, similar to PINNACLE, the protein representations are pretrained using a self-supervised link prediction loss and finetuned for the downstream therapeutic task.

Next, as per your suggestion, we benchmark our score-gap analysis between binding and non-binding proteins against four context-free approaches (**Response Figure 3**):

- **GAT only**: Context-free protein representations generated by a graph attention neural network pretrained via self-supervised link prediction loss on the global reference interactome. The score gap between binding and non-binding proteins is -0.1319 (permutation statistical test is statistically insignificant).
- **GAT + 3D structure**: Concatenate GAT protein representations with 3D structure-based protein representations. The score gap between binding and non-binding proteins is -0.0486 (permutation statistical test is statistically insignificant).
- **BIONIC only**: Context-free protein representations pretrained via self-supervised link prediction loss on the cell type-specific protein interaction networks by BIONIC, a graph convolutional neural network designed for multi-modal network integration. The score gap between binding and non-binding proteins is 0.0046 (permutation statistical test is statistically insignificant).
- **BIONIC + 3D structure**: Concatenate BIONIC protein representations with 3D structure-based protein representations. The score gap between binding and non-binding proteins is 0.0043 (permutation statistical test is statistically insignificant).

These results indicate that context-free protein representations cannot predict intercellular communication (i.e., protein interactions between different cell types).

7. The score defined for binding and non-binding proteins is cosine similarity. Is it a good metric to measure protein embedding quality for binding and non-binding proteins? Since the message-passing method makes protein embeddings that frequently connect with each other more similar, that gap is enforced by the model.

MaSIF tried to measure prediction accuracy, so if contextualized protein representation brings the performance increase to a binding prediction accuracy then it would prove that the embedding quality of the protein is good.

Thank you for your question! To clarify, we are not measuring the protein embedding quality of the two types of proteins using cosine similarity. We are using cosine similarity to measure and compare the similarity of the embeddings of binding and non-binding proteins. MaSIF is a deep learning approach that generates protein representations to capture (and identify) the interacting components of a pair of proteins. It is trained such that proteins that interact have more similar embeddings (using metrics like cosine similarity) than those that do not interact. As such, we use cosine similarity as the performance metric in our analysis.

Note that we are unable to use MaSIF's pretrained protein-protein interaction prediction module because of two reasons:

- **Incompatible input dimensions of the prediction module**: The dimensions of the concatenated PINNACLE and MaSIF representations exceed the input dimensions of

the prediction module. Using other aggregation approaches (e.g., taking the sum or average) would dilute, or even eliminate, any signal from MaSIF and PINNACLE.

- Small sample size of inter-cellular protein interactions with 3D structure: No large-scale dataset with *matched structural biology and genomic readouts* exists to perform systematic analyses (reference human PPI network does not contain cell type specific information). Concretely, there does not exist a large systematic dataset with information on cell type specificity of protein-protein interactions that would also contain 3D structure information, meaning that it is not possible to define a cell type specific PPI prediction task as a benchmarking task, for which either MaSIF or PINNACLE could be fine-tuned. Because of that, we focus on PD-1/PD-L1 interacting proteins and B7-1/CTLA-4 interacting proteins, which are important interactions taking place in the immune checkpoint context. These cell type specific PPIs are critical in cancer immunotherapies, which are the pillars of treatments in modern oncology.

Our findings in this case study on immuno-oncology interactions suggest that incorporating context can improve 3D structure prediction of protein interactions, which has the potential to deepen our understanding of how these four proteins are used in cancer immunotherapies.

8. What is the exact definition of the predictiveness of protein representation? The output of the MLP is the prediction for the supervised target and does not represent the confidence score of the model. In our understanding, it's better to measure the uncertainty of the model using conformal prediction techniques to statistically measure the predictiveness of the model. The authors should at least show if the range of the output of MLP is well aligned with the predictiveness of the model.

Thank you for raising these excellent points!

- We recognize that our usage of "predictiveness" is inconsistent and misleading. We have corrected our language throughout the paper to reflect the intended usage. There is no notion of "predictiveness" of protein representations. Rather, we use "predictiveness" to refer to how predictive each cell type context is towards distinguishing RA (IBD) protein targets vs. non-RA (IBD) protein targets (Figure 5a,d). In other words, the "predictiveness" of a cell type is defined as the ability to identify candidate therapeutic target(s) for RA (Figure 5a) or IBD (Figure 5d) using the contextualized protein embeddings from that cell type.
- You are correct that predictive scores do not necessarily capture uncertainty. The downstream model outputs/predictions range between 0 and 1 after being passed through a nonlinear activation layer. As such, they cannot be interpreted as measures of uncertainty. This issue of uncalibrated scores plagues deep learning models in general. Recent techniques on conformal prediction and evidential layers can be used with any graph neural network (Huang et al. Uncertainty Quantification over Graph with Conformalized Graph Neural Networks, NeurIPS 2023), including PINNACLE. We discuss these limitations and future directions in the penultimate paragraph of our Discussion section. For your convenience, here is the added text:

Lastly, to move towards a "lab-in-the-loop" framework, where computational and experimental scientists can iteratively refine the machine learning model and

validate hypotheses via experiments, recent techniques on conformal prediction (Huang et al. 2023) and evidential layers can be integrated with PINNACLE to quantify uncertainty of model outputs.

9. All the contexts for RA and IBD in Fig. 5a-b are cell types. Is a complex representation required to obtain these cell type lists? Similarly, for Fig. 5b-f, if we measure the gene set enrichment score that includes JAK3, IL6R, or others, are we able to obtain the same cell type lists?

Thank you very much for the insightful suggestions! To answer your questions:

- We correlate PINNACLE's performance (measured by APR@5) and ENRICHMENT's performance (calculated as the ratio of positive to negative proteins) in each cell type for nominating RA and IBD therapeutic targets (**Response Figures 8 and 9** in this document / Supplementary Figure S13 in the revised manuscript). We observe a positive correlation between PINNACLE's performance and ENRICHMENT's performance for RA (Spearman $\rho = 0.53$; p -value = 8.7×10^{-13}) and IBD (Spearman $\rho = 0.54$; p -value = 8.5×10^{-13}).
- We additionally correlate the rankings produced by PINNACLE and ENRICHMENT for RA and IBD (**Response Tables 7 and 8**). For tied rankings by ENRICHMENT, we calculate and assign the average rank. We find that the ranked lists produced by PINNACLE and ENRICHMENT are not significantly correlated for RA (Spearman $\rho = 0.22$; p -value = 0.11) and moderately correlated for IBD (Spearman $\rho = 0.44$; p -value $< 10^{-4}$).

As expected, there is a positive correlation between PINNACLE's predictions and ENRICHMENT, a gene set enrichment score calculated based on a differential expression analysis of positive and negative proteins in each cell type. Still, a substantial amount of information produced by PINNACLE cannot be explained by enrichment analysis alone. Furthermore, PINNACLE's contextualized protein representations enable a wide range of other analyses above and beyond gene set enrichment, as demonstrated throughout the manuscript.

10. Do all results in Fig 4d-e and Fig 5. a-f. come from test datasets?

Thank you for your question!

- Figure 4d-e shows benchmarking results that compare our model to baseline models. We use the same train/test data splits to ensure a fair comparison. The results in Figure 4d-e are generated using the test set. The results are comparable since the test set is consistent across all models.
- Similar to Figure 4d-e, Figure 5a,d show results generated from the test dataset split.
- Figure 5b-c, e-f are case studies of proteins in the validation or test dataset splits.

Minor Points

1. The authors seem to use GAT for each context level. It would be important to clarify this point to ensure the reader understands the model.

Thank you for your suggestion. We have included a clarification note in the Methods sections 3.1 (“Protein-level attention with cell type specificity”) and 3.2 (“Metagraph-level attention on cellular interactions and tissue hierarchy”). For your convenience, here is the added sentence: “Practically, we leverage the attention function in graph attention neural networks (i.e., GATv2) (Brody et al. 2022).”

Response Figures

Response Figure 1: Comparing the cosine distance of PINNACLE's tissue representations and the fraction of overlapping cell types neighbors between the tissue pair. Spearman $\rho = -0.46$ with p-value = 8.01×10^{-30} .

Response Figure 2: Comparing PINNACLE's tissue embedding distance to tissue ontology distance for leaf nodes in the metagraph. Spearman $\rho = 0.11$ with p-value = 0.01.

Response Figure 3: Shown are binding and non-binding scores (i.e., cosine similarity) of proteins when using only 3D structure-based protein representations, PINNACLE's contextualized protein representations (without 3D structural information; p-value = 0.0299), contextualized structure-based protein representations (p-value $< 10^{-5}$), and baseline models. The baseline models are: random context only (i.e., randomly sampling pairs of PINNACLE's protein representations from different cell type contexts), concatenating random context protein representations with 3D structure-based protein representations, GAT only (i.e., context-free protein representations generated by a graph attention neural network on the global reference interactome), concatenating GAT protein representations with 3D structure-based protein representations, BIONIC only (i.e., context-free protein representations generated by BIONIC, a graph convolutional neural network designed for multi-modal network integration), and concatenating BIONIC protein representations with 3D structure-based protein representations. Note that all protein representations have consistent dimensions (328 = 200 structure-based protein representation + 128 context-aware/-free protein representation) to ensure that they are comparable. The protein representations without 3D structure are padded with 0's (i.e., null 3D structure-based protein representation).

Response Figure 4: Comparison of embedding similarities of a marker (orange) or housekeeping (gray) gene's contextualized protein representation (from PINNACLE) across different cell type contexts. The marker genes are specific to cell types in the family of T lymphocytes (a total of 10 T lymphocyte cell types). For each marker/housekeeping gene, its cell type-specific protein representations are compared in similar contexts (i.e., between different T lymphocyte cell types) or different contexts (i.e., between a T lymphocyte cell type and a non-immune cell type; a total of 115 non-immune cell types). All comparisons between these four groups shown are statistically significant. Cosine embedding similarity is used to compare contextualized protein representations.

Response Figure 5: Degree distributions of the metagraph (composed of cell type-cell type, cell type-tissue, and tissue-tissue edges), tissue-tissue graph, and cell type-cell type graph. The median, maximum, and minimum degrees for the metagraph are 24, 169, 1; for the tissue-tissue graph are 2, 15, 1; and for the cell type-cell type graph are 24, 157, 4.

Response Figure 6: Distribution of the median node degree of each cell type-specific PPI network. The median, maximum, and minimum of median node degree across cell type-specific PPI networks are 6, 11, and 3, respectively.

Response Figure 7: Correlation between the node degrees of proteins (in the cell type-specific protein interaction networks) and the downstream performance of their learned representations. Combining the RA and IBD prediction results, the Spearman $\rho = 0.087$ with p -value = 0.223. For RA only, the Spearman $\rho = 0.205$ with p -value = 0.041. For IBD only, the Spearman $\rho = 0.024$ with p -value = 0.810.

Response Figure 8: Correlation between PINNACLE's predicted performance (APR@5) and the ratio of positive to negative proteins in each cell type for RA. Spearman $\rho = 0.53$ with $p\text{-value} = 8.7 \times 10^{-13}$.

Response Figure 9: Correlation between PINNACLE's predicted performance (APR@5) and the ratio of positive to negative proteins in each cell type for IBD. Spearman $\rho = 0.54$ with p -value = 8.5×10^{-13} .

Response Tables

Dataset	Type of protein target	Proteins in train dataset (Unique proteins in train dataset)	Proteins in validation dataset (Unique proteins in validation dataset)	Proteins in test dataset (Unique proteins in test dataset)
RA	Total	17,408 (600)	6,647 (195)	25,137 (818)
	Positive	1,319 (53)	570 (21)	2,226 (78)
	Negative	16,089 (547)	6,077 (174)	22,911 (740)
IBD	Total	27,652 (896)	9,363 (297)	8,864 (294)
	Positive	1,210 (62)	673 (26)	731 (26)
	Negative	26,442 (834)	8,690 (271)	8,133 (268)

Response Table 1: Sizes of the train, validation, and test datasets for the rheumatoid arthritis (RA) PINNACLE model and inflammatory bowel disease (IBD) PINNACLE model. The numeric value outside the parentheses represents the number of protein representations across cell type contexts, and the numeric value inside the parentheses represents the number of unique protein identities. The numbers represent both positive (label = 1) and negative (label = 0) proteins. Note that the validation dataset set is sampled from the train dataset, which is fixed, at each run of the model. The numbers for train and validation datasets (columns 3-4) are from seed 1.

Cell type	Cell Subtype	Train	Validation	Test
T cell	CD4+ helper T cell	50	18	75
	CD4+ $\alpha\beta$ memory T cell	51	19	62
	Regulatory T cell	78	34	105
	CD8+ $\alpha\beta$ cytotoxic T cell	65	22	98
	DN1 thymic pro-T cell	66	20	81
	Mature natural killer T cell	67	15	76
	Naive regulatory T cell	82	32	128
	Type I natural killer T cell	48	12	54
	Naive thymus-derived CD4+ $\alpha\beta$ T cell	50	19	60
	CD8+ $\alpha\beta$ cytokine secreting effector T cell	67	15	89
Dendritic cell	CD1c+ myeloid dendritic cell	143	53	211
	CD141+ myeloid dendritic cell	137	56	209
	Dendritic cell	108	35	155
	Mature conventional dendritic cell	113	37	173
	Myeloid dendritic cell	131	57	202
	Plasmacytoid dendritic cell	103	35	150
	Liver dendritic cell	134	47	197
B cell	Memory B cell	70	29	88
	B cell	48	16	56
Natural killer cell	Natural killer cell	72	17	84
	Immature natural killer cell	47	17	75
Monocyte	Intermediate monocytes	123	37	176
	Non-classical monocytes	80	25	116
	Monocyte	141	39	201
	Classical monocytes	147	45	209

Myeloid cell	Myeloid progenitor	124	51	204
	Myeloid cell	156	48	207

Response Table 2: Numbers of proteins in the train, validation, and test sets in the rheumatoid arthritis prediction task in representative cell types, according to the existing literature for being involved with rheumatoid arthritis. Note that the validation dataset set is sampled from the train dataset, which is fixed, at each run of the model. The numbers for train and validation datasets (columns 3-4) are from seed 1.

Cell type	Cell Subtype	Train	Validation	Test
T cell	CD4+ $\alpha\beta$ memory T cell	81	18	21
	Naive thymus-derived CD4+ $\alpha\beta$ T cell	76	20	22
	Regulatory T cell	112	38	32
	DN1 thymic pro-T cell	96	26	25
	Mature natural killer T cell	82	26	26
	CD8+ $\alpha\beta$ cytokine secreting effector T cell	87	36	36
	Type I natural killer T cell	53	18	22
	CD4+ helper T cell	75	21	26
	Naive regulatory T cell	121	43	40
	CD8+ $\alpha\beta$ cytotoxic T cell	97	31	35
Enterocyte	Enterocyte of epithelium of large intestine	88	36	33
	Mature enterocyte	124	43	36
	Immature enterocyte	255	87	75
	Enterocyte of epithelium of small intestine	155	61	58
	Intestinal enteroendocrine cell	193	77	76
Dendritic cell	Myeloid dendritic cell	235	72	60
	Dendritic cell	169	55	45
	CD1c+ myeloid dendritic cell	234	79	69
	CD141+ myeloid dendritic cell	230	74	66
	Liver dendritic cell	213	70	66
	Mature conventional dendritic cell	181	57	54
	Plasmacytoid dendritic cell	156	56	49
Goblet cell	Large intestine goblet cell	76	30	32
	Goblet cell	153	53	56
	Small intestine goblet cell	109	38	49

	Respiratory goblet cell	227	74	72
	Tracheal goblet cell	196	64	59
B cell	B cell	66	23	13
	Memory B cell	90	39	37
Monocyte	Intermediate monocyte	186	63	65
	Non-classical monocyte	132	42	34
	Monocyte	208	76	71
	Classical monocyte	219	80	75
Glial cell	Microglial cell	223	77	67
	Radial glial cell	255	81	86
Natural killer cell	Immature natural killer cell	63	27	25
	Natural killer cell	93	32	28
Fibroblast	Fibroblast	244	75	85
Macrophage	Macrophage	237	91	75

Response Table 3: Numbers of proteins in the train, validation, and test sets in the inflammatory bowel disease prediction task in representative cell types, according to the existing literature for being involved with inflammatory bowel diseases. Note that the validation dataset set is sampled from the train dataset, which is fixed, at each run of the model. The numbers for train and validation datasets (columns 3-4) are from seed 1.

Model	Tissue Embedding Distance vs. Tissue Ontology Distance	Tissue Embedding Distance vs. Tissue Ontology Distance (leaves only)	Tissue Embedding Distance vs. Fraction of Cell Type Overlap (leaves only)
Complete model	Spearman $\rho = 0.36$ p -value = 4.6×10^{-119}	Spearman $\rho = 0.11$ p -value = 0.01	Spearman $\rho = -0.46$ p -value = 8.01×10^{-30}
Drop cell type-cell type graph (i.e., shuffle cell type identity)	Spearman $\rho = 0.38$ p -value = 2.3×10^{-132}	Spearman $\rho = 0.10$ p -value = 0.02	Spearman $\rho = -0.21$ p -value = 4.25×10^{-7}
Drop tissue-tissue graph (i.e., shuffle tissue identity)	Spearman $\rho = -0.13$ p -value = 1.2×10^{-14}	Spearman $\rho = -0.15$ p -value = 6.5×10^{-4}	Spearman $\rho = -0.16$ p -value = 2.5×10^{-4}
Drop metagraph loss (i.e., $\theta = 1$)	Spearman $\rho = 0.30$ p -value = 4.6×10^{-79}	Spearman $\rho = -0.10$ p -value = 0.02	Spearman $\rho = -0.19$ p -value = 1.1×10^{-5}

Response Table 4: Ablation studies to interrogate the contribution of the metagraph. The first row consists of results from the complete model. The remaining three rows show results from three types of ablations: removing cell type to cell type relationships (i.e., shuffling the cell type nodes' identities), removing tissue-to-tissue relationships (i.e., shuffling the tissue nodes' identities), and removing the metagraph (i.e., setting the weight of the metagraph-related terms in the loss function to zero). The performance metrics evaluate the models' ability to capture cell type and tissue organization in the embedding space. The second column is the correlation between tissue embedding distance (computed using the model's tissue representations) and tissue ontology distance; we expect a positive correlation. The third column is the correlation between tissue embedding distance and tissue ontology distance among the tissue leaf nodes of the metagraph; we expect a positive correlation (see Response Figures 1 and 2 for more details). The fourth column is the correlation between tissue embedding distance and fraction of overlapping cell types; we expect a strong negative correlation (see Response Figures 1 and 2 for more details).

Cutoff for Significant LRs	Component of Metagraph	Number of Nodes	Number of Edges	Average Degree
Cutoff = 1 (Original)	CCI Graph	156	3,567	45.7
	Metagraph (All)	218	4,018	36.9
Cutoff = 2	CCI Graph	156	1,808	23.2
	Metagraph (All)	218	2,259	20.7
Cutoff = 3	CCI Graph	156	1,736	22.3
	Metagraph (All)	218	2,187	20.1
Cutoff = 4	CCI Graph	156	1,640	21.0
	Metagraph (All)	218	2,091	19.2
Cutoff = 5	CCI Graph	156	1,576	20.2
	Metagraph (All)	218	2,027	18.6

Response Table 5: Data statistics of the metagraph with different cutoffs for the minimum number of significant ligand-receptor interactions (p -value < 0.001) between a pair of cell types to create an edge. Shown are the number of nodes, number of edges, and the average degree of the cell type-cell type interaction (CCI) graph and the metagraph (includes cell type-cell type, cell type-tissue, and tissue-tissue edges).

Model	Tissue Embedding Distance vs. Tissue Ontology Distance	Tissue Embedding Distance vs. Fraction of Cell Type Overlap (leaves only)
Complete model (Cutoff = 1)	Spearman $\rho = 0.36$ p -value = 4.6×10^{-119}	Spearman $\rho = -0.46$ p -value = 8.01×10^{-30}
Cutoff = 2	Spearman $\rho = 0.21$ p -value = 1.0×10^{-37}	Spearman $\rho = -0.31$ p -value = 2.4×10^{-13}
Cutoff = 3	Spearman $\rho = 0.22$ p -value = 1.7×10^{-41}	Spearman $\rho = -0.31$ p -value = 7.8×10^{-14}
Cutoff = 4	Spearman $\rho = 0.25$ p -value = 1.4×10^{-53}	Spearman $\rho = -0.29$ p -value = 2.3×10^{-12}
Cutoff = 5	Spearman $\rho = 0.38$ p -value = 8.8×10^{-129}	Spearman $\rho = -0.25$ p -value = 3.7×10^{-9}

Response Table 6: Sensitivity analysis to examine the impact of the cutoff value for the minimum required number of significant ligand-receptor interactions (p -value < 0.001) in the cell-type-to-cell-type graph on PINNACLE's embedding space. The first row consists of results from the complete model. The remaining four rows show results from cutoff values 2, 3, 4, and 5. Refer to Response Table 5 for data statistics about the resulting metagraph (i.e., cell-type-to-cell-type and tissue-to-tissue edges).

Cell type	Cell Subtype	PINNACLE Rank	ENRICHMENT Rank
T cell	CD4+ helper T cell	1	3
	CD4+ $\alpha\beta$ memory T cell	2	6
	Regulatory T cell	6	14
	CD8+ $\alpha\beta$ cytotoxic T cell	27	8
	DN1 thymic pro-T cell	33	5
	Mature natural killer T cell	34	4
	Naive regulatory T cell	39	11
	Type I natural killer T cell	49	1
	Naive thymus-derived CD4+ $\alpha\beta$ T cell	77	12.5
	CD8+ $\alpha\beta$ cytokine secreting effector T cell	104	9
Dendritic cell	CD1c+ myeloid dendritic cell	3	39
	CD141+ myeloid dendritic cell	11	31
	Dendritic cell	37	10
	Mature conventional dendritic cell	38	26
	Myeloid dendritic cell	42	71
	Plasmacytoid dendritic cell	43	22
	Liver dendritic cell	50	24
B cell	Memory B cell	12	16
	B cell	29	2
Natural killer cell	Natural killer cell	17	7
	Immature natural killer cell	113	19
Monocyte	Intermediate monocytes	18	25
	Non-classical monocytes	68	18
	Monocyte	85	49

	Classical monocytes	91	43
Myeloid cell	Myeloid progenitor	71	54
	Myeloid cell	95	41

Response Table 7: Rankings of selected cell types according to PINNACLE and ENRICHMENT for rheumatoid arthritis. Shown are predicted ranks of subtypes of T cells, natural killer (NK) cells, dendritic cells, B cells, monocytes, and myeloid cells. These cell types have been demonstrated in existing literature to be involved with rheumatoid arthritis, as described in Supplementary Table S1.

Cell type	Cell Subtype	PINNACLE Rank	ENRICHMENT Rank
T cell	CD4+ $\alpha\beta$ memory T cell	1	2
	Naive thymus-derived CD4+ $\alpha\beta$ T cell	11	5
	Regulatory T cell	14	12
	DN1 thymic pro-T cell	15	1
	Mature natural killer T cell	16	9.5
	CD8+ $\alpha\beta$ cytokine secreting effector T cell	19	16
	Type I natural killer T cell	30	3
	CD4+ helper T cell	41	7
	Naive regulatory T cell	43	18
	CD8+ $\alpha\beta$ cytotoxic T cell	56	11
Enterocyte	Enterocyte of epithelium of large intestine	2	36.5
	Mature enterocyte	22	41.5
	Immature enterocyte	90	90.5
	Enterocyte of epithelium of small intestine	94	70
	Intestinal enteroendocrine cell	138	79.5
Dendritic cell	Myeloid dendritic cell	5	75
	Dendritic cell	10	17
	CD1c+ myeloid dendritic cell	13	31
	CD141+ myeloid dendritic cell	50	25.5
	Liver dendritic cell	54	36.5
	Mature conventional dendritic cell	67	32
	Plasmacytoid dendritic cell	72	52
Goblet cell	Large intestine goblet cell	7	33
	Goblet cell	33	67

	Small intestine goblet cell	75	35
	Respiratory goblet cell	109	145
	Tracheal goblet cell	110	148
B cell	B cell	9	9.5
	Memory B cell	45	43
Monocyte	Intermediate monocyte	26	34
	Non-classical monocyte	81	14
	Monocyte	89	74
	Classical monocyte	93	45.5
Glial cell	Microglial cell	29	19
	Radial glial cell	107	76
Natural killer cell	Immature natural killer cell	31	8
	Natural killer cell	73	22.5
Fibroblast	Fibroblast	49	63
Macrophage	Macrophage	74	45.5

Response Table 8: Rankings of selected cell types according to PINNACLE and ENRICHMENT for inflammatory bowel diseases. Shown are predicted ranks of subtypes of T cell, fibroblast, goblet cell, enterocyte, monocyte, natural killer cell, B cell, glial cell, dendritic cell, and macrophage. These cell types have been demonstrated in existing literature to be involved with inflammatory bowel diseases, as described in Supplementary Table S2.

Response References

- Braberg, H., Echeverria, I., Kaake, R. M., Sali, A., & Krogan, N. J. (2022). From systems to structure—using genetic data to model protein structures. *Nature Reviews Genetics*, 23(6), 342-354.
- Hu, W., Liu, B., Gomes, J., Zitnik, M., Liang, P., Pande, V., & Leskovec, J. (2020). Strategies for Pre-training Graph Neural Networks. *International Conference on Learning Representations (ICLR)*.
- Houkpe, B. W., Chenou, F., de Lima, F., & De Paula, E. V. (2021). HRT Atlas v1. 0 database: redefining human and mouse housekeeping genes and candidate reference transcripts by mining massive RNA-seq datasets. *Nucleic Acids Research*, 49(D1), D947-D955.
- Greene, C. S., Krishnan, A., Wong, A. K., Ricciotti, E., Zelaya, R. A., Himmelstein, D. S., ... & Troyanskaya, O. G. (2015). Understanding multicellular function and disease with human tissue-specific networks. *Nature Genetics*, 47(6), 569-576.
- Hie, B., Cho, H., DeMeo, B., Bryson, B., & Berger, B. (2019). Geometric sketching compactly summarizes the single-cell transcriptomic landscape. *Cell Systems*, 8(6), 483-493.
- Huang, K., Jin, Y., Candes, E., & Leskovec, J. (2023). Uncertainty Quantification over Graph with Conformalized Graph Neural Networks. *Proceedings of the 37th Conference on Neural Information Processing Systems (NeurIPS)*.
- Brody, S., Alon, U., & Yahav, E. (2022). How attentive are graph attention networks?. *ICLR*.

Decision Letter, first revision:

Dear Marinka,

Your Article, "Contextualizing protein representations using deep learning on protein networks and single-cell data", has now been seen by 2 reviewers. As you will see from their comments below, although both the reviewers find your work of considerable potential interest, Reviewer #2 has some remaining concerns. We are interested in the possibility of publishing your paper in Nature Methods, but would like to consider your response to these concerns before we reach a final decision on publication.

We therefore invite you to revise your manuscript to address these concerns.

[Redacted]

We hope to receive your revised paper within 4 weeks. If you cannot send it within this time, please let us know. In this event, we will still be happy to reconsider your paper at a later date so long as nothing similar has been accepted for publication at Nature Methods or published elsewhere.

OPEN SCIENCE REQUIREMENTS

REPORTING SUMMARY AND EDITORIAL POLICY CHECKLISTS

IMAGE INTEGRITY

When submitting the revised version of your manuscript, please pay close attention to our Digital Image Integrity Guidelines and to the following points below:

DATA AVAILABILITY

All novel DNA and RNA sequencing data, protein sequences, genetic polymorphisms, linked genotype and phenotype data, gene expression data, macromolecular structures, and proteomics data must be deposited in a publicly accessible database, and accession codes and associated hyperlinks must be provided in the "Data Availability" section.

CODE AVAILABILITY

Please include a "Code Availability" subsection in the Online Methods which details how your custom code is made available. Only in rare cases (where code is not central to the main conclusions of the paper) is the statement "available upon request" allowed (and reasons should be specified).

For more information on our code sharing policy and requirements, please see: <https://www.nature.com/nature-research/editorial-policies/reporting-standards#availability-of-computer-code>

MATERIALS AVAILABILITY

SUPPLEMENTARY PROTOCOL

To help facilitate reproducibility and uptake of your method, we ask you to prepare a step-by-step Supplementary Protocol for the method described in this paper. We encourage authors to share their step-by-step experimental protocols on a protocol sharing platform of their choice and report the protocol DOI in the reference list. Nature Portfolio 's Protocol Exchange is a free-to-use and open resource for protocols; protocols deposited in Protocol Exchange are citable and can be linked from the published article. More details can found at www.nature.com/protocolexchange/about.

ORCID

Nature Methods is committed to improving transparency in authorship. As part of our efforts in this direction, we are now requesting that all authors identified as 'corresponding author' on published papers create and link their Open Researcher and Contributor Identifier (ORCID) with their account on the Manuscript Tracking System (MTS), prior to acceptance. This applies to primary research papers only. ORCID helps the scientific community achieve unambiguous attribution of all scholarly contributions. You can create and link your ORCID from the home page of the MTS by clicking on 'Modify my Springer Nature account'. For more information please visit please visit www.springernature.com/orcid.

Sincerely,
Arunima

Arunima Singh, Ph.D.
Senior Editor
Nature Methods

Reviewers' Comments:

Reviewer #1:

Remarks to the Author:

The authors have done a fantastic job addressing our earlier concerns in this significantly revised manuscript. No further comments.

-Clara Hu and Trey Ideker

Reviewer #2:

Remarks to the Author:

We thank the authors for their thorough revision and responsiveness to our and Reviewer 1's comments. The authors have conducted additional experiments and added more data to answer the questions, address the suggestions and resolve the concerns raised by the reviewers.

However, there are a few key items that still require resolution prior to publication. We note that we remain enthusiastic about the paper and the authors' ability to address the remaining comments.

Major comments

1. To understand the training objective, the loss function is critical. Even though the authors clarify the model is trained in a self-supervised manner using masked link prediction loss, it's tough to recognize how it is used ****in the loss function****. For example, if message-passing only occurred between edge-unmasked nodes and then the loss function is calculated to predict the edge-masked nodes' attributes using the unmasked nodes', then it should be clarified on the loss function. Also, please clarify the ratio of nodes or edges that are masked during the training. Overall, these key details are crucial to understanding the inductive bias of the model, they ***must*** be clarified in the loss function in both the paper text and the code prior to publication. (We checked the code.)

2. While the model is stated as (and started from) a protein embedding, all the analysis was done at the cell level and cell data (including expression) is very dominant. The final set of results highlights selecting a 'druggable' (target) cell type based on the 'contextualized protein embedding'. This makes us wonder how much of the ultimate results is driven by cell type embedding rather than protein embedding. (That is, this is a meaningful contextualized protein embedding, but the embedding can't be used as many downstream tasks unless the cell type is important.). We fully realize that the quality of protein embedding per se it's hard to quantify (a similar situation occurs with gene embeddings from single-cell foundation models). We ask that the authors address this in the Discussion section.

In particular, the embedding similarities of marker/housekeep genes between similar and different contexts are quite interesting! We thank the authors for following our suggestion on this topic. Although the authors claim that the model learns meaningful protein embeddings, it's hard to conclude this based on Response Figure 4. Why is the similarity of housekeeping genes higher than that of marker genes? Why is there a significant drop in the similarity of housekeeping genes between different contexts? We know it's hard to measure the quality of representation learning, but, ideally, both housekeeping genes of similar contexts and different contexts would have comparably high similarities (or there would be a biological argument that their behavior is not "housekeeping"), and marker genes would have larger gaps than housekeeping genes. As the authors noted, PINNACLE is more focused on cell-type contextualized protein embedding, such that although PINNACLE learns a meaningful protein embedding that has more correlation between functionally similar cell types, it will be biased against genes that are unaffected by cell type specificity. Such biases are also shown in Figure S4, where the protein embedding space appears more constructed along cell type expression.

Do the authors observe the embedding space of the same genes in different cell types? These points can be addressed in the Discussion.

Minor comments

1. The authors explain their "self-supervised training" as follows: "PINNACLE is trained in a self-supervised manner using cell type identity and graph connectivity (i.e., cell type-specific protein interaction networks and metagraph) as a supervised signal to define positive vs. negative links in self-supervised link prediction."

We find this confusing (and think other readers will also be confused on whether this is supervised learning or not). We found this closer to knowledge-guided training.

The goal of the training is representation learning that makes a contextualized protein embedding for each cell type. These properties don't come naturally through training. In particular, the loss function includes a "center loss", which uses the cell-type label during training to make proteins from the same cell more similar to each other. It might be better to not categorize PINNACLE as self-supervised training and just explain the loss function.

Author Rebuttal, first revision:

Response to Reviewers

Contextualizing protein representations using deep learning on protein networks and single-cell data

Michelle M. Li¹, Yepeng Huang¹, Marissa Sumathipala¹, Man Qing Liang¹, Alberto Valdeolivas², Ashwin N. Ananthakrishnan^{1,3}, Katherine Liao^{1,4}, Daniel Marbach², and Marinka Zitnik^{1,5,6,7}

¹Harvard Medical School, Boston, MA, USA; ²Roche Pharma Research and Early Development, Pharmaceutical Sciences, Roche Innovation Center Basel, Basel, Switzerland; ³Division of Gastroenterology, Massachusetts General Hospital, Boston, MA, USA; ⁴Division of Rheumatology, Inflammation, and Immunity, Brigham and Women's Hospital, Boston, MA, USA; ⁵Kempner Institute for the Study of Natural and Artificial Intelligence, Harvard University, MA, USA; ⁶Broad Institute of MIT and Harvard, Cambridge, MA, USA; ⁷Harvard Data Science Initiative, Cambridge, MA, USA.

We thank the Reviewers for their valuable feedback and positive evaluation of this article. In response to the Reviewers' feedback, we have substantially changed the manuscript to improve the clarity and address the limitations of our analyses. **In addition to revising the main and supplementary manuscript and answering the Reviewers' queries, we include one new Response Table in this document.**

Our response is structured as follows:

- **Response to Reviewer 1 (page 3)**
- **Response to Reviewer 2 (page 4)**
- **Response Tables (page 12)**
- **Response References (page 13)**

Comments from Reviewers are in boxes, and our responses to each comment follow. Changes are marked in blue in the revised manuscript.

We strengthen our Introduction and Methods sections with more precise language about our loss function. Specifically, as requested by the Reviewers, we clarify the descriptions of the training strategy for PINNACLE, including details about the setup of the link prediction task and the sampling of negative edges. We have also revised our codebase so that the code modules implementing the loss functions discussed in the Methods section are easily identifiable and searchable.

Additionally, we have revised the figures with results on SAFE (Spatial Analysis of Functional Enrichment) and downstream therapeutic target identification task to make it clear that analyses are centered on protein-level representations and that they evaluate protein representations as opposed to cell-level representations. As requested by the Reviewers, we address in the Discussion section the possibility of extending PINNACLE for learning additional representations of our data, such as cell-level representations, to enable even more expansive capabilities.

Finally, we clarify the extensive analyses performed in the manuscript to evaluate PINNACLE's embedding space. We discuss the limitations of our analysis on proteins with shared functions/roles. The biological questions inspired by our preliminary analyses warrant further investigation with collaborations across domains and expertise. It is clear that there are many exciting opportunities sparked by our manuscript, and we look forward to continuing our exploration in future follow-up work.

We believe that this revision, which carefully addresses the Reviewers' valuable feedback, has confirmed the potential of context-specific AI models for biological analyses and laid solid foundations that can stimulate many exciting directions of future research.

Response to Reviewer #1

Remarks to the Author:

The authors have done a fantastic job addressing our earlier concerns in this significantly revised manuscript. No further comments.

-Clara Hu and Trey Ideker

We thank the Reviewer for recognizing our efforts to address the concerns previously raised. Again, we are very grateful for your valuable feedback that has improved our manuscript.

Response to Reviewer #2

Remarks to the Author:

We thank the authors for their thorough revision and responsiveness to our and Reviewer 1's comments. The authors have conducted additional experiments and added more data to answer the questions, address the suggestions and resolve the concerns raised by the reviewers.

However, there are a few key items that still require resolution prior to publication. We note that we remain enthusiastic about the paper and the authors' ability to address the remaining comments.

We thank the Reviewer for recognizing our efforts to address both reviewers' comments and providing additional valuable feedback to further improve this manuscript. We thoroughly address each of the Reviewer's comments in a detailed point-by-point response below.

Major comments

1. To understand the training objective, the loss function is critical. Even though the authors clarify the model is trained in a self-supervised manner using **masked link prediction loss**, it's tough to recognize how it is used ****in the loss function****. For example, if message-passing only occurred between edge-unmasked nodes and then the loss function is calculated to predict the edge-masked nodes' attributes using the unmasked nodes', then it should be clarified on the loss function. **Also, please clarify the ratio of nodes or edges that are masked during the training.** Overall, these key details are crucial to understanding the inductive bias of the model, they ***must*** be clarified in the loss function in both the paper text and the code prior to publication. (We checked the code.)

We thank the Reviewer for raising these great points. We apologize for the confusion regarding our loss function. To address the comments, we have revised our Results section, Methods section, and our codebase:

- In the Results section, we clarify the language about the protein-level pretraining tasks so that we use a more precise word than "mask." **For your convenience, we provide the revised sentence below** (edit in **blue** text):
The protein-level pretraining tasks consider self-supervised link prediction on protein interactions and cell type **classification masking** on protein nodes.
- We have provided more details about link prediction in Section 3.5 ("Training details for PINNACLE") of the Methods section. To clarify, link prediction is a task in which the model predicts whether or not an edge exists between a pair of nodes; this is an established self-supervised learning strategy for graph neural networks (Hu et al. *ICLR*

2020). We do not utilize or predict attributes about individual nodes/edges in our link prediction setup. Practically, link prediction means that the model takes as *input* a pair of nodes and *outputs* the value 1 if there is an edge between them or 0 otherwise. We can additionally consider edge type information; as in, given a pair of nodes and a candidate edge type r , predict whether edge type r exists between them (binary classification). **For your convenience, we have included the text in Methods (section 3.5 “Training details for PINNACLE”) below:**

PINNACLE is trained using cell type identity of the protein interaction networks and graph connectivity of the cell type specific protein interaction networks and metagraph. To learn cell type identity, PINNACLE predicts the cell type(s) that the node(s) corresponding to each protein are activated in. For capturing graph connectivity, PINNACLE performs self-supervised link prediction; it predicts whether an edge (and its type) exists between a pair of nodes.

- We have added details about how edges are split into train, validation, and test sets in Section 3.5 (“Training details for PINNACLE”) of the Methods section. The word “masked” is used to indicate that an entity (e.g., edge) is hidden from the model inputs (“unmasked” means that an entity is revealed as an input to the model). Message passing is performed through the unmasked edges in the graphs. With the embeddings generated through such message passing, the masked edges are predicted as “positive” edges (i.e., edges that exist; label = 1) and randomly generated false edges are predicted as “negative” edges (i.e., edges that do not exist; label = 0). **For your convenience, we have included the text in our Methods section here:**

For link prediction, a randomly selected subset of edges is masked (or hidden) from the model, and the model must be able to predict that such edges exist (and that the randomly generated false edges do not exist). [...] Protein-protein edges are randomly split into train (80%), validation (10%), and test (10%) sets. [...] For link prediction, false (or negative) edges have the label of 0 and are randomly generated (via `structured_negative_sampling` function in Pytorch Geometric). The ratio of positive to negative edges is 1:1.

- We agree that clarifying the objective function in the text and codebase is crucial for understanding the inductive biases of the model. To be explicit: link prediction enables PINNACLE to learn the topology in the cell type specific protein interaction networks and metagraph, borrowing from network biology principles (Li et al. *Nature Biomedical Engineering* (2022)), and cell type classification enforces PINNACLE to distinguish proteins activated in different cell type contexts. These inductive biases (i.e., assumptions that models use to make predictions for inputs they have not encountered during training) are discussed in the second paragraph of the results subsection titled “Overview of PINNACLE model.”
- To improve the clarity and reproducibility of our codebase, we have created a new script, `loss.py`, in our GitHub repository containing all of the functions that calculate each term of the loss function (Github commit). The codebase is now easier to interpret and

connect with the equations in the manuscript. **For your convenience, we directly relate the codebase to the equations below:**

- `calc_link_pred_loss()` corresponds to equations 12 (protein-protein edges), 15-16 (cell type to cell type edges, cell type to tissue edges), and 18-19 (tissue to tissue edges, tissue to cell type edges)
- `calc_center_loss()` corresponds to equation 13; and `el_dot()` corresponds to equation 20
- Two of the scripts containing the code to train PINNACLE (namely, `utils.py` and `minibatch_utils.py`) have been revised to clarify which loss is computed and updated during training

2. While the model is stated as (and started from) a protein embedding, all the analysis was done at the cell level and cell data (including expression) is very dominant. The final set of results highlights selecting a 'druggable' (target) cell type based on the 'contextualized protein embedding'. **This makes us wonder how much of the ultimate results are driven by cell type embedding rather than protein embedding.** (That is, this is a meaningful contextualized protein embedding, but the embedding can't be used as many downstream tasks unless the cell type is important.). **We fully realize that the quality of protein embedding per se it's hard to quantify (a similar situation occurs with gene embeddings from single-cell foundation models). We ask that the authors address this in the Discussion section.**

We thank the Reviewer for these valuable comments!

First, we want to clarify that most of our results leverage cell **type specific protein embeddings** (with the exception of Figure 3c, which uses tissue embeddings). We have made the following modifications to minimize misinterpretation:

- Revise the illustration in Figure 1h to make it clear that we are examining protein targets of drugs across cell types
- Copy the illustration in Figure 1h to Figure 5a/d, which shows the results of a protein-level prediction task (binary classification for whether a protein is a candidate therapeutic target for RA or IBD)

We emphasize that Figure 5 shows the results of a protein-level downstream prediction task that provides additional insights about the "importance" of a cell type, which the Reviewer highlights as a meaningful result.

Secondly, we acknowledge the existing research on cell and cell type embeddings in the Discussion section. For instance, there is an exciting area in machine learning with approaches such as SIMBA, Geneformer, CellPLM, scGPT, and scBERT that focus on generating cell-level or gene-level embeddings to advance cell-level downstream prediction tasks, such as batch correction, cell type and cell state clustering, and cell type annotation (Chen et al., *Nature Methods* (2023); Theodoris et al., *Nature* (2023); Cui et al. *bioRxiv* (2023)).

The primary objective of this manuscript is on protein-level embeddings, protein physical interaction networks, and context-specific transfer learning. As such, PINNACLE generates embeddings of cell types and tissues as a secondary objective and solely to inject cellular and tissue organization (via the metagraph) into the **unified protein embedding space**. We provide this distinction in the Discussion. **For your convenience, here is the added text:**

Unlike approaches that generate cell embeddings to advance cell-level downstream tasks, such as batch correction and cell type annotation [66, 152, 153], PINNACLE generates protein representations across cell types for precise protein-level prediction at cell type resolution. As such, PINNACLE generates embeddings of cell types and tissues as a means to inject cellular and tissue organization (via the metagraph) into the unified protein embedding space. To enable cell-level characterization, PINNACLE can be extended to learn cell embeddings. In addition to prioritizing candidate therapeutic targets, PINNACLE's representations can be fine-tuned to identify populations of cells with specific characteristics [...].

In particular, the embedding similarities of marker/housekeep genes between similar and different contexts are quite interesting! We thank the authors for following our suggestion on this topic. **Although the authors claim that the model learns meaningful protein embeddings, it's hard to conclude this based on Response Figure 4. Why is the similarity of housekeeping genes higher than that of marker genes? Why is there a significant drop in the similarity of housekeeping genes between different contexts? We know it's hard to measure the quality of representation learning, but, ideally, both housekeeping genes of similar contexts and different contexts would have comparably high similarities (or there would be a biological argument that their behavior is not "housekeeping"), and marker genes would have larger gaps than housekeeping genes. As the authors noted, PINNACLE is more focused on cell-type contextualized protein embedding, such that although PINNACLE learns a meaningful protein embedding that has more correlation between functionally similar cell types, it will be biased against genes that are unaffected by cell type specificity. Such biases are also shown in Figure S4, where the protein embedding space appears more constructed along cell type expression. Do the authors observe the embedding space of the same genes in different cell types? These points can be addressed in the Discussion.**

We thank the Reviewer again for the great suggestion to evaluate the embedding similarity of proteins with similar functions and recognizing the value of our findings.

Thank you, also, for raising these important questions! The core of this comment (specifically, "Although the authors claim that the model learns meaningful protein embeddings, it's hard to conclude this based on Response Figure 4") is focused on the meaningfulness of PINNACLE's embedding space. We acknowledge this essential point and agree that Response Figure 4 from the previous response document (also known as Supplementary Figure S7) alone may not be sufficiently strong evidence to demonstrate that PINNACLE learns meaningful protein embeddings. In addition to that figure/analysis on the embeddings of proteins with shared

functions/roles, which is admittedly limited, we evaluate the meaningfulness of PINNACLE's embedding in four other ways:

- PINNACLE's representations of proteins activated in the same cell type are spatially enriched in particular regions of the embedding space (Figure 2, Figure 3a-b, and Supplementary Figures S4-S6).
- PINNACLE's representations of tissues are organized according to the tissue hierarchy, and such structure is propagated to PINNACLE's cell type specific protein representations (Figure 3c and Supplementary Figure S9).
- PINNACLE's contextualized protein representations complement 3D structure-based protein representations, improving their ability to differentiate between binding and non-binding proteins (Figure 3d-f and Supplementary Figure S10).
- PINNACLE's contextualized protein representations outperform context-free protein representations in nominating therapeutic targets for rheumatoid arthritis and inflammatory bowel diseases (Figure 4-5 and Supplementary Figures S11-S14).

Taken together, these **five** analyses of PINNACLE's representations provide evidence that the model learns a meaningful embedding space of proteins, cell types, and tissues.

We would like to reiterate that PINNACLE and the underlying protein interaction networks do not explicitly contain information about protein function and that the model does not encounter and functional annotation information during training. This means that our analyses on the embedding similarity of housekeeping/marker genes are zero-shot evaluations of protein function similarity. Following is our attempt at answering your questions while noting that answering them fully would require extensive additional resources and data generation that goes beyond the scope of this manuscript. As such, following your suggestion, we address many of these points in the revised Discussion:

- Regarding your hypothesis, "ideally, both housekeeping genes of similar contexts and different contexts would have comparably high similarities": There are indeed many housekeeping genes (as well as marker genes) with comparably high embedding similarities (see **Response Table 1**). We find that these genes with relatively high embedding similarity in similar contexts (i.e., between T cell types) and different contexts (i.e., between T cell types and non-immune cell types) have low tissue specificity according to the literature (e.g., tissue specificity analysis by the Human Protein Atlas). This raises the question about how these marker and housekeeping genes are defined and the generalizability of these gene lists to other single cell transcriptomic atlases. It seems that further investigation is needed to understand whether these genes have specialized "marker" and "housekeeping" roles. We expect that our evaluation of PINNACLE's embedding space can be improved with more refined lists of housekeeping and marker genes.
- Regarding the question, "Why is there a significant drop in the similarity of housekeeping genes between different contexts?": This is an important point, and we agree that additional follow-up investigation is necessary. In our zero-shot evaluation on the embeddings of proteins with similar functions, the gap in the embedding similarities between "similar contexts" and "different contexts" likely reflects the differences between cell type categories (i.e., immune versus non-immune cell types). PINNACLE is

designed to enforce high embedding similarity between proteins from the same cell type context, not necessarily with shared functions/roles. Still, aligned with prior research examining the tissue-specific function of proteins, an algorithm's accuracy in predicting tissue-specific functions correlates with the algorithm's ability to capture distinct interaction neighborhoods in different tissues (Greene et al. *Nature Genetics* (2015); Zitnik et al. *Bioinformatics* (2017), Pan et al. *Cell Systems* (2022)). As such, when comparing housekeeping and marker genes in "similar contexts" and "different contexts" independently, we observe that housekeeping genes indeed have higher embedding similarities than marker genes, which is in alignment with one of your hypotheses.

We emphasize that these essential questions that our analysis has inspired remain open for further investigation beyond the scope of this manuscript. We discuss these as limitations in the manuscript, specifically in the third paragraph of the results section entitled "PINNACLE's representations capture cellular and tissue organization." **For your convenience, we provide the added text below:**

These analyses suggest that the protein embedding regions in PINNACLE are organized according to cellular contexts, potentially capturing subtle nuances not explicitly included in the training dataset or the model itself. This encompasses the possibility of cell type-dependent roles for proteins, a complexity that can enhance our understanding of protein functions across different biological contexts. Such insights warrant further investigation into proteins with context-specific and non-specific functions.

Lastly, to address your points highlighted in green text:

- You make an important observation that PINNACLE's meaningful protein embeddings are "biased against genes that are unaffected by cell type specificity" because PINNACLE indeed does not generate representations of proteins that are not present in the contextualized PPI networks. The scope of our manuscript is primarily to target biological questions where cell type specificity is essential. With the current setup, it is possible that the genes unaffected by cell type specificity play a role in the cell type yet are not modeled by PINNACLE. This warrants future work to build higher quality cell type specific PPI networks where the nodes and edges are experimentally validated to be cell type specific (or at least play a significant role in the cell type). We already describe this limitation about the construction of our networks in the second paragraph of the Discussion section, but we include another sentence that explicitly discusses the point about the bias that you have pointed out. **For your convenience, we provide the added text below:**

Further, PINNACLE does not currently model proteins that may play a role in the cell type yet are unaffected by cell type specificity.

- To clarify Supplementary Figure S4: The results shown are outputs of a published method, SAFE, that computes the spatial enrichment of nodes in a network with shared characteristics. We apply SAFE to cell type specific *protein* networks constructed based on embedding similarity; these networks have protein nodes that are connected based on the similarity of their corresponding PINNACLE representation. The goal of the analysis is to demonstrate that PINNACLE's cell type specific *protein* representations are

spatially enriched based on cell type context. **We make the following modifications to minimize misinterpretation:**

- Change the header of each panel in Supplementary Figure S4 and Figure 2a-f to start with "Protein embedding region of cell type context: ____" (the blank is filled with the name of the cell type).
- Ensure that Figure 2, main text, and figure captions use consistent language about the SAFE analysis
- Regarding your question, "Do the authors observe the embedding space of the same genes in different cell types?": Yes, for proteins that are in cell type specific networks of different cell types, PINNACLE will generate a separate embedding of those proteins for each cell type that the protein is a part of. These protein embeddings form a unified embedding space, meaning that embedding space of the same proteins in different cell types can be compared to each other. However, note that the PINNACLE model is trained based on self-supervised link prediction instead of supervised prediction of functional protein annotations (e.g., based on gene-to-Gene-Ontology labels), meaning that PINNACLE's objective function does not utilize any information about protein function or identity. Because of that, embeddings of proteins with similar functions are not optimized solely for the purpose of being spatially enriched/localized in the embedding space. However, as our various analyses in the manuscript suggest, it is exciting to see that without prior knowledge, some functional information emerges in the learned representations. A similar capability of learned protein representations has been recently demonstrated for models that apply self-supervised learning to protein sequences (e.g., Rives et al., *PNAS* (2021)).

Minor comments

1. The authors explain their "self-supervised training" as follows: "PINNACLE is trained in a self-supervised manner using cell type identity and graph connectivity (i.e., cell type-specific protein interaction networks and metagraph) as a supervised signal to define positive vs. negative links in self-supervised link prediction."

We find this confusing (and think other readers will also be confused on whether this is supervised learning or not). We found this closer to knowledge-guided training. The goal of the training is representation learning that makes a contextualized protein embedding for each cell type. These properties don't come naturally through training. **In particular, the loss function includes a "center loss", which uses the cell-type label during training to make proteins from the same cell more similar to each other. It might be better not to categorize PINNACLE as self-supervised training and just explain the loss function.**

Thank you for raising this important point. We have changed the language such that PINNACLE is no longer described as a model trained in a "self-supervised manner." **For your convenience, here are the revised sentences in the text** (edits are shown in blue):

- **[Introduction]** PINNACLE is a self-supervised geometric deep learning model adept at generating protein representations through the analysis of protein interactions within various cellular contexts.
- **[Results: “Overview of PINNACLE Model”]** PINNACLE is a self-supervised geometric deep learning model capable of generating protein representations predicated on protein interactions within a spectrum of cell type contexts.
- **[Results: “PINNACLE’s representations capture cellular and tissue organization”]** Although PINNACLE learns protein representations ~~in a self-supervised manner~~ using context-aware protein, cell type, and tissue networks alone, it can...
- **[Methods: “Training details for PINNACLE”]** PINNACLE is trained in a self-supervised manner using cell type identity of the protein interaction networks and graph connectivity of the cell type specific protein interaction networks and metagraph.

Response Tables

Type of Gene	Gene Name	Average Embedding Similarity	
		Similar Contexts	Different Contexts
Marker gene	EZR	0.023 +/- 0.120	0.023 +/- 0.123
	MAP3K5	0.017 +/- 0.125	0.017 +/- 0.131
	RHOG	0.016 +/- 0.085	0.016 +/- 0.130
	KIF2A	0.013 +/- 0.075	0.013 +/- 0.109
	SUZ12	0.013 +/- 0.068	0.013 +/- 0.092
Housekeeping gene	MED10	0.019 +/- 0.099	0.020 +/- 0.100
	DDX18	0.014 +/- 0.062	0.014 +/- 0.093
	RPS21	0.020 +/- 0.095	0.021 +/- 0.097
	NPM1	0.014 +/- 0.053	0.014 +/- 0.093
	RBM8A	0.024 +/- 0.101	0.025 +/- 0.117
	PRPS1	0.011 +/- 0.111	0.011 +/- 0.106
	PPP1CA	0.010 +/- 0.055	0.009 +/- 0.080
	HUWE1	0.012 +/- 0.000	0.011 +/- 0.082

Response Table 1. Average embedding similarities of a marker or housekeeping gene, as specified in the column "Gene Name," in similar contexts (i.e., between T cell types) or different contexts (i.e., between T cell types and non-immune cell types).

Response References

- Hu, W., Liu, B., Gomes, J., Zitnik, M., Liang, P., Pande, V., & Leskovec, J. (2020). Strategies for Pre-training Graph Neural Networks. *International Conference on Learning Representations (ICLR)*.
- Li, M. M., Huang, K., & Zitnik, M. (2022). Graph representation learning in biomedicine and healthcare. *Nature Biomedical Engineering*, 6(12), 1353-1369.
- Chen, H., Ryu, J., Vinyard, M. E., Lerer, A., & Pinello, L. (2023). SIMBA: Single-cell eMBedding Along with features. *Nature Methods*, 1-11.
- Theodoris, C. V., Xiao, L., Chopra, A., Chaffin, M. D., Al Sayed, Z. R., Hill, M. C., ... & Ellinor, P. T. (2023). Transfer learning enables predictions in network biology. *Nature*, 1-9.
- Cui, H., Wang, C., Maan, H., Pang, K., Luo, F., & Wang, B. (2023). scGPT: towards building a foundation model for single-cell multi-omics using generative AI. *bioRxiv*, 2023-04.
- Greene, C. S., Krishnan, A., Wong, A. K., Ricciotti, E., Zelaya, R. A., Himmelstein, D. S., ... & Troyanskaya, O. G. (2015). Understanding multicellular function and disease with human tissue-specific networks. *Nature Genetics*, 47(6), 569-576.
- Zitnik, M., & Leskovec, J. (2017). Predicting multicellular function through multi-layer tissue networks. *Bioinformatics*, 33(14), i190-i198.
- Pan, J., Kwon, J. J., Talamas, J. A., Borah, A. A., Vazquez, F., Boehm, J. S., ... & Hahn, W. C. (2022). Sparse dictionary learning recovers pleiotropy from human cell fitness screens. *Cell Systems*, 13(4), 286-303.
- Rives, A., Meier, J., Sercu, T., Goyal, S., Lin, Z., Liu, J., ... & Fergus, R. (2021). Biological structure and function emerge from scaling unsupervised learning to 250 million protein sequences. *Proceedings of the National Academy of Sciences*, 118(15), e2016239118.

Decision Letter, second revision:

Dear Marinka,

Thank you for submitting your revised manuscript "Contextualizing protein representations using deep learning on protein networks and single-cell data" (N METH-A53552B). It has now been seen by the original referees and their comments are below. The reviewers find that the paper has improved in revision, and therefore we'll be happy in principle to publish it in Nature Methods, pending minor revisions to comply with our editorial and formatting guidelines.

TRANSPARENT PEER REVIEW

Please note: we allow redactions to authors' rebuttal and reviewer comments in the interest of confidentiality. If you are concerned about the release of confidential data, please let us know specifically what information you would like to have removed. Please note that we cannot incorporate redactions for any other reasons. Reviewer names will be published in the peer review files if the reviewer signed the comments to authors, or if reviewers explicitly agree to release their name. For more information, please refer to our FAQ page.

ORCID

Sincerely,
Arunima

Arunima Singh, Ph.D.
Senior Editor
Nature Methods

Reviewer #2 (Remarks to the Author):

We thank the authors for an excellent revision and have no further questions or comments.

The authors did an excellent job at clarifying the training objectives (via the code, which is good). The authors also recognized the current model's limitations, toning down the self-supervised training part. We thank the Authors for their excellent work and all the revisions and looking forward to seeing the paper in press.

Reviewer #2 (Remarks on code availability):

See in earlier reviews.

Final Decision Letter:

Dear Marinka,

I am pleased to inform you that your Article, "Contextual AI models for single-cell protein biology", has now been accepted for publication in Nature Methods. The received and accepted dates will be August 17, 2023 and June 10, 2024. This note is intended to let you know what to expect from us over the next month or so, and to let you know where to address any further questions.

Over the next few weeks, your paper will be copyedited to ensure that it conforms to Nature Methods style. Once your paper is typeset, you will receive an email with a link to choose the appropriate publishing options for your paper and our Author Services team will be in touch regarding any additional information that may be required. It is extremely important that you let us know now whether you will be difficult to contact over the next month. If this is the case, we ask that you send us the contact information (email, phone and fax) of someone who will be able to check the proofs and deal with any last-minute problems.

Please note that *Nature Methods* is a Transformative Journal (TJ). Authors may publish their research with us through the traditional subscription access route or make their paper immediately open access through payment of an article-processing charge (APC). Authors will not be required to make a final decision about access to their article until it has been accepted. Find out more about Transformative Journals

Authors may need to take specific actions to achieve compliance with funder and institutional open access mandates. If your research is supported by a funder that requires immediate open access (e.g. according to Plan S principles) then you should select the gold OA route,

and we will direct you to the compliant route where possible. For authors selecting the subscription publication route, the journal's standard licensing terms will need to be accepted, including self-archiving policies. Those licensing terms will supersede any other terms that the author or any third party may assert apply to any version of the manuscript.

If you are active on Twitter/X, please e-mail me your and your coauthors' handles so that we may tag you when the paper is published.

Best regards,
Arunima

Arunima Singh, Ph.D.
Senior Editor
Nature Methods